# PDZD-8 and TEX-2 regulate endosomal PI(4,5)P$_2$ homeostasis via lipid transport to promote embryogenesis in *C. elegans*

Darshini Jeyasimman[1], Bilge Ercan[1,3], Dennis Dharmawan[1,3], Tomoki Naito[1], Jingbo Sun[1] & Yasunori Saheki [1,2✉]

Different types of cellular membranes have unique lipid compositions that are important for their functional identity. PI(4,5)P$_2$ is enriched in the plasma membrane where it contributes to local activation of key cellular events, including actomyosin contraction and cytokinesis. However, how cells prevent PI(4,5)P$_2$ from accumulating in intracellular membrane compartments, despite constant intermixing and exchange of lipid membranes, is poorly understood. Using the *C. elegans* early embryo as our model system, we show that the evolutionarily conserved lipid transfer proteins, PDZD-8 and TEX-2, act together with the PI(4,5)P$_2$ phosphatases, OCRL-1 and UNC-26/synaptojanin, to prevent the build-up of PI(4,5)P$_2$ on endosomal membranes. In the absence of these four proteins, large amounts of PI(4,5)P$_2$ accumulate on endosomes, leading to embryonic lethality due to ectopic recruitment of proteins involved in actomyosin contractility. PDZD-8 localizes to the endoplasmic reticulum and regulates endosomal PI(4,5)P$_2$ levels via its lipid harboring SMP domain. Accumulation of PI(4,5)P$_2$ on endosomes is accompanied by impairment of their degradative capacity. Thus, cells use multiple redundant systems to maintain endosomal PI(4,5)P$_2$ homeostasis.

[1] Lee Kong Chian School of Medicine, Nanyang Technological University, Singapore 308232, Singapore. [2] Department of Molecular Physiology, Faculty of Life Sciences, Kumamoto University, Kumamoto 860-8556, Japan. [3]These authors contributed equally: Bilge Ercan, Dennis Dharmawan. ✉email: yasunori.saheki@ntu.edu.sg

Cells establish and maintain specific lipid compositions within different cellular membranes[1]. In particular, the unique distributions of seven phosphoinositides (PIPs) within various membrane-bound organelles and the plasma membrane (PM) contribute to the distinct identities and functions of these membranes[2]. PIPs are generated by phosphorylation of the 3, 4, or 5 positions of the inositol head-group of its precursor, phosphatidylinositol, by dedicated kinases and phosphatases that localize to specific membrane compartments[3]. Among the seven PIPs, phosphatidylinositol 4,5-bisphosphate [$PI(4,5)P_2$] is particularly enriched in the PM[2,4]. $PI(4,5)P_2$ recruits proteins involved in non-muscle actomyosin contraction, including anillin, actin, non-muscle myosin (myosin II), RhoA GEF, and RhoA, either directly or indirectly to the PM. This enables contractility of the cell cortex[5–10]. During cell division, recruitment of these proteins to the contractile ring—a unique region of the PM—is essential for cleavage furrow ingression and subsequent cytokinesis[10–13]. $PI(4,5)P_2$ is also essential for cytoskeletal remodeling and clathrin-mediated endocytosis[14–19]. Despite the critical importance of $PI(4,5)P_2$ in these cellular processes, the mechanisms that prevent $PI(4,5)P_2$ from accumulating in intracellular membrane compartments are not fully understood.

Even with the constant intermixing of the PM with endosomal compartments through vesicular transport involving membrane budding and fusion reactions, levels of $PI(4,5)P_2$ in endosomal membranes are remarkably low. This is achieved, in part, through the coordinated actions of various $PI(4,5)P_2$ phosphatases. For example, the $PI(4,5)P_2$ 5-phosphatase, OCRL1, facilitates the removal of $PI(4,5)P_2$ from endosomes by acting at multiple sites, including in endosomes and during late stages of clathrin-mediated endocytosis[20–27]. In addition, synaptojanin, another major $PI(4,5)P_2$ phosphatase with 4- and 5-phosphatase activities, removes $PI(4,5)P_2$ from clathrin-coated membranes and facilitates clathrin disassembly during clathrin-mediated endocytosis[28–32]. Importantly, mutations in OCRL1 and synaptojanin have been linked to human disorders, supporting the critical importance of maintaining cellular $PI(4,5)P_2$ homeostasis [Lowe syndrome and Dent's disease for OCRL1[33–35] and Parkinson's disease for synaptojanin[36–39]]. However, even in the absence of these phosphatases, levels of $PI(4,5)P_2$ in endosomes are not severely elevated in many cell types, indicating that there are other mechanisms for suppressing $PI(4,5)P_2$ accumulation in endosomes. Growing evidence suggests that lipid transfer proteins (LTPs) transport specific lipids, including PIPs, between different cellular compartments (independent of membrane trafficking), thereby helping to maintain unique lipid compositions of cellular membranes[40–50]. Thus, LTPs may act in parallel with $PI(4,5)P_2$ phosphatases to facilitate the removal of $PI(4,5)P_2$ from endosomal membranes to counteract the intermixing of bilayer lipids.

LTPs function primarily at membrane contact sites, where membrane-bound organelles are in close apposition to one another and the PM[51–59]. One evolutionarily conserved family of LTPs that may regulate cellular phosphoinositide homeostasis are proteins that contain the synaptotagmin-like mitochondrial-lipid binding protein (SMP) domain. SMP proteins localize to various membrane contact sites[43,60] and transport a wide variety of lipid species (e.g., glycerolipids and phospholipids) through their characteristic lipid-harboring SMP domain[61–65]. In metazoans, there are four classes of SMP proteins: E-Syts, TMEM24, PDZD8, and TEX2. E-Syts and TMEM24 localize to ER-PM contact sites and facilitate non-vesicular transport of diacylglycerol and phosphatidylinositol, respectively[48,64,66]. The functions of PDZD8 and TEX2 are less clear. In yeast, the TEX2 homolog, Nvj2, localizes to ER-vacuole contact sites (the vacuole is the lysosome equivalent in yeast) at a steady state and moves to ER-Golgi contacts upon ER stress or ceramide overproduction. Nvj2

then facilitates the non-vesicular transport of ceramide from the ER to the Golgi to counteract ceramide toxicity[67,68]. In contrast, PDZD8 localizes to ER-mitochondria and ER-late endosome contacts. PDZD8 has been shown to control $Ca^{2+}$ dynamics at ER-mitochondria contacts and suggested to regulate endosomal maturation at ER-endosome contacts[69–72]. Whether SMP proteins contribute to cellular phosphoinositide homeostasis remains unknown.

In this study, we identified a critical role for SMP proteins in cellular $PI(4,5)P_2$ homeostasis using the *Caenorhabditis elegans* (*C. elegans*) early embryo as our model system. By time-lapse imaging the earliest stages of embryogenesis, which is highly stereotypic in *C. elegans*, and systematically knocking out all four SMP proteins, we found that homologs of PDZD8 and TEX2 (PDZD-8 and TEX-2 in *C. elegans*) acted redundantly to suppress the build-up of endosomal $PI(4,5)P_2$. In the absence of PDZD-8 and TEX-2, $PI(4,5)P_2$ accumulated within endosomal membranes, resulting in ectopic recruitment of proteins that are normally involved in cortical actomyosin contraction (e.g., myosin II, actin, and anillin) to endosomes. Additional depletion of two $PI(4,5)P_2$ phosphatases, namely OCRL-1 and UNC-26 (homologs of OCRL1 and synaptojanin in *C. elegans*), exacerbated $PI(4,5)P_2$ accumulation, leading to massive appearance/clustering of abnormally large $PI(4,5)P_2$-enriched endosomes, lack of myosin II-mediated PM contraction, failure of cytokinesis, and embryonic lethality. In addition, the accumulation of $PI(4,5)P_2$ in endosomes results in major dysfunction of their degradative capacity. We found that both PDZD-8 and TEX-2 localized to the ER and that PDZD-8 is additionally localized to contacts that are formed between the ER and Rab-7-positive late endosomes. Further, our in vitro lipid transport assay revealed that the SMP domain of PDZD-8 transports various PIPs, including $PI(4,5)P_2$, between membranes. Accordingly, specific disruption of the PDZD-8 SMP domain was sufficient to induce aberrant accumulation of $PI(4,5)P_2$ on endosomes. Taken together, these results demonstrate that PDZD-8 regulates endosomal $PI(4,5)P_2$ levels via its SMP domain, acting together with $PI(4,5)P_2$ phosphatases and TEX-2 to counteract the build-up of endosomal $PI(4,5)P_2$. This maintains cellular membrane identities required for normal embryogenesis and cell division.

## Results

**Simultaneous depletion of all four SMP proteins results in cytoplasmic $PI(4,5)P_2$ accumulation in *C. elegans* early embryos.** $PI(4,5)P_2$ within the PM plays a critical role in the coordinated recruitment of various regulators of actomyosin contraction to the cell cortex[73–75]. To investigate the potential role of SMP proteins in $PI(4,5)P_2$ homeostasis in vivo, we chose the *C. elegans* early embryo as our model system because of its highly stereotypic cell divisions and transparency. In *C. elegans* there are four SMP proteins, namely C53B4.4, F55C12.5, R11G1.6, and ESYT-2 (hereafter referred to as PDZD-8, TEX-2, TMEM-24, and ESYT-2, respectively) (Fig. 1a). In addition to the lipid-harboring SMP domain, these proteins contain N-terminal hydrophobic stretches that anchor them to the ER and several other domains. These include C1 and PDZ domains for PDZD-8, a PH domain for TEX-2, C2 and C1 domains for TMEM-24, and multiple C2 domains for ESYT-2 (Fig. 1a). The functions of these proteins in *C. elegans* are largely elusive. Thus, we first examined whether simultaneous depletion of all four SMP proteins altered the distribution of $PI(4,5)P_2$.

Individual knockouts of PDZD-8, TEX-2, and ESYT-2 were obtained via CRISPR/Cas9-mediated gene editing, targeting exons that encode the SMP domain (Supplementary Fig. 1a). Quadruple knock-out (QKO) embryos lacking all four SMP

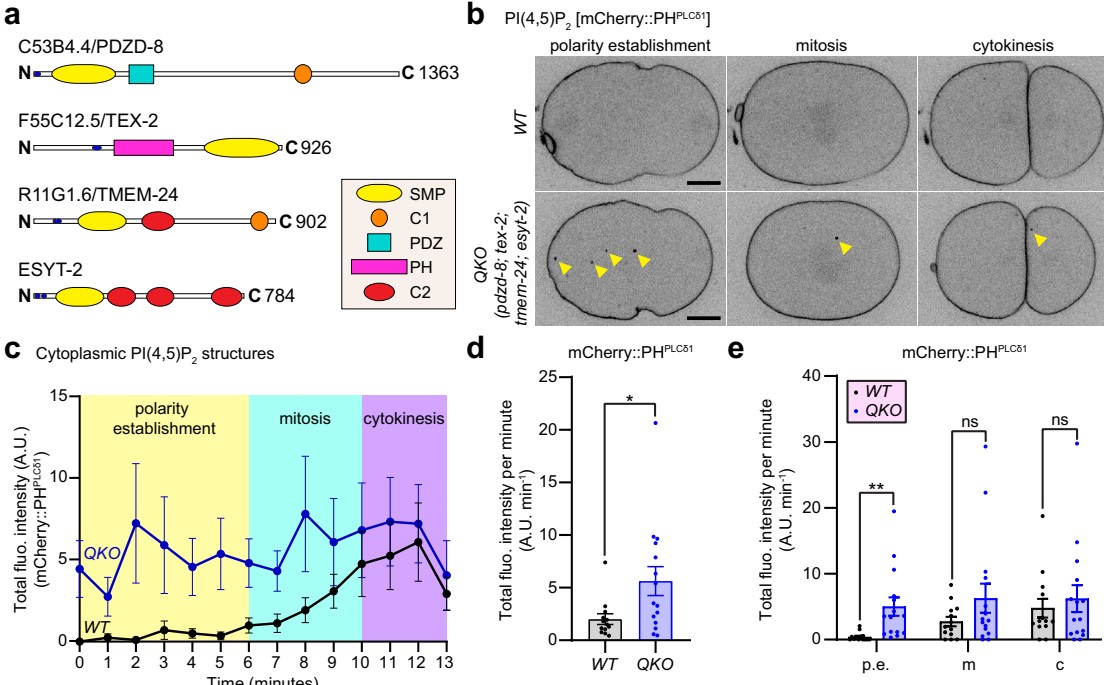

**Fig. 1 Ectopic accumulation of PI(4,5)P₂ in the absence of SMP proteins in *C. elegans* early embryos. a** Schematics of *C. elegans* SMP proteins, C53B4.4/PDZD-8, F55C12.5/TEX-2, R11G1.6/TMEM-24, and ESYT-2. Blue circles indicate hydrophobic stretches that anchor these proteins to the endoplasmic reticulum. Domains for each protein are annotated as indicated. **b** Representative live spinning disc confocal (SDC) images of equatorial planes of early embryos from wild-type control (WT) and mutant lacking all the four SMP proteins [quadruple knockout: QKO (*pdzd-8; tex-2; tmem-24; esyt-2*)], expressing PI(4,5)P₂ biosensor (mCherry::PH$^{PLC\delta1}$). Images of each row are from time-lapse movies of the same embryos at different phases as indicated. Note the presence of ectopic PI(4,5)P₂-positive puncta, indicated by yellow arrowheads, in a QKO embryo compared to a WT embryo. Scale bars, 10 μm. **c** Time-course analysis of the total fluorescence intensity of PI(4,5)P₂-positive puncta, as assessed by mCherry::PH$^{PLC\delta1}$, during polarity establishment (0–6 min), mitosis (6–10 min), and cytokinesis (10–13 min) phases for WT and QKO early embryos as shown in (**b**) [mean ± SEM, *n* = 13 embryos (WT), *n* = 15 embryos (QKO)]. **d** Quantification of the total fluorescence intensity of PI(4,5)P₂-positive puncta, as assessed by mCherry::PH$^{PLC\delta1}$, per minute during early embryogenesis (0–13 min). Comparisons between WT and QKO early embryos are shown [mean ± SEM, *n* = 13 embryos (WT), *n* = 15 embryos (QKO); two-tailed unpaired Student's *t*-test, *\*p* = 0.0274]. **e** Quantification of the total fluorescence intensity of PI(4,5)P₂-positive puncta per minute, as assessed by mCherry::PH$^{PLC\delta1}$, in each phase of the early embryogenesis as indicated for WT and QKO embryos [mean ± SEM, *n* = 13 embryos (WT), *n* = 15 embryos (QKO); two-tailed unpaired Student's *t*-test, *\*\*p* = 0.005523 (p.e.), ns not significant]. p.e., m, and c denote polarity establishment, mitosis, and cytokinesis, respectively.

proteins were generated by crossing these mutants to the *tm10626* mutant, which carries a *tmem-24* deletion allele (Supplementary Fig. 1a). Wild-type control (WT) and QKO embryos that expressed a PI(4,5)P₂ marker—the PH domain of PLCδ1 tagged with mCherry (mCherry::PH$^{PLC\delta1}$) were generated. Early embryogenesis was imaged under spinning disc confocal (SDC) microscopy to assess potential changes in PI(4,5)P₂ distribution.

After fertilization, *C. elegans* embryos undergo a highly stereotypical cell division process that involves the establishment of anterior-posterior polarity, mitosis, and cytokinesis[76–79]. As previously reported, PI(4,5)P₂, as assessed by mCherry::PH$^{PLC\delta1}$, was uniformly distributed across the entire cell cortex in WT embryos throughout this process[80] (Fig. 1b and Supplementary Movie 1). In WT embryos, PI(4,5)P₂ was barely detectable in the cytoplasm, except for very low levels during mitosis and cytokinesis (Fig. 1b and Supplementary Movie 1). In QKO embryos, distinct puncta of PI(4,5)P₂ were clearly present in the cytoplasm (Fig. 1b and Supplementary Movie 1). Up to four distinct PI(4,5)P₂ puncta were observed in the cytoplasm of QKO embryos. Analysis of cytoplasmic mCherry::PH$^{PLC\delta1}$ signals revealed that the aberrant accumulation of cytoplasmic PI(4,5)P₂ in QKO embryos was particularly pronounced during polarity establishment and maintained through mitosis and cytokinesis (Fig. 1c–e and Supplementary Fig. 1b–d). These data suggest that SMP proteins prevent the ectopic accumulation of PI(4,5)P₂ in

the cytoplasm. In their absence, PI(4,5)P₂ aberrantly accumulates in the cytoplasm as distinct puncta.

**Aberrant cytoplasmic PI(4,5)P₂ accumulation is accompanied by ectopic recruitment of actomyosin and anillin**. The PM in *C. elegans* early embryos is enriched in PI(4,5)P₂, and therefore generally associated with high levels of non-muscle myosin II, NMY-2, which controls contractility of the PM and subsequent cytokinetic events[6,81–83]. To investigate whether aberrant accumulation of PI(4,5)P₂ in the cytoplasm results in ectopic myosin recruitment, QKO embryos expressing both mCherry::PH$^{PLC\delta1}$ and NMY-2-tagged with GFP (NMY-2::GFP) were imaged. Strikingly, NMY-2::GFP was present in most mCherry::PH$^{PLC\delta1}$ puncta, indicating ectopic recruitment of NMY-2 to aberrant, PI(4,5)P₂-positive cellular structures in QKO embryos (Fig. 2a and Supplementary Movie 2). This association was maintained throughout the 2-min imaging period before the puncta moved out of the imaging plane (Fig. 2b).

To investigate the structural details of the cytoplasmic NMY-2 puncta [i.e., the aberrant PI(4,5)P₂-positive cellular structures] and whether they are associated with other proteins involved in actomyosin contraction, QKO embryos expressing NMY-2::GFP, wrmScarlet-tagged anillin [wrmScarlet::ANI-1], and RFP-tagged actin [lifeACT::RFP][84,85], were imaged under SDC structural illumination microscopy (SDC-SIM). SDC-SIM revealed that

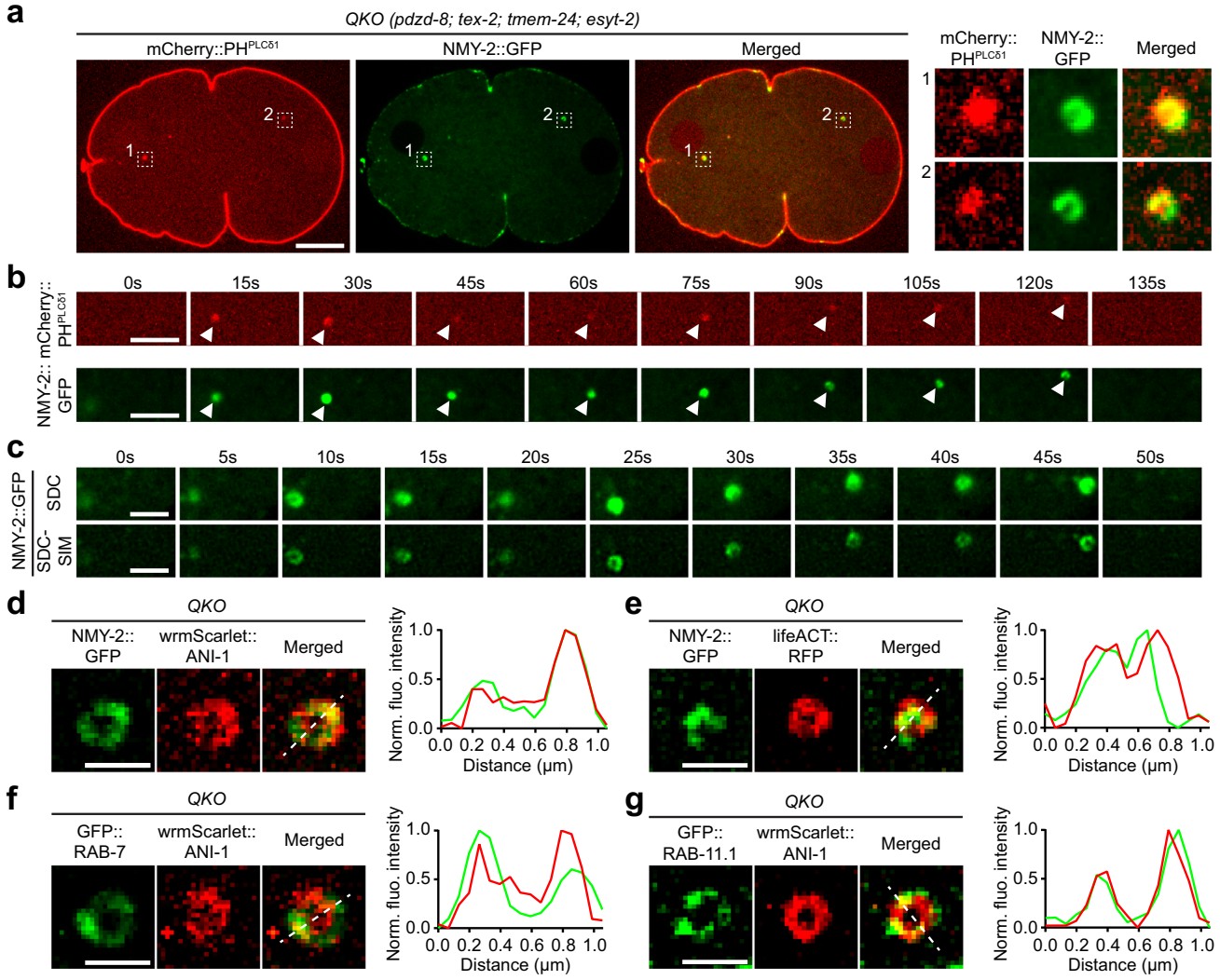

**Fig. 2 Aberrant recruitment of proteins involved in actomyosin contraction to PI(4,5)P₂-positive endosomal membranes in the absence of SMP proteins. a** Left: A representative live spinning disc confocal (SDC) image of an equatorial plane of an early embryo from a mutant lacking all the four SMP proteins [quadruple knock-out: QKO (*pdzd-8; tex-2; tmem-24; esyt-2*)], co-expressing PI(4,5)P₂ biosensor (mCherry::PH^PLCδ1) and NMY-2-tagged with GFP (NMY-2::GFP). The image is from the polarity establishment phase. Right: Magnified insets, showing PI(4,5)P₂-positive puncta, as determined by the presence of mCherry::PH^PLCδ1, that are outlined by indicated white dotted boxes. Scale bar, 10 μm. **b** Time-lapse SDC images of a representative PI(4,5)P₂-positive puncta from an early embryo from QKO, co-expressing mCherry::PH^PLCδ1 and NMY-2::GFP, over the period of 2 min. White arrowheads indicate the positions of the PI(4,5)P₂-positive puncta. Note the extensive overlap of NMY-2::GFP signals with the PI(4,5)P₂-positive puncta. Scale bars, 5 μm. **c** Time-lapse images of representative NMY-2::GFP puncta from an early embryo from QKO over the period of 50 s. Top panel: images from SDC microscopy. Bottom panel: images from SDC structure illumination microscopy (SDC-SIM). Note the lumen inside the NMY-2::GFP puncta in SDC-SIM images. Scale bars, 5 μm. **d, e** Left: Live SDC-SIM images of early embryos from QKO, expressing NMY-2::GFP together with either **d** wrmScarlet-tagged ANI-1 (wrmScarlet::ANI-1) or **e** lifeACT-tagged with RFP (lifeACT::RFP), showing representative NMY-2::GFP puncta that are co-localized with the marker proteins as indicated. Right: Line scan profiles of NMY-2::GFP (green) and either **d** wrmScarlet::ANI-1 or **e** lifeACT::RFP (red) fluorescence signals along the white dotted lines as indicated. Scale bars, 1 μm. **f, g** Left: Live SDC-SIM images of early embryos from QKO, expressing wrmScarlet::ANI-1 together with either **f** GFP-tagged RAB-7 (GFP::RAB-7) or **g** GFP-tagged RAB-11.1 (GFP::RAB-11.1), showing representative wrmScarlet::ANI-1 puncta that are co-localized with the marker proteins as indicated. Right: Line scan profiles of wrmScarlet::ANI-1 (red) and either **f** GFP::RAB-7 or **g** GFP::RAB-11.1 (green) fluorescence signals along the white dotted lines as indicated. Scale bars, 1 μm.

NMY-2::GFP was recruited to highly mobile small vesicles (Fig. 2c). Extensive co-localization between NMY-2::GFP and wrmScarlet::ANI-1 (Fig. 2d), as well as between NMY-2::GFP and lifeACT::RFP (Fig. 2e) was observed, indicating ectopic recruitment of multiple proteins involved in actomyosin contraction to the PI(4,5)P₂-positive vesicles.

**PI(4,5)P₂ accumulates on endosomal membranes in the absence of SMP proteins.** To further investigate the identity of the cellular membranes that were decorated by cytoplasmic

PI(4,5)P₂, ANI-1, actin, and NMY-2, we examined the extent of co-localization between wrmScarlet::ANI-1 (or NMY-2::GFP) and various organelle markers, including lysotracker dye for lysosomes, mitotracker dye for mitochondria, GFP-tagged EEA-1 (GFP::EEA-1) for early endosomes[86], GFP-tagged RAB-7 (GFP::RAB-7) for late endosomes, and GFP-tagged RAB-11.1 (GFP::RAB-11.1) for recycling endosomes[87]. No apparent co-localization was observed between NMY-2::GFP or wrmScarlet::ANI-1 and lysotracker, mitotracker, and GFP::EEA-1 (Supplementary Fig. 2a, b). However, 81.25% of wrmScarlet::ANI-1

puncta ($n = 64$ puncta from 26 embryos) were transiently positive for GFP::RAB-11.1, whereas 22.41% of wrmScarlet::ANI-1 puncta ($n = 58$ puncta from 27 embryos) were transiently positive for GFP::RAB-7, indicating that PI(4,5)P$_2$ accumulated on endosomal structures that possessed markers of late and recycling endosomes (Fig. 2f, g).

The vast majority of PI(4,5)P$_2$ in the cell is generated by phosphatidylinositol 4-phosphate-5 kinases (type I PIP-kinases or PIP5Ks)[3]. In *C. elegans*, PPK-1 is the sole PIP5K[88]. To determine the subcellular localization of PPK-1 in early embryos, we introduced a sequence encoding EGFP-tagged PPK-1 (PPK-1::EGFP) into its endogenous locus by using CRISPR/Cas9 (see Methods). In WT embryos, PPK-1::EGFP signals were detected exclusively on the PM, where PI(4,5)P$_2$ is most enriched (Supplementary Fig. 2c). In contrast, cytoplasmic PI(4,5)P$_2$ puncta in QKO embryos were not associated with PPK-1, suggesting that aberrant accumulation of PI(4,5)P$_2$ in QKO embryos was not caused by ectopic recruitment of PPK-1 to endosomes (Supplementary Fig. 2c).

These results suggest that SMP proteins inhibit the build-up of PI(4,5)P$_2$ on endosomes. In their absence, PI(4,5)P$_2$ aberrantly accumulates on endosomes, leading to ectopic recruitment of various proteins involved in actomyosin contraction (including anillin, actin, and myosin) to these membrane compartments.

**Redundant functions of PDZD-8 and TEX-2.** Our results suggest that SMP proteins are essential for preventing the accumulation of PI(4,5)P$_2$ and non-muscle myosin on endosomes. To identify which SMP proteins are involved in this process, we generated single, double, and triple knockouts of the four SMP proteins. These mutants were systematically examined for the presence of NMY-2::GFP puncta in early embryos. WT animals, as well as all single knockouts showed no cytoplasmic accumulation of NMY-2::GFP (Supplementary Fig. 3a, b). Double knock-out of *esyt-2* and *tmem-24* had no effect, whereas *pdzd-8* and *tex-2* double knockouts (DKOs) exhibited aberrant accumulation of NMY-2::GFP puncta (Fig. 3a–d, Supplementary Fig. 3a–c and Supplementary Movie 3). Triple knockouts of SMP proteins showed aberrant accumulation of NMY-2::GFP puncta only when *pdzd-8* and *tex-2* were simultaneously depleted (Supplementary Fig. 3a, b). Further analysis of NMY-2::GFP puncta revealed that both QKO and *pdzd-8; tex-2* DKOs showed aberrant cytoplasmic accumulation of NMY-2::GFP as distinct puncta (Fig. 3b-d and Supplementary Fig. 3c). Interestingly, the brood size was slightly reduced in both DKO and QKO compared to WT (Supplementary Fig. 3d). However, all the eggs hatched successfully and early embryos showed normal cell division in DKO and QKO (Supplementary Fig. 3e, f). These data suggest that PDZD-8 and TEX-2 function redundantly to suppress endosomal PI(4,5)P$_2$ levels. For most subsequent assays we used *pdzd-8; tex-2* DKOs. For some experiments, *pdzd-8; tex-2; tmem-24; esyt-2* QKO animals were used, each time yielding phenotypes similar to *pdzd-8; tex-2* DKO.

**PDZD-8 and TEX-2 act together with the PI(4,5)P$_2$ phosphatases, OCRL-1, and UNC-26/synaptojanin, to prevent the build-up of endosomal PI(4,5)P$_2$.** Levels of PI(4,5)P$_2$ in the PM and endosomes are tightly controlled by a balance of activities between PI(4,5)P$_2$ kinases and phosphatases, which localize to different cellular compartments. Several PI(4,5)P$_2$ phosphatases, including OCRL1 and synaptojanin, have been shown to suppress the accumulation of PI(4,5)P$_2$ on endosomal membranes[20–32]. We hypothesized that PDZD-8 and TEX-2 may act together with the PI(4,5)P$_2$ phosphatases, OCRL-1 and UNC-26 (homologs of OCRL1 and synaptojanin in *C. elegans*), to inhibit the build-up of PI(4,5)P$_2$ on endosomes. To examine this possibility, we simultaneously depleted OCRL-1, UNC-26, PDZD-8, and TEX-2, and

the resulting phenotype was compared to that of OCRL-1 and UNC-26 depletion alone.

WT animals, *unc-26* null mutants[89], and *unc-26; pdzd-8; tex-2* triple mutants (hereafter referred to as *unc-26; DKO*) were treated with RNA interference (RNAi) against OCRL-1. The resulting early embryos were imaged using SDC microscopy to examine the distribution of PI(4,5)P$_2$ (mCherry::PH$^{PLC\delta1}$) and NMY-2 (NMY-2::GFP). In WT or *unc-26* mutant early embryos treated with OCRL-1 RNAi, PI(4,5)P$_2$ and NMY-2 remained tightly associated with the PM, although some cytoplasmic PI(4,5)P$_2$-positive structures were observed in *unc-26* mutant embryos treated with OCRL-1 RNAi compared to WT embryos treated with OCRL-1 RNAi (Fig. 4a–c and Supplementary Fig. 4a; Supplementary Movie 4; compare with Fig. 1b and Supplementary Movie 1). In contrast, *unc-26; DKO* early embryos treated with OCRL-1 RNAi showed massive accumulation of exceptionally large PI(4,5)P$_2$-positive vesicles with diameters that range from 0.5 to 3.8 μm in the cytoplasm (Fig. 4d). This was often accompanied by pronounced NMY-2 recruitment (Fig. 4a–c, Supplementary Fig. 4a and Supplementary Movie 4). During the polarity establishment phase, multiple large PI(4,5)P$_2$-positive vesicles appeared in the cytoplasm. These were often decorated with NMY-2::GFP. Once embryos entered mitosis, the PI(4,5)P$_2$-positive vesicles clustered at the cellular periphery and became a sink for NMY-2::GFP, absorbing most NMY-2::GFP from the cell cortex (Fig. 4a, Supplementary Fig. 4a, and Supplementary Movie 4). The aberrant clusters of PI(4,5)P$_2$-positive vesicles persisted throughout mitosis and cytokinesis, whereas NMY-2::GFP dissociated from these vesicles as cells entered the cytokinesis phase (Fig. 4a–c, Supplementary Fig. 4a, and Supplementary Movie 4).

Imaging mCherry::PH$^{PLC\delta1}$ and NMY-2::GFP at a particular time point during mitosis revealed that NMY-2 was enriched only in a subset of the large PI(4,5)P$_2$-positive vesicles. This heterogeneity was explained by real-time imaging of NMY-2::GFP and the PI(4,5)P$_2$ vesicles over time. The association of NMY-2::GFP to large PI(4,5)P$_2$-positive vesicles was often transient, generally lasting only ~10 s (Fig. 4e and Supplementary Movie 5). This suggested that the recruitment of NMY-2::GFP to PI(4,5)P$_2$-positive membranes was unstable and highly dynamic. We quantified the ratio of PI(4,5)P$_2$ levels in the PM and cytoplasm (including the large PI(4,5)P$_2$-positive vesicles) using mCherry::PH$^{PLC\delta1}$ and found a dramatic increase in PI(4,5)P$_2$ levels in intracellular membrane compartments in *unc-26; DKO* mutants treated with OCRL-1 RNAi compared to WT or *unc-26* single mutants treated with OCRL-1 RNAi (Fig. 4f). Anillin/ANI-1, which acts as a scaffold protein for linking the contractile actomyosin machinery to the PM, is normally recruited to the PM via its direct interaction with PI(4,5)P$_2$[5,11,90–92]. Accordingly, *unc-26; DKO* mutants treated with OCRL-1 RNAi showed aberrant accumulation of wrmScarlet::ANI-1 in the cytoplasm, while *unc-26* mutants treated with OCRL-1 RNAi showed no such strong accumulation (Supplementary Fig. 4b).

No cytokinesis defects were observed in WT or *unc-26* mutants. In contrast, occasional cytokinesis defects were observed in *unc-26; DKO* mutants, where 10% of embryos showed no cleavage furrow formation (Supplementary Fig. 4c). Strikingly, massive failures in cytokinesis were observed in *unc-26; DKO* mutants treated with OCRL-1 RNAi compared to WT or *unc-26* mutants treated with OCRL-1 RNAi. In *unc-26; DKO* mutants treated with OCRL-1 RNAi, complete cell division occurred only in 53.6% of imaged embryos, whereas 25% and 21.4% of embryos showed retraction of the cleavage furrow and no cleavage furrow formation ($n = 28$ embryos), respectively (Fig. 4g, h). In contrast, OCRL-1 RNAi treatment did not affect cytokinesis in WT animals and resulted in mild defects in *unc-26* mutants, where

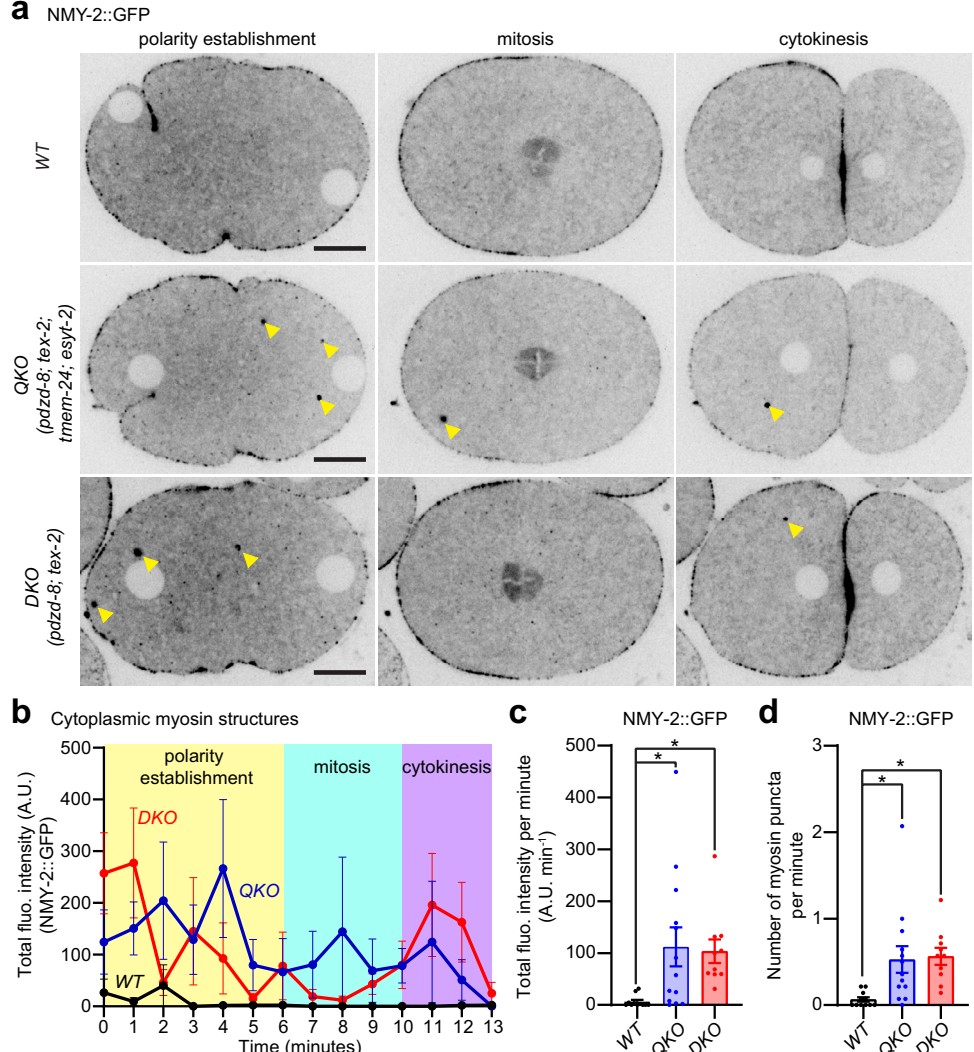

**Fig. 3 Redundant functions of PDZD-8 and TEX-2. a** Representative live spinning disc confocal (SDC) images of equatorial planes of early embryos from wild-type control (WT), mutant lacking all the four SMP proteins [quadruple knock-out: QKO (*pdzd-8; tex-2; tmem-24; esyt-2*)] and mutant lacking PDZD-8 and TEX-2 [double knock-out: DKO (*pdzd-8; tex-2*)], expressing NMY-2-tagged with GFP (NMY-2::GFP). Images of each row are from time-lapse movies of the same embryos at different phases as indicated. Note the presence of ectopic NMY-2::GFP puncta, indicated by yellow arrowheads, in QKO and DKO embryos compared to a WT embryo. Scale bars, 10 μm. **b** Time-course analysis of the total fluorescence intensity of NMY-2::GFP puncta during polarity establishment (0–6 min), mitosis (6–10 min), and cytokinesis (10–13 min) phases for WT, QKO, and DKO early embryos as shown in (**a**) [mean ± SEM, *n* = 11 embryos (WT), *n* = 13 embryos (QKO), *n* = 10 embryos (DKO)]. **c** Quantification of the total fluorescence intensity of NMY-2::GFP puncta per minute during early embryogenesis (0–13 min). Comparisons between WT, QKO, and DKO early embryos are shown [mean ± SEM, *n* = 11 embryos (WT), *n* = 13 embryos (QKO), *n* = 10 embryos (DKO); Dunnett's multiple comparisons test, *\*p* = 0.0171 (QKO), *\*p* = 0.0417 (DKO)]. **d** Quantification of the number of NMY-2::GFP puncta per minute during early embryogenesis (0–13 min). Comparisons between WT, QKO, and DKO early embryos are shown [mean ± SEM, *n* = 11 embryos (WT), *n* = 13 embryos (QKO), *n* = 10 embryos (DKO); Dunnett's multiple comparisons test, *\*p* = 0.0123 (QKO), *\*p* = 0.0113 (DKO)].

complete cell division occurred in 76% of imaged embryos (*n* = 30 embryos) (Fig. 4g, h).

Finally, to determine whether large aberrant PI(4,5)P$_2$-positive vesicles in *unc-26; DKO* early embryos treated with OCRL-1 RNAi were originated from the PM, endocytosis was blocked by treating them with RNAi to deplete dynamin, an essential protein that catalyzes membrane fission during clathrin-mediated endocytosis[93]. RNAi of DYN-1, a sole homolog of dynamin in *C. elegans*, resulted in severe early embryonic lethality after fertilization, making it difficult to isolate early embryos for microscopy. Thus, one-cell embryos were directly imaged within the uterus of adult worms without dissection (Supplementary

Fig. 5a). Remarkably, cytoplasmic accumulation of PI(4,5)P$_2$ was largely suppressed by DYN-1 RNAi (Supplementary Fig. 5a, b), suggesting that the bulk of PI(4,5)P$_2$ accumulated in the cytoplasm of *unc-26; DKO* early embryos treated with OCRL-1 RNAi originated from the PM.

These results support our hypothesis that PDZD-8 and TEX-2 act together with PI(4,5)P$_2$ phosphatases to inhibit the build-up of PI(4,5)P$_2$ on endosomes after PM PI(4,5)P$_2$ enters endosomal systems through endocytosis. Simultaneous depletion of all these components results in massive accumulation of PI(4,5)P$_2$ within large, abnormal vesicles, leading to aberrant distribution of actomyosin and cytokinesis defects.

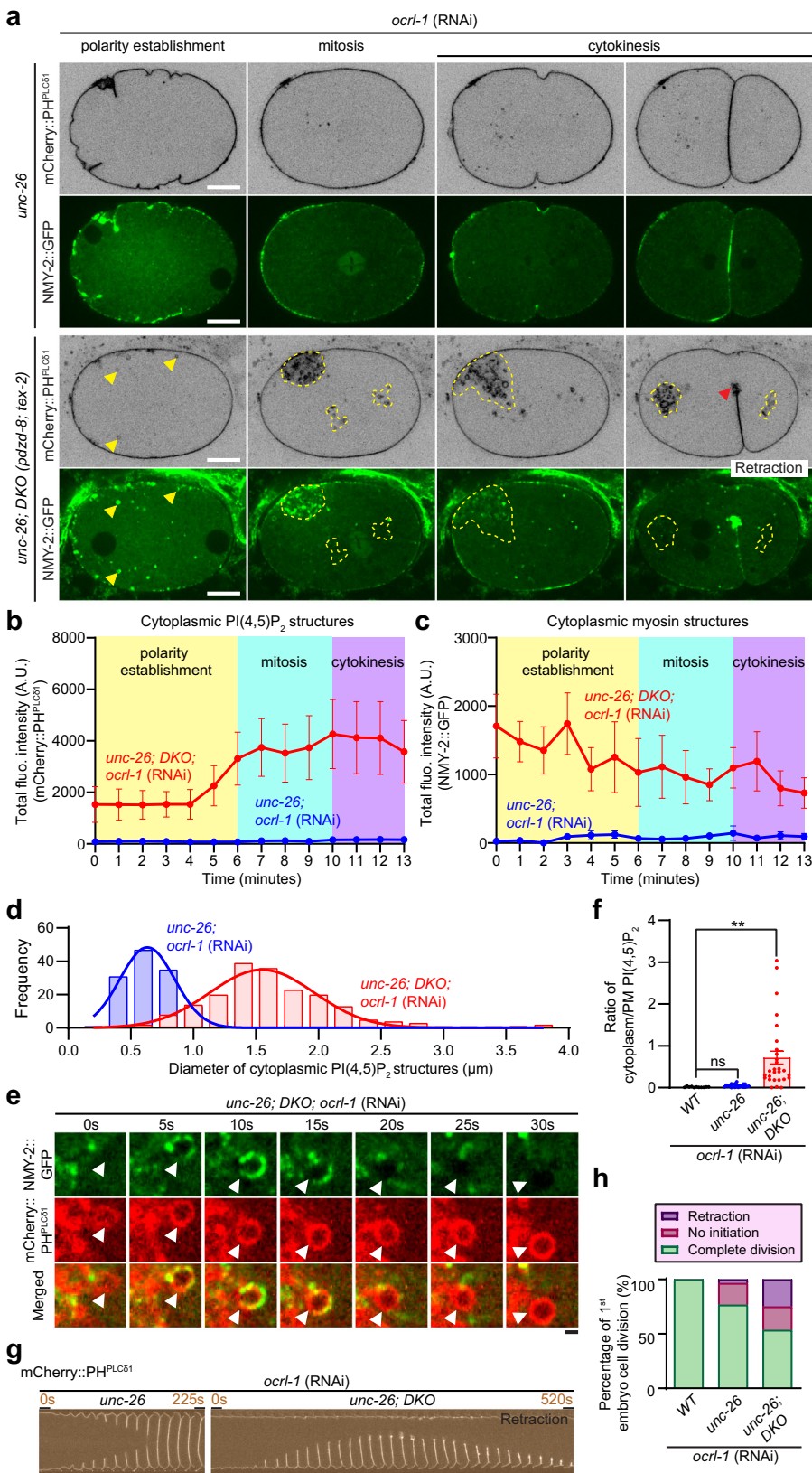

**Contractility of the cell cortex is lost in the absence of PDZD-8, TEX-2, and PI(4,5)P₂ phosphatases**. During the first cell division of a *C. elegans* embryo, a network of actin and non-muscle myosin forms at the cell cortex and induces actomyosin contractility[76]. This actomyosin contractility then regulates the periodic contraction of the PM during polarity establishment (i.e.,

cortical ruffling)[6,82], which leads to ingression of the pseudo-cleavage furrow. Mislocalization of NMY-2 from the PM to aberrant, large PI(4,5)P₂-positive vesicles in *unc-26;* DKO mutants treated with OCRL-1 RNAi (Fig. 4a) indicated that actomyosin contractility at the cell cortex may be affected in these animals. Thus, we asked whether simultaneous depletion of

**Fig. 4 Simultaneous depletion PDZD-8, TEX-2, and PI(4,5)P$_2$ phosphatases results in massive accumulation of endosomal PI(4,5)P$_2$ and cytokinetic defects. a** Representative live spinning disc confocal (SDC) images of equatorial planes of early embryos from OCRL-1 RNAi-treated *unc-26* mutants and OCRL-1 RNAi-treated *unc-26;* double knock-out [DKO (*pdzd-8; tex-2*)] mutants, co-expressing PI(4,5)P$_2$ biosensor (mCherry::PH$^{PLCδ1}$) and NMY-2-tagged with GFP (NMY-2::GFP). Images of each row are from time-lapse movies of the same embryos at different phases as indicated. Yellow arrowheads in the polarity establishment phase indicate PI(4,5)P$_2$-positive vesicles that are co-localized with NMY-2::GFP. Yellow dotted lines in mitosis and cytokinesis phases indicate clustering of PI(4,5)P$_2$-positive vesicles. The red arrowhead shows a cleavage furrow that eventually retracts. Scale bars, 10 μm. **b, c** Time-course analysis of the total fluorescence intensity of **b** mCherry::PH$^{PLCδ1}$ and **c** NMY-2::GFP during polarity establishment (0–6 min), mitosis (6–10 min), and cytokinesis (10–13 min) phases for OCRL-1 RNAi-treated *unc-26* mutants and OCRL-1 RNAi-treated *unc-26;* DKO mutants early embryos as shown in (**a**) [mean ± SEM, $n = 14$ embryos (*unc-26; ocrl-1* RNAi), $n = 14$ embryos (*unc-26; DKO; ocrl-1* RNAi)]. **d** Histogram showing the size distribution of cytoplasmic PI(4,5)P$_2$ structures from early embryos as indicated. Gaussian distribution curve is superimposed on the histogram [$n = 138$ structures (*unc-26; ocrl-1* RNAi), $n = 190$ structures (*unc-26; DKO; ocrl-1* RNAi)]. **e** Time-lapse SDC images of a representative PI(4,5)P$_2$-positive vesicle from an early embryo from OCRL-1 RNAi-treated *unc-26; DKO* mutants, co-expressing mCherry::PH$^{PLCδ1}$ and NMY-2::GFP, over the period of 30 sec. White arrowheads indicate the positions of a PI(4,5)P$_2$-enriched vesicle that transiently recruits NMY-2::GFP (5, 10, and 15 s). Scale bar, 1 μm. **f** Quantification of the ratio of PI(4,5)P$_2$ levels in the cytoplasm (including PI(4,5)P$_2$-positive vesicles) to PI(4,5)P$_2$ levels in the plasma membrane (PM) at the stage prior to cytokinesis, as assessed by mCherry::PH$^{PLCδ1}$, in early embryos from OCRL-1 RNAi-treated wild-type control (WT) animals, OCRL-1 RNAi-treated *unc-26* mutants, and OCRL-1 RNAi-treated *unc-26; DKO* mutants [mean ± SEM, $n = 11$ embryos (WT; *ocrl-1* RNAi), $n = 20$ embryos (*unc-26; ocrl-1* RNAi), $n = 28$ embryos (*unc-26; DKO; ocrl-1* RNAi); Dunnett's multiple comparisons test, **$p = 0.0021$, ns denotes not significant]. **g** Time-lapse montages of equatorial planes of early embryos from indicated mutants. Note the failure of cytokinesis in early embryos from OCRL-1 RNAi-treated *unc-26; DKO* mutants. Scale bars, 10 μm. **h** Quantification of the first embryonic cell divisions in early embryos from indicated conditions [$n = 11$ embryos (WT; *ocrl-1* RNAi), $n = 30$ embryos (*unc-26; ocrl-1* RNAi), $n = 28$ embryos (*unc-26; DKO; ocrl-1* RNAi)].

PDZD-8 and TEX-2 together with the PI(4,5)P$_2$ phosphatases, OCRL-1 and UNC-26, affected cortical ruffling.

WT animals, *unc-26* mutants, and *unc-26; DKO* mutants were treated with OCRL-1 RNAi, and resulting early embryos were imaged under SDC microscopy to examine PM contractility. Most of the *unc-26* mutants treated with OCRL-1 RNAi exhibited high contractility of the PM, leading to extensive cortical ruffling during polarity establishment followed by the expansion of a noncontractile area throughout the cortex during mitosis, similar to WT animals treated with OCRL-1 RNAi (Fig. 5a). Strikingly, *unc-26; DKO* mutants treated with OCRL-1 RNAi showed loss of PM contractility, resulting in the total absence of cortical ruffling and the lack of pseudocleavage furrow ingression (Fig. 5a). We compared the perimeter of embryos at polarity establishment to the perimeter of the same embryos at mitosis (i.e., cortical ruffling index), revealing that cortical ruffling was abolished in *unc-26; DKO* mutants treated with OCRL-1 RNAi (Fig. 5b). These results demonstrate that actomyosin contractility was disrupted within the cell cortex following the simultaneous depletion of OCRL-1, UNC-26, PDZD-8, and TEX-2.

To determine the impact of failed cytokinesis (Fig. 4h) and loss of cortical ruffling (Fig. 5b) on embryonic development, we compared the percentage of eggs that hatched after OCRL-1 RNAi treatment in WT, *unc-26* mutants, and *unc-26; DKO* mutants. Reduced brood size was observed in *unc-26* mutants (33.7% of WT) and *unc-26; DKO* mutants (34.8% of WT) (Supplementary Fig. 5c). Nearly 100% of the eggs hatched successfully in these mutants (Supplementary Fig. 5d). Upon OCRL-1 RNAi treatment, brood size was further reduced equally in *unc-26* mutants (18.4% of WT) and *unc-26; DKO* mutants (17.8% of WT) (Fig. 5c). However, OCRL-1 RNAi treatment resulted in a massive reduction in the percentage of eggs that hatched in *unc-26; DKO* mutants (~50% reduction) but not in *unc-26* mutants (only ~10% reduction) (Fig. 5d). Thus, simultaneous depletion of PDZD-8, TEX-2, UNC-26, and OCRL-1 causes catastrophic consequences in embryonic development.

**PDZD-8 localizes to ER-late endosome contacts in early embryos.** Our results suggest important roles for PDZD-8 and TEX-2 in endosomal PI(4,5)P$_2$ homeostasis. To gain mechanistic insights into their functions, we examined the subcellular

localization of these proteins in early embryos by using CRISPR/Cas9-based methods to introduce sequences encoding mNeonGreen-tagged PDZD-8 (PDZD-8::mNeonGreen) or EGFP-tagged TEX-2 (TEX-2::EGFP) into their endogenous loci (see Methods). Both PDZD-8::mNeonGreen and TEX-2::EGFP signals were detected in nerve rings within the head region of adult worms, confirming the successful tagging of the endogenous loci (Supplementary Fig. 6a). Importantly, PDZD-8::mNeon-Green and TEX-2::EGFP signals were detected, albeit faintly, in early embryos, allowing us to determine their localization at endogenous expression levels. Both PDZD-8::mNeonGreen and TEX-2::EGFP were found in the ER based on extensive overlap with the ER marker, mCherry::SP-12[94], consistent with the localization of their mammalian homologs to the ER[69,95] (Supplementary Fig. 6b).

As a complementary approach, we assessed the subcellular localization of PDZD-8 and TEX-2 by overexpressing EGFP-tagged PDZD-8 (PDZD-8::EGFP) or TEX-2::EGFP heterologously in COS-7 cells. PDZD-8::EGFP and TEX-2::EGFP co-localized with the ER marker, RFP-Sec61β, consistent with their localization in *C. elegans* early embryos (Supplementary Fig. 6b–d). Given the impact of PDZD-8 and TEX-2 on endosomal PI(4,5)P$_2$ homeostasis (Fig. 2f, g), we examined the potential association of PDZD-8::EGFP and TEX-2::EGFP with endosomes by co-expressing various endosome markers. Strikingly, co-expression of PDZD-8::EGFP and the late endosome marker, mCherry-Rab7A, resulted in extensive co-localization of these two proteins (Supplementary Fig. 6c). In contrast, co-expression of PDZD-8::EGFP with iRFP-FRB-Rab-5 (an early endosome marker) or DsRed-Rab-11 (a recycling endosome marker) did not affect the localization of PDZD-8::EGFP (Supplementary Fig. 6c). TEX-2::EGFP did not particularly associate with any of these endosome markers (Supplementary Fig. 6d).

Based on the results obtained in COS-7 cells, we examined whether PDZD-8 additionally localizes to the sites of contact between the ER and late endosomes in *C. elegans* early embryos. To this end, we generated another knock-in strain that expresses mCherry-tagged RAB-7 from its endogenous locus (mCherry::RAB-7) (see Methods). mCherry::RAB-7 signals were well detected on vesicular structures in early embryos (Fig. 6a), allowing us to determine the association of the late endosome marker, RAB-7, and PDZD-8 both at

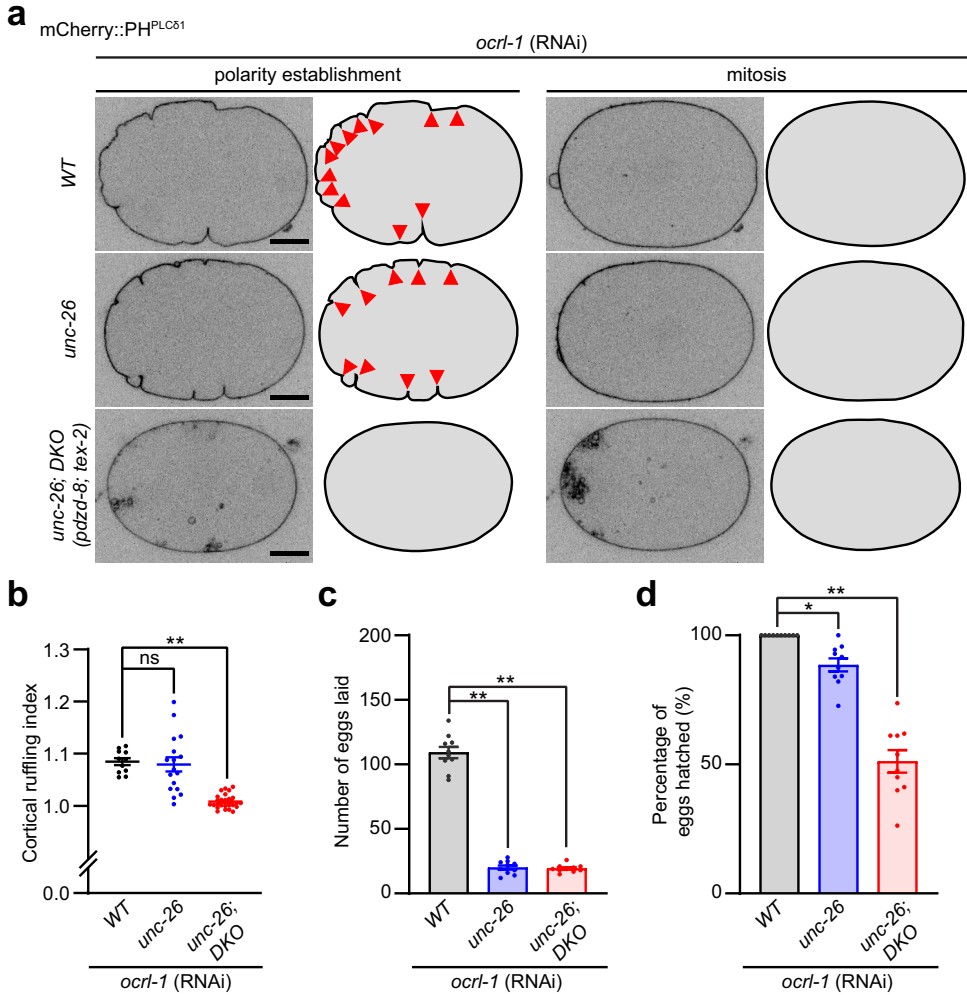

**Fig. 5 Simultaneous depletion of PDZD-8, TEX-2, and PI(4,5)P₂ phosphatases results in reduced cortical ruffling and embryonic lethality. a** Representative live spinning disc confocal (SDC) images of equatorial planes of early embryos from OCRL-1 RNAi-treated wild-type control (WT) animals, OCRL-1 RNAi-treated *unc-26* mutants and OCRL-1 RNAi-treated *unc-26*; double knock-out [DKO (*pdzd-8; tex-2*)] mutants, expressing PI(4,5)P₂ biosensor (mCherry::PH^PLCδ1). Images of each row are from time-lapse movies of the same embryos at different phases as indicated. Cartoons depicting the outline of the embryos. Red arrowheads indicate cortical ruffling. Note the loss of cortical ruffling in OCRL-1 RNAi-treated *unc-26*; DKO mutants, during the polarity establishment phase. Scale bars, 10 μm. **b** Quantification of cortical ruffling. The cortical ruffling index was calculated by taking the ratio of the perimeter of the plasma membrane of embryos from polarity establishment phase and mitosis phase, as assessed in (**a**) (see Methods) [mean ± SEM, *n* = 11 embryos (WT; *ocrl-1* RNAi), *n* = 17 embryos (*unc-26*; *ocrl-1* RNAi), *n* = 26 embryos (*unc-26*; DKO; *ocrl-1* RNAi); Dunnett's multiple comparisons test, **p < 0.0001 (*unc-26*; DKO; *ocrl-1* RNAi), ns denotes not significant]. **c** Quantification of the number of eggs laid by OCRL-1 RNAi-treated WT animals, OCRL-1 RNAi-treated *unc-26* mutants and OCRL-1 RNAi-treated *unc-26*; DKO mutants [mean ± SEM, *n* = 10 animals for all conditions; Dunnett's multiple comparisons test, **p < 0.0001 (*unc-26*; *ocrl-1* RNAi), **p < 0.0001 (*unc-26*; DKO; *ocrl-1* RNAi)]. **d** Quantification of the percentage of eggs successfully hatched in indicated conditions as in (**c**) [mean ± SEM, *n* = 10 animals for all conditions; Dunnett's multiple comparisons test, *p = 0.0182 (*unc-26*; *ocrl-1* RNAi), **p < 0.0001 (*unc-26*; DKO; *ocrl-1* RNAi)].

endogenous expression levels. The ER marker, GFP::C34B2.10, frequently wrapped around late endosomes that were marked by mCherry::RAB-7, supporting the occurrence of ER-late endosome contacts in early embryos (Fig. 6a). Remarkably, distinct PDZD-8::mNeonGreen puncta were present on a small fraction of mCherry::RAB-7-positive late endosomes (Fig. 6b). The association was often transient, with the longest association lasting up to 45 s during the imaging period (Fig. 6c and Supplementary Movie 6). Because PDZD-8::mNeonGreen localizes to the ER (Supplementary Fig. 6b), these results support the presence of PDZD-8 at ER-late endosome contacts. In contrast, TEX-2::EGFP showed no particular association with mCherry::RAB-7-positive late endosomes in early embryos, consistent with the results obtained from the COS-7 cells (Supplementary Fig. 6e).

Collectively, these results demonstrate that both TEX-2 and PDZD-8 localize to the ER and that PDZD-8 also localizes to ER-late endosome contacts in early embryos.

**The SMP domain of PDZD-8 transports various phosphoinositides, including PI(4,5)P₂.** PDZD-8 possesses an SMP domain, which may facilitate lipid transport between closely apposed membranes at ER-late endosome contacts. We purified the SMP domain of PDZD-8 (SMP_PDZD-8) and asked whether it could transport PIPs, including PI(4,5)P₂, between membranes. We assessed transport via a fluorescence resonance energy transfer (FRET)-based in vitro lipid transport assay. We prepared donor liposomes [DOPC (94%), PIP [either one of the following lipids: PI(3)P, PI(4)P, PI(3,5)P₂, PI(4,5)P₂, or PI(3,4,5)P₃] (4%), and Rhodamine-PE (2%)] and acceptor liposomes [DOPC (100%)]

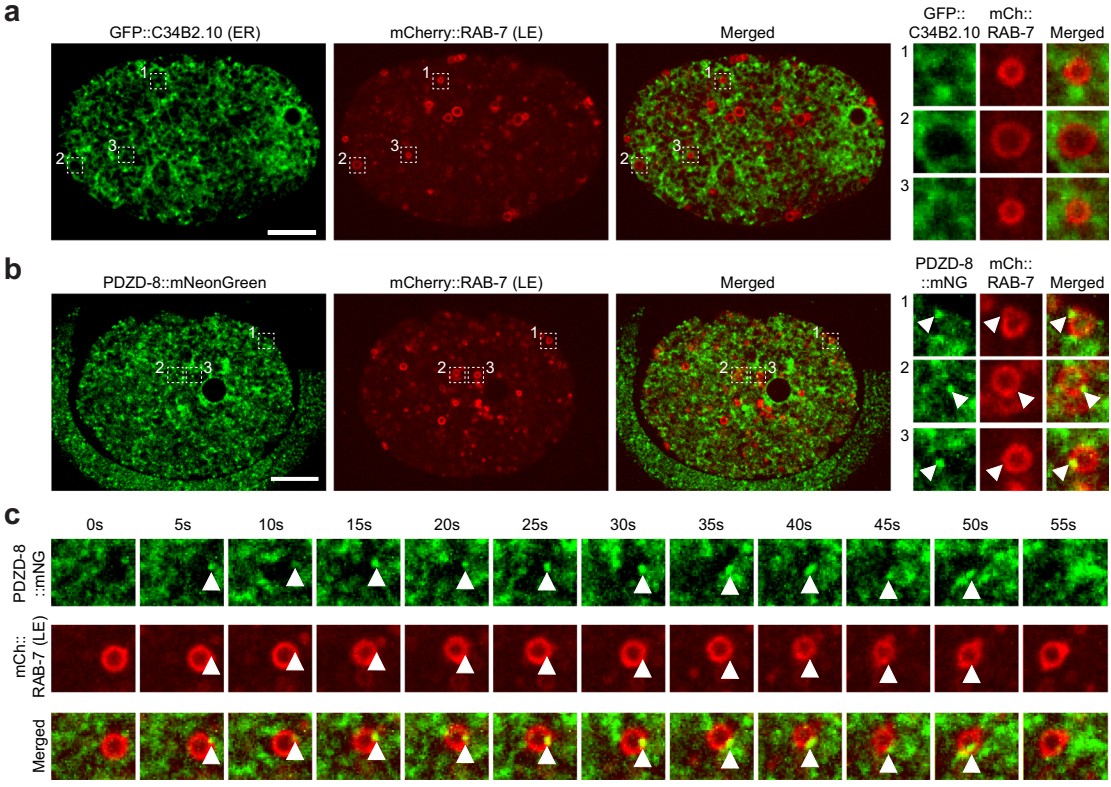

**Fig. 6 PDZD-8 localizes to ER-late endosome contacts. a** Left: A representative live spinning disc confocal (SDC) image of an equatorial plane of an early embryo from wild-type animals, co-expressing endoplasmic reticulum (ER) marker (GFP::C34B2.10) and late endosome (LE) marker, endogenous RAB-7-tagged with mCherry (mCherry::RAB-7). The image is from the polarity establishment phase. Right: Magnified insets, showing Rab-7 positive LEs wrapped around by the ER, that are outlined by indicated white dotted boxes. Scale bar, 10 μm. **b** Left: A representative live SDC image of an equatorial plane of an early embryo from wild-type animals, co-expressing endogenous PDZD-8-tagged with mNeonGreen (PDZD-8::mNeonGreen) and LE marker (mCherry::RAB-7). The image is from the polarity establishment phase. Right: Magnified insets, showing distinct PDZD-8::mNeonGreen puncta present on Rab-7 positive LEs, that are outlined by indicated white dotted boxes. White arrowheads indicate PDZD-8::mNeonGreen puncta that are present on LEs. Scale bar, 10 μm. **c** Time-lapse SDC images of a representative Rab-7 positive LE from an early embryo, co-expressing PDZD-8::mNeonGreen and mCherry::RAB-7 (both at endogenous expression levels), over the period of 55 s. White arrowheads indicate PDZD-8::mNeonGreen puncta that is transiently associated with a LE. Scale bar, 10 μm.

and mixed them with an NBD-labeled PH domain from the FAPP protein [NBD-PH$_{FAPP}$, which selectively binds various PIPs[50,96,97]]. We then monitored the transport of PIP from donor to acceptor liposomes (Fig. 7a) in the presence or absence of SMP$_{PDZD-8}$. On donor liposomes, the fluorescence of NBD-PH$_{FAPP}$ bound to PIP is quenched due to FRET with Rhodamine-PE. Upon transport of PIP from donor to acceptor liposomes, NBD-PH$_{FAPP}$ would bind acceptor liposomes via its interaction with transported PIP, resulting in dequenching and an increase in fluorescence signals[50,96,97] (Fig. 7a and Supplementary Fig. 7a).

Addition of the purified SMP$_{PDZD-8}$ to the mixture of donor liposomes containing PI(4,5)P$_2$ and acceptor liposomes resulted in the dequenching of NBD signals over time in a concentration-dependent manner, revealing that SMP$_{PDZD-8}$ can transport PI(4,5)P$_2$ between liposomes (Fig. 7b–d and Supplementary Fig. 7b). SMP$_{PDZD-8}$ also transported PI(3)P, PI(4)P, and PI(3,5)P$_2$ with similar transport efficiency to PI(4,5)P$_2$, while it transported PI(3,4,5)P$_3$ less efficiently (Fig. 7d and Supplementary Fig. 7c). Using a similar FRET-based approach, we also examined whether SMP$_{PDZD-8}$ could transport phosphatidylserine (PS) between membranes. In this assay, donor liposomes [DOPC (94%), PS (4%), and Rhodamine-PE (2%)] and acceptor liposomes [DOPC (100%)] were prepared and mixed with an NBD-labeled C2 domain from Lactadherin [NBD-C2$_{Lact}$, which selectively binds PS[98]] (instead of an NBD-PH$_{FAPP}$ used for PIP

transport assays), and transport of PS from donor to acceptor liposomes was monitored. The addition of the purified SMP$_{PDZD-8}$ to the mixture of donor and acceptor liposomes resulted in the dequenching of NBD signals in a concentration-dependent manner over time, similar to PI(4,5)P$_2$ (Supplementary Fig. 7d). The transport efficiency of PS was comparable to that of PI(4,5)P$_2$ (Supplementary Fig. 7d). Thus, SMP$_{PDZD-8}$ is capable of transporting various PIPs and other anionic phospholipids, including PS, between liposomes.

We also performed lipid transport assays using various NBD-labeled lipids. In these assays, we used donor liposomes [DOPC (94%), NBD-labeled lipid (either one of the following lipids: NBD-PE, NBD-PA, or NBD-ceramide) (4%), and Rhodamine-PE (2%)] and acceptor liposomes [DOPC (100%)] and monitored dequenching of NBD fluorescence of NBD-labeled lipids over time. We could not detect significant dequenching for NBD-PE and NBD-ceramide with the addition of SMP$_{PDZD-8}$ (Supplementary Fig. 7e, f). In the case of NBD-PA, dequenching was observed overtime and accelerated in the presence of SMP$_{PDZD-8}$ in a concentration-dependent manner (Supplementary Fig. 7g). These results suggest that SMP$_{PDZD-8}$ is not capable of transporting NBD-PE and NBD-ceramide between liposomes but is capable of transporting NBD-PA (Supplementary Fig. 7h). Further, we tested whether SMP$_{PDZD-8}$ could transport cholesterol, using FRET-based in vitro lipid transport assay[99]. For this

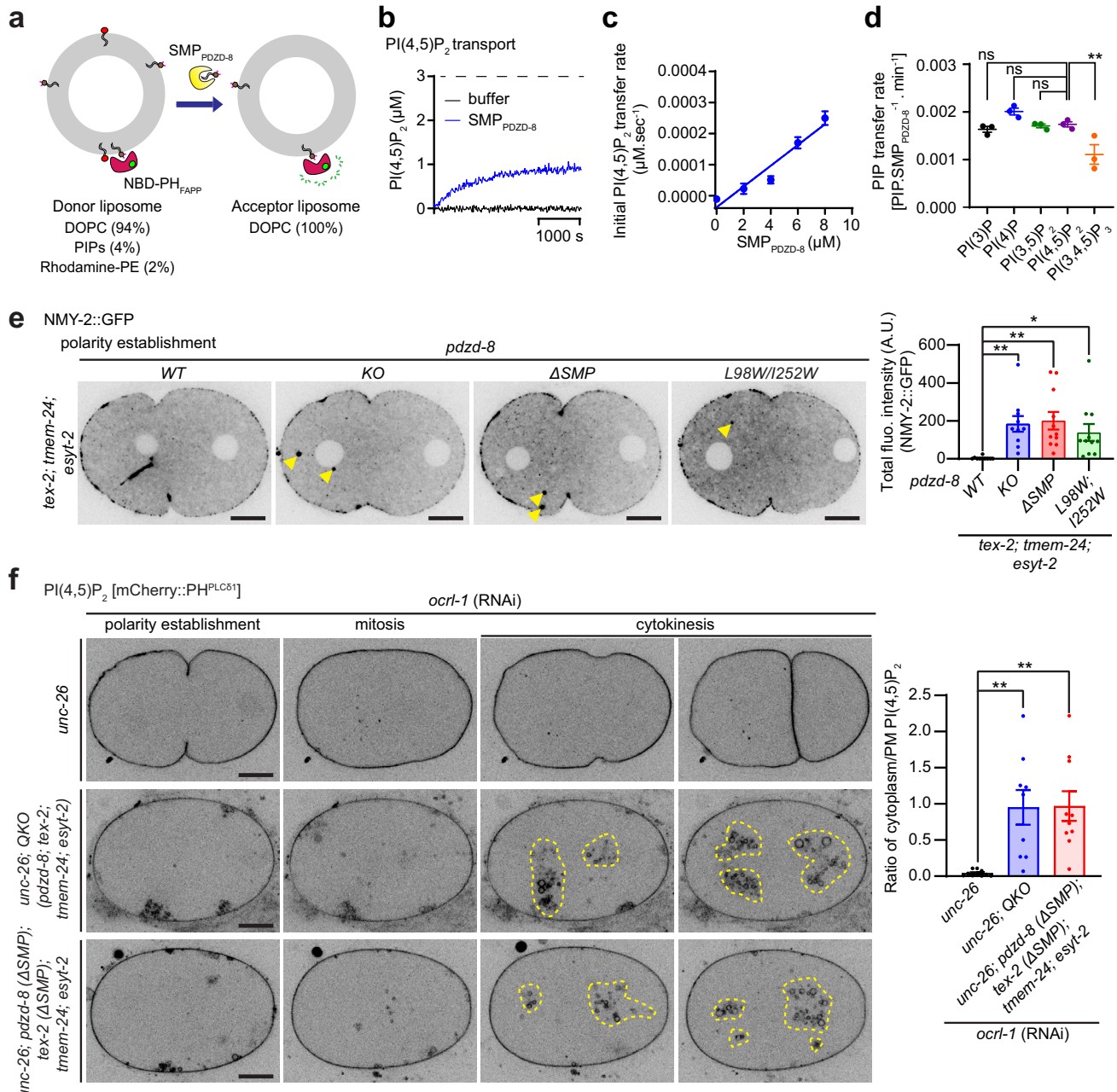

assay, we used donor liposomes [10% dehydroergosterol (DHE), 90% DOPC] and acceptor liposomes [2.5% Dansyl-PE (DNS-PE), 97.5% DOPC] and monitored FRET between transported DHE and DNS-PE on acceptor liposomes. The addition of $SMP_{PDZD-8}$ did not result in DHE transport between liposomes (Supplementary Fig. 7i).

Finally, we tested whether the presence of anionic phospholipids in donor membranes would affect the efficiency of PI(4,5)$P_2$ transport mediated by $SMP_{PDZD-8}$. The addition of either one of PS, PI, or PA, in donor liposomes did not reduce the efficiency of PI(4,5)$P_2$ transport between liposomes, suggesting that the presence of other anionic lipids in donor membranes does not interfere with the property of $SMP_{PDZD-8}$ to transport PI(4,5)$P_2$ (Supplementary Fig. 7j, k).

Taken together, these results suggest that $SMP_{PDZD-8}$ transports various PIPs, including PI(4,5)$P_2$, and other phospholipids, such as PS and possibly PA, but not cholesterol and some lipids.

**The SMP domain is critical for in vivo functions of PDZD-8 and TEX-2.** To further investigate the importance of the SMP domain for PDZD-8 function in vivo, we generated a mutant strain that carries *pdzd-8* lacking the SMP domain [*pdzd-8* (ΔSMP)] at the endogenous locus via CRISPR/Cas9-based gene editing (Supplementary Fig. 1a). This newly generated mutant was crossed into mutant animals that lack TEX-2, ESYT-2, and TMEM-24 (*tex-2; esyt-2; tmem-24*). The resulting early embryos were imaged under SDC microscopy to examine the presence of cytoplasmic NMY-2::GFP puncta. In *tex-2; esyt-2; tmem-24* mutants, *pdzd-8* (ΔSMP) induced cytoplasmic puncta of NMY-2::GFP to levels comparable to those seen in QKO animals, confirming the importance of the SMP domain to PDZD-8 function in vivo (Fig. 7e and Supplementary Movie 7). It was shown previously that E-Syt1 transports lipids between membranes via its SMP domain and that such SMP domain-dependent lipid transport is attenuated by changing two key amino acids within the hydrophobic grove of the SMP domain to

**Fig. 7 The SMP domain of PDZD-8 transports various PIPs, including PI(4,5)P$_2$, between membranes in vitro and plays a critical role for endosomal PI(4,5)P$_2$ homeostasis in vivo. a** Schematic of the PI(4,5)P$_2$ transfer assay in vitro. Donor liposomes [4% phosphoinositides (PIPs), 2% Rhodamine-PE, 94% DOPC] and acceptor liposomes [100% DOPC] (0.6 mM total lipids) were incubated with purified NBD-PH$_{FAPP}$ proteins (0.5 μM) and purified SMP$_{PDZD-8}$ proteins. Dequenching of NBD fluorescence signals that corresponds to the transfer of PI(4,5)P$_2$ from donor to acceptor liposomes were monitored using a fluorometer (see Methods). **b** PI(4,5)P$_2$ transfer from donor to acceptor liposomes by purified SMP$_{PDZD-8}$ proteins (8 μM). The dashed line corresponds to the condition that mimics full PI(4,5)P$_2$ equilibration between donor and acceptor liposomes. **c** The protein concentration dependence of the initial PI(4,5)P$_2$ transport rates of SMP$_{PDZD-8}$. The trend line shows a linear fit of the data points [mean ± SEM, $n = 6$ independent experiments for each condition]. **d** Quantification of PIP transport rates of SMP$_{PDZD-8}$ (8 μM) [mean ± SEM, $n = 3$ independent experiments for each condition; Dunnett's multiple comparisons test, **$p = 0.0056$ (PI(4,5)P$_2$ vs. PI(3,4,5)P$_3$); ns denotes not significant]. **e** Left: Representative live spinning disc confocal (SDC) images of equatorial planes of early embryos from *tex-2; tmem-24; esyt-2* mutants carrying indicated alleles of *pdzd-8*, expressing NMY-2-tagged with GFP (NMY-2::GFP). Images are from polarity establishment phase. Scale bars, 10 μm. Right: Quantification of the total fluorescence intensity of NMY-2::GFP puncta per minute during polarity establishment of the early embryogenesis [mean ± SEM, $n = 10$ embryos (*pdzd-8* WT), $n = 10$ embryos (*pdzd-8* KO), $n = 11$ embryos (*pdzd-8* ΔSMP), $n = 11$ embryos (*pdzd-8* [L98W, I252W]); Dunnett's multiple comparisons test, **$p = 0.0075$ (*pdzd-8* KO), **$p = 0.0027$ (*pdzd-8* ΔSMP), *$p = 0.0474$ (*pdzd-8* [L98W, I252W])]. **f** Left: Representative live SDC images of equatorial planes of early embryos from OCRL-1 RNAi-treated *unc-26* mutants, OCRL-1 RNAi-treated *unc-26; quadruple knock-out [QKO (pdzd-8; tex-2; tmem-24; esyt-2)]* mutants and OCRl-1 RNAi-treated *unc-26; pdzd-8 (ΔSMP); tex-2 (ΔSMP); tmem-24; esyt-2* mutants expressing PI(4,5)P$_2$ biosensor (mCherry::PH$^{PLCδ1}$). Images of each row are from time-lapse movies of the same embryos at different phases as indicated. Yellow dotted lines in mitosis and cytokinesis phases indicate clustering of PI(4,5)P$_2$-positive vesicles. Scale bars, 10 μm. Right: Quantification of the ratio of PI(4,5)P$_2$ levels in cytoplasm (including PI(4,5)P$_2$-positive vesicles) to PI(4,5)P$_2$ levels in the plasma membrane (PM) at the stage prior to cytokinesis, as assessed by mCherry::PH$^{PLCδ1}$, in early embryos from indicated conditions [mean ± SEM, $n = 10$ embryos (*unc-26; ocrl-1* RNAi), $n = 9$ embryos (*unc-26; QKO; ocrl-1* RNAi), $n = 10$ embryos (*unc-26; pdzd-8 (ΔSMP); tex-2 (ΔSMP); tmem-24; esyt-2; ocrl-1* RNAi); Dunnett's multiple comparisons test, **$p = 0.0026$ (*unc-26; QKO; ocrl-1* RNAi), **$p = 0.0017$ (*unc-26; pdzd-8 (ΔSMP); tex-2 (ΔSMP); tmem-24; esyt-2; ocrl-1* RNAi)].

tryptophan[48]. As the structure of the SMP domain of PDZD-8 is not available, the available crystal structure of the E-Syt2 SMP domain[65] was used to model the structure of the SMP domain of PDZD-8. Based on the modeled PDZD-8 structure, we hypothesized that changing leucine-98 and isoleucine-252 of the PDZD-8 SMP domain to tryptophan (W) would block lipid insertion into its hydrophobic groove (Supplementary Fig. 7l). Thus, we generated another mutant strain via CRISPR/Cas9-based gene editing that expresses *pdzd-8* with an SMP domain that carries these double mutations [*pdzd-8* (L98W, I252W)] from the endogenous locus (Supplementary Fig. 1a). Similar to *pdzd-8* (ΔSMP), crossing the *pdzd-8* (L98W, I252W) mutant into *tex-2; esyt-2; tmem-24* mutants induced cytoplasmic puncta of NMY-2::GFP, supporting the importance of SMP-mediated lipid transport to PDZD-8 function (Fig. 7e and Supplementary Movie 7).

We also generated a mutant strain in which *tex-2* lacking the SMP domain [*tex-2* (ΔSMP)] was inserted into the endogenous locus via CRISPR/Cas9-based gene editing (Supplementary Fig. 1a). Crossing the mutant carrying *tex-2* (ΔSMP) into mutant animals that lack PDZD-8, ESYT-2, and TMEM-24 (*pdzd-8; esyt-2; tmem-24*) induced cytoplasmic puncta of NMY-2::GFP similar to those seen in QKO animals, supporting the importance of the TEX-2 SMP domain in TEX-2 function in vivo (Supplementary Fig. 7m). Finally, we generated *unc-26* mutants that lacked all four SMP proteins [*unc-26; QKO*], as well as *unc-26* mutants that carried *pdzd-8* (ΔSMP) and *tex-2* (ΔSMP) with additional knockouts of *tmem-24* and *esyt-2* [*unc-26; pdzd-8* (ΔSMP); *tex-2* (ΔSMP); *tmem-24; esyt-2*]. We then treated these mutants with OCRL-1 RNAi and compared the resulting early embryos under SDC microscopy to assess changes in PI(4,5)P$_2$ distribution by mCherry::PH$^{PLCδ1}$. We observed massive clustering of large PI(4,5)P$_2$-positive vesicles, with increased levels of PI(4,5)P$_2$ in the cytoplasm compared with the PM, in both mutant backgrounds compared to *unc-26* mutant background (Fig. 7f), further supporting the importance of the SMP domains for the functions of PDZD-8 and TEX-2 in vivo.

**Massive accumulation of PI(4,5)P$_2$ on endosomes disrupts their degradative capacity.** In the simultaneous absence of PDZD-8, TEX-2, and the PI(4,5)P$_2$ phosphatases, OCRL-1 and UNC-26,

massive accumulation of PI(4,5)P$_2$ occurs on large abnormal vesicles (Fig. 4a, b). While both PDZD-8 and TEX-2 are present throughout the ER, PDZD-8 additionally localizes to ER-late endosome contacts that are positive for Rab-7 (Fig. 6b, c). Further, our in vitro lipid transport assays revealed that the SMP domain of PDZD-8 is able to transport PI(4,5)P$_2$ between membranes (Fig. 7a–d). Thus, the absence of the SMP proteins and OCRL-1 and UNC-26 might have caused PI(4,5)P$_2$ to accumulate in late endosomes. To examine this possibility, early embryos from OCRL-1 RNAi-treated *unc-26* mutants lacking the four SMP proteins [*unc-26; QKO; ocrl-1 (RNAi)*] were monitored under SDC microscopy to assess the potential accumulation of PI(4,5)P$_2$ on late endosomes. This was accomplished using mCherry::PH$^{PLCδ1}$ and the late endosome marker, GFP::RAB-7 (Supplementary Fig. 8a). Indeed, 19.4 ± 1.2% of large PI(4,5)P$_2$-positive vesicles were transiently decorated by GFP::RAB-7 in mutant embryos ($n = 14$ embryos) (Supplementary Fig. 8a, b). However, a significant fraction of the large PI(4,5)P$_2$-positive vesicles was not decorated by GFP::RAB-7. Thus, we assessed the potential co-localization of large PI(4,5)P$_2$-positive vesicles with two other endosomal markers, namely GFP::RAB-5 for early endosomes and GFP::RAB-11.1 for recycling endosomes (Supplementary Fig. 8a). A significant fraction of the large PI(4,5)P$_2$-positive vesicles was transiently decorated by GFP::RAB-5 [11.1 ± 1.0% ($n = 12$ embryos)] and GFP::RAB-11.1 [39.2 ± 3.4% ($n = 14$ embryos)], suggesting that PI(4,5)P$_2$ accumulates throughout the endosomal system in the absence of PDZD-8, TEX-2, and the PI(4,5)P$_2$ phosphatases (Supplementary Fig. 8a, b). Single vesicle tracking experiments revealed that the association of GFP::RAB-5 to PI(4,5)P$_2$-positive vesicles was generally short-lived, lasting only ~15 s (Supplementary Fig. 8d and Supplementary Movie 9). In contrast, the association of PI(4,5)P$_2$-positive vesicles with GFP::RAB-7 and GFP::RAB-11.1 often lasted more than 55 s until the vesicle left the plane of focus (Supplementary Fig. 8c, e and Supplementary Movies 8, 10).

Massive accumulation of PI(4,5)P$_2$ on endosomes in the simultaneous absence of PDZD-8, TEX-2, and the PI(4,5)P$_2$ phosphatases may have caused disruption of endosomal functions. In *C. elegans*, CAV-1, a homolog of caveolin-1, is rapidly endocytosed from the PM and degraded via clathrin-mediated endocytosis within one cell cycle after fertilization[100]. Using GFP-tagged CAV-1 (CAV-1::GFP) as model cargo for degradation, we assessed potential changes in the degradative capacity of

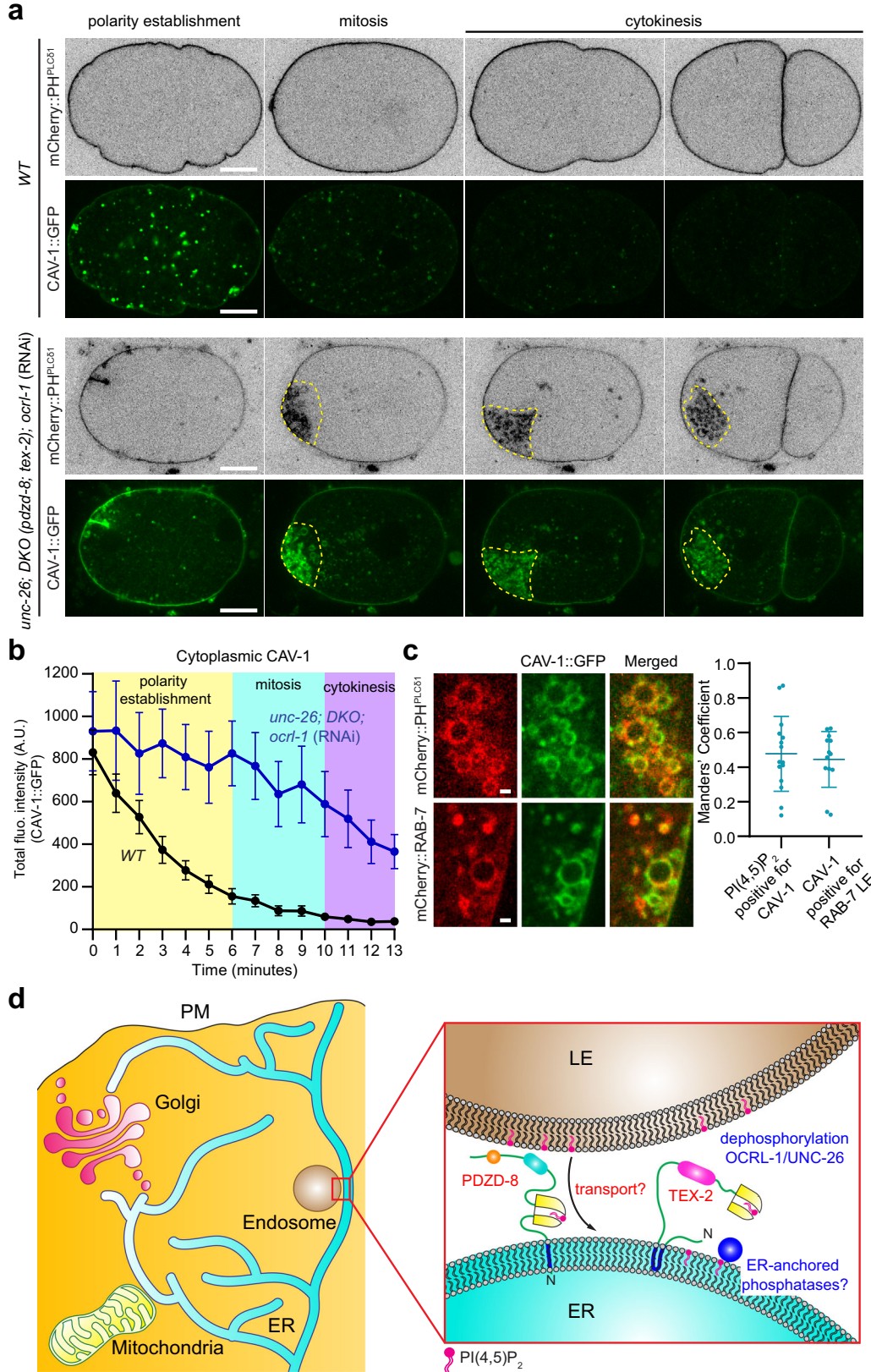

endosomes in early embryos from OCRL-1 RNAi-treated *unc-26* mutants lacking both PDZD-8 and TEX-2 [*unc-26; DKO; ocrl-1 (RNAi)*]. In WT embryos, CAV-1::GFP was rapidly endocytosed from the PM and degraded before completion of cytokinesis as previously reported[100] (Fig. 8a, b and Supplementary Movie 11). In embryos from *unc-26; DKO; ocrl-1 (RNAi)*, however,

degradation of CAV-1::GFP was markedly delayed, and undegraded CAV-1::GFP signals were clearly present in the cytoplasm even after completion of cytokinesis (Fig. 8a, b and Supplementary Movie 11). Strikingly, these cytoplasmic CAV-1::GFP signals extensively overlapped with PI(4,5)P₂-positive vesicles (marked by mCherry::PH^PLCδ1) and late endosomes (marked by

**Fig. 8 Simultaneous depletion of PDZD-8, TEX-2, and PI(4,5)P$_2$ phosphatases disrupts the degradative capacity of endosomes. a** Representative live spinning disc confocal (SDC) images of equatorial planes of early embryos from wild-type control (WT) and OCRL-1 RNAi-treated *unc-26;* double knock-out [DKO *(pdzd-8; tex-2)*] mutants, co-expressing PI(4,5)P$_2$ biosensor (mCherry::PH$^{PLC\delta1}$) and CAV-1-tagged with GFP (CAV-1::GFP). Images of each row are from time-lapse movies of the same embryos at different phases as indicated. Yellow dotted lines in mitosis and cytokinesis phases indicate clustering of PI(4,5)P$_2$-positive vesicles and CAV-1::GFP. Scale bars, 10 μm. **b** Time-course analysis of the total fluorescence intensity of cytoplasmic CAV-1::GFP during polarity establishment (0–6 min), mitosis (6–10 min), and cytokinesis (10–13 min) phases for WT and OCRL-1 RNAi-treated *unc-26;* DKO mutant early embryos as shown in (**a**) [mean ± SEM, $n = 10$ embryos (WT), $n = 9$ embryos (*unc-26;* DKO; *ocrl-1* RNAi)]. **c** Left: Live SDC images of representative PI(4,5)P$_2$-positive vesicles and RAB-7-positive late endosomes (LEs) from OCRL-1 RNAi-treated *unc-26;* DKO mutants, co-expressing CAV-1::GFP and either PI(4,5)P$_2$ biosensor (mCherry::PH$^{PLC\delta1}$) or LE marker (mCherry::RAB-7). Scale bars, 1 μm. Right: Quantification of the association of CAV-1::GFP with either PI(4,5)P$_2$-positive vesicles or RAB-7-positive LEs in OCRL-1 RNAi-treated *unc-26;* DKO mutants as indicated [mean ± SD, $n = 15$ embryos (PI(4,5) P$_2$), $n = 13$ embryos (LE); Manders' coefficient]. **d** Hypothetical model of how PDZD-8 and TEX-2 may maintain the distribution of PI(4,5)P$_2$ in cellular membranes. PI(4,5)P$_2$ is normally enriched in the plasma membrane (PM). PDZD-8 and TEX-2 localize to the endoplasmic reticulum (ER), and PDZD-8 also localizes to ER-LE contacts. PDZD-8 may regulate endosomal PI(4,5)P$_2$ levels by transporting PI(4,5)P$_2$ from late endosomes to the ER via its SMP domain at ER-late endosome contacts, although we do not have direct evidence supporting this vectorial transport in the current study. It is also plausible that PDZD-8 may primarily act as a tethering factor for ER-LE contacts. ER-anchored PI(4,5)P$_2$ phosphatases, such as INPP5K, may participate in dephosphorylating PI(4,5)P$_2$ and prevent PI(4,5)P$_2$ from accumulating on ER membranes. TEX-2 acts redundantly with PDZD-8 to regulate endosomal PI(4,5)P$_2$ levels, whose function also depends on the SMP domain. TEX-2 may transiently populate ER-LE contacts. PI(4,5)P$_2$ phosphatases, including OCRL-1 and UNC-26/synaptojanin, work together with PDZD-8 and TEX-2 to prevent the build-up of PI(4,5)P$_2$ in the endosomal systems.

endogenously tagged mCherry::RAB-7) (Fig. 8a, c and Supplementary Fig. 9a). 47.7 ± 5.6% of the PI(4,5)P$_2$-positive structures were positive for CAV-1::GFP ($n = 15$ embryos), while 30.8 ± 3.1% of late endosomes were positive for CAV-1::GFP ($n = 13$ embryos) (Fig. 8c and Supplementary Fig. 9b). These results demonstrate major disruption of the degradative capacity of the endosomal system in the absence of PDZD-8, TEX-2, and the PI(4,5)P$_2$ phosphatases. We also noted that PM CAV-1::GFP signals were more sustained in early embryos from *unc-26;* DKO; *ocrl-1 (RNAi)* compared to WT (Supplementary Fig. 9c), indicating that endocytosis of CAV-1::GFP was also delayed in these mutant embryos. PM PI(4,5)P$_2$ plays a critical role in clathrin-mediated endocytosis via its property to recruit various endocytic factors to the PM[16,17,19]. Thus, aberrant cytoplasmic accumulation of PI(4,5)P$_2$ in these mutant embryos (Fig. 4f) may have also resulted in defects in clathrin-mediated endocytosis, which is required for CAV-1::GFP internalization[100].

Taken together, our results demonstrate that the simultaneous absence of SMP proteins and PI(4,5)P$_2$ phosphatases lead to massive accumulation of PI(4,5)P$_2$ in endosomes, which disrupted their degradative capacity. This alteration in turn resulted in the ectopic recruitment of the actomyosin machinery to aberrant endosomes, defects in cytokinesis, and the absence of cortical ruffling, which collectively led to massive embryonic lethality in *C. elegans*.

## Discussion

We have demonstrated that endosomal PI(4,5)P$_2$ homeostasis is maintained during early embryogenesis in vivo by the SMP proteins, PDZD-8 and TEX-2. We have also shown that these proteins work together with PI(4,5)P$_2$ phosphatases to prevent the build-up of PI(4,5)P$_2$ on endosomes. Key findings of the current study are the following:

1. Using *C. elegans* early embryos as our in vivo model system, we found that depleting all four SMP proteins led to accumulation of PI(4,5)P$_2$ on endosomes. This PI(4,5)P$_2$ accumulation resulted in ectopic recruitment of anillin and actomyosin (i.e., myosin II and actin), from the PM to endosomes. Further, we examined single, double, triple, and quadruple knockouts of SMP proteins and found that PDZD-8 and TEX-2 played critical roles in preventing PI(4,5)P$_2$ accumulation on endosomes.
2. We found that PDZD-8 and TEX-2 act together with the PI(4,5)P$_2$ phosphatases, OCRL-1, and UNC-26/

synaptojanin, to suppress the build-up of PI(4,5)P$_2$ on endosomes. In their simultaneous absence, PI(4,5)P$_2$ accumulated within aberrantly large endosomal membranes, inducing ectopic recruitment of myosin II to these membranes. This resulted in parallel reductions in actomyosin contraction at the PM, leading to embryonic lethality due to defects in both cleavage furrow ingression and cortical ruffling. These data indicate that it is critically important to limit endosomal PI(4,5)P$_2$ levels for cell viability.
3. Both TEX-2 and PDZD-8 localize to the ER and PDZD-8 also localizes to the site of contacts formed between the ER and late endosomes that are marked by RAB-7. Using an in vitro lipid transfer assay, we found that purified PDZD-8 SMP domains transported various PIPs, including PI(4,5) P$_2$, between membranes. Targeted deletion of the PDZD-8 and TEX-2 SMP domains disrupted PI(4,5)P$_2$ homeostasis similar to their complete absence. Further, introducing mutations into the PDZD-8 SMP domain that are predicted to prevent it from capturing lipids was sufficient to disrupt the cellular function of PDZD-8.
4. We found that PI(4,5)P$_2$ accumulated not only on late endosomes but also on early and recycling endosomes in PDZD-8, TEX-2, OCRL-1, UNC-26 quadruple mutants. This disrupted the degradative capacity of the endosomal system. Our results support a model whereby PDZD-8 regulates endosomal PI(4,5)P$_2$ levels at ER-late endosome contact sites via the lipid-harboring SMP domain to help maintain the distribution of PI(4,5)P$_2$ in cellular membranes. Further, PDZD-8 functioned redundantly with TEX-2, OCRL-1, and UNC-26 (see below for further discussion).

In *C. elegans*, PI(4,5)P$_2$ forms a dynamic cortical structure during early embryogenesis that overlaps with F-actin and is coincident with various actin regulators, such as RHO-1, CDC-42, and RhoGEF/ECT-2[7]. However, how PI(4,5)P$_2$ is restricted to the PM and its physiological relevance to early embryogenesis have both remained unclear. Emerging evidences from various model systems, including yeast, flies, and human cells, support the importance of PI(4,5)P$_2$ enrichment in the PM in the coordinated recruitment of various proteins to the PM for actomyosin contractility and cytokinesis[74]. In humans and yeast, PI(4,5)P$_2$ directly binds key regulators of these processes, including Rho-GEF/ECT2, RhoA (RhoA in humans and Rho-1 in budding

yeast), and anillin (Mid1 in yeast)[5,9,10,92]. Anillin is an evolutionarily conserved scaffold protein that links RhoA, F-actin, and myosin II to the cell cortex for coordinating cell divisions and various other contractile processes, including epithelial mechanics and cell-cell adherens junctions[11,90,91,101–107]. A recent study in human cells suggested that anillin promotes cell contractility by inhibiting the dissociation of active GTP-bound RhoA from the PM[8]. On the other hand, pulsed activation and inactivation of RhoA was reported to precede PM recruitment of anillin during cortical ruffling (i.e., pulsed actomyosin contractility near the PM) in *C. elegans* early embryos[6,82]. Regardless of the sequence by which anillin and RhoA are recruited to the PM, the actomyosin contractile network that is regulated by these proteins is essential for both cortical contractility and subsequent furrow formation[5,6,82,103,104,108–113]. We found that the simultaneous depletion of PDZD-8 and TEX-2 resulted in aberrant accumulation of $PI(4,5)P_2$ on a subset of endosomes that was accompanied by the ectopic presence of anillin, myosin II, and actin. Additional depletion of the $PI(4,5)P_2$ phosphatases, OCRL-1, and UNC-26, induced the aberrant appearance of large $PI(4,5)P_2$-positive endosomes that further sequestered myosin II and anillin, leading to severe defects in cytokinesis and embryonic lethality. In this context, cortical ruffling was halted, reflecting general defects in actomyosin contractility at the cell cortex. Thus, our results are consistent with the critical importance of $PI(4,5)P_2$ in the PM for anchoring the actomyosin contractile network to the cell cortex. When levels of endosomal $PI(4,5)P_2$ rose above a certain threshold, regulators of the actomyosin contractility, including anillin, myosin II, and actin, were ectopically recruited to endosomes, causing insufficient cortical contractility and failure to form a furrow. Although growing evidence suggests a role for endosomal $PI(4,5)P_2$ in cell physiology[4], our results demonstrate that highly redundant functions of $PI(4,5)P_2$ phosphatases and SMP proteins keep levels of endosomal $PI(4,5)P_2$ below a certain threshold to prevent failure of cytokinesis and animal development in vivo.

We found that PDZD-8 and TEX-2 require their SMP domains to function properly to control endosomal $PI(4,5)P_2$ levels. Targeted deletion of the SMP domains from these two proteins was sufficient to cause the accumulation of $PI(4,5)P_2$ in endosomes. TEX-2 and PDZD-8 are both localized to the ER, whereas PDZD-8 also localized to contact sites between the ER and late endosomes. Further, we found that the SMP domain of PDZD-8 was able to extract and transport various PIPs, including $PI(4,5)P_2$, between membranes in vitro. Based on these results, we suggest that PDZD-8 may act at ER-late endosome contacts to extract $PI(4,5)P_2$ from late endosomes via its SMP domain, while a possibility that PDZD-8 may primarily act as a tethering protein for ER-late endosome contacts cannot be excluded (Fig. 8d). There are two possible fates for $PI(4,5)P_2$ that is extracted from endosomes. First, $PI(4,5)P_2$ could be transported to the ER for its dephosphorylation at the ER. Second, $PI(4,5)P_2$ could be presented to OCRL-1 and UNC-26 for its rapid dephosphorylation on endosomes. As the phenotype resulting from the combined depletion of PDZD-8, TEX-2, OCRL-1, and UNC-26 was much stronger than the phenotype resulting from the depletion of OCRL-1 and UNC-26 alone, we favor the first scenario in which other $PI(4,5)P_2$ phosphatases act on the ER "in cis" to remove $PI(4,5)P_2$ that is transported from endosomes, although we do not have direct evidence supporting a vectorial transfer of $PI(4,5)P_2$ from endosomes to the ER by PDZD-8/TEX-2 in the current study. One potential candidate for an ER-anchored $PI(4,5)P_2$ phosphatase is INPP5K[114]. Indeed, ER-endosome contacts have been implicated in the homeostatic regulation of another phosphoinositide, phosphatidylinositol 4-phosphate (PI4P), via concerted actions of non-vesicular PI4P transport from endosomes to

the ER by an LTP [oxysterol-binding protein (OSBP)] and dephosphorylation of PI4P "in cis" by an ER-anchored PI4P phosphatase, Sac1[115]. Whether INPP5K and/or any other $PI(4,5)P_2$ phosphatases are involved in endosomal $PI(4,5)P_2$ homeostasis requires further investigation. Further studies are also needed to better understand the cross-talk between PDZD-8 and TEX-2. While we could not detect a particular association of TEX-2 at ER-endosome contacts, TEX-2 may transiently populate ER-endosome contacts to help maintain endosomal $PI(4,5)P_2$ homeostasis. As the SMP domain of PDZD-8 is able to transport various lipids between membranes in vitro, PDZD-8 may also regulate the distribution of other lipids in addition to $PI(4,5)P_2$.

In the absence of PDZD-8, TEX-2, OCRL-1, and UNC-26, levels of $PI(4,5)P_2$ were dramatically elevated on endosomes that were positive for early, recycling, and late endosome markers. Massive accumulation of $PI(4,5)P_2$ on endosomes disrupted their degradative capacity as evidenced by the failure of the degradation of CAV-1 in early embryos. Our results suggest that $PI(4,5)P_2$ on these endosomes originated from the PM based on (1) the absence of ectopic accumulation of PPK-1, a sole homolog of PIP5K in *C. elegans*, on $PI(4,5)P_2$-positive endosomes in mutants lacking SMP proteins; (2) the suppression of cytoplasmic $PI(4,5)P_2$ accumulation in early embryos from *unc-26*; DKO mutants treated with OCRL-1 RNAi by blocking endocytosis with RNAi against DYN-1, a sole homolog of dynamin in *C. elegans*. In mammals, ORP2, an LTP that belongs to the OSBP family, has been implicated in regulating the levels of $PI(4,5)P_2$ in various cellular compartments, including the PM[50] and recycling endosomes[116]. Thus, potential dysregulation of other LTPs involved in $PI(4,5)P_2$ regulation may have also contributed to aberrant $PI(4,5)P_2$ accumulation in the absence of SMP proteins.

In mammalian cells, PDZD8 (the homolog of *C. elegans* PDZD-8) localizes to various membrane contact sites, including ER-mitochondria and ER-endosome contact sites[69–72]. PDZD8 interacts with Protrudin and localizes to ER-late endosome contacts via its interaction with Rab7[69,71,72]. Based on its localization, it was proposed that PDZD8 may play a role in endosomal maturation[71,72]. The SMP domain of PDZD8 was reported to extract and transport various NBD-labeled lipids in vitro[72,117]. Further, Shirane et al. showed that co-expression of PDZD8 and Protrudin induces abnormally large vacuoles, proposing a model whereby PDZD8 transports lipids from the ER to late endosomes[72]. Our current study shows that the SMP domain of PDZD-8 transports PIPs, including $PI(4,5)P_2$, in vitro. Furthermore, we show that PDZD-8 function together with TEX-2 and $PI(4,5)P_2$ phosphatases to play a critical role in maintaining endosomal $PI(4,5)P_2$ homeostasis via its SMP domain in *C. elegans* early embryos, providing insights into the physiological function of this evolutionarily conserved protein at ER-late endosome contacts.

Finally, mutations in OCRL1 and synaptojanin (homologs of *C. elegans* OCRL-1 and UNC-26, respectively) have both been linked to human disorders, supporting the critical importance of cellular $PI(4,5)P_2$ homeostasis in normal cell physiology[33–39,118]. In *Drosophila* S2 cells, removal of dOCRL (the homolog of OCRL1 in *Drosophila*) results in ectopic accumulation of $PI(4,5)P_2$ at enlarged endosomal compartments. This leads to the aberrant recruitment of the cytokinetic machinery and severe defects in cleavage furrow formation and cytokinesis[20], similar to what we observed in mutants simultaneously lacking PDZD-8, TEX-2, UNC-26, and OCRL-1 in *C. elegans* early embryos. However, cytokinesis defects are less pronounced in human cells lacking OCRL1[24,119]. Likewise, mice lacking synaptojanin 1 are viable at birth[29]. Thus, other proteins likely function redundantly to control endosomal $PI(4,5)P_2$ homeostasis in human cells. Our results are consistent with this notion, providing supporting evidence that OCRL1 and synaptojanin may act redundantly with the SMP proteins, PDZD8 and

TEX2, to prevent build-up of PI(4,5)P$_2$ in endosomes. Future studies are needed to examine the role of mammalian PDZD8 and TEX2 in endosomal PI(4,5)P$_2$ homeostasis. Modulating their functions may help alleviate some disease conditions caused by mutations in OCRL1 or synaptojanin.

## Methods

**C. elegans strains.** All strains were grown on nematode growth medium (NGM) plates that were seeded with *E. coli* strain OP50 [as described in ref. [120]] and maintained at room temperature unless stated otherwise. Wild-type worms were Bristol variety N2. Some strains were provided by the Caenorhabditis Genetics Center and the National Bioresource Project. They were *ltIs44[pie-1p::mCherry::PH(PLC1delta1) + unc-119(+)]*, *zuIs45[nmy-2::NMY-2::GFP + unc-119(+)]*, *zbIs2[pie-1::lifeACT::RFP]*, *pwIs20[pie-1p::GFP::rab-11.1 + unc-119(+)]*, *ojIs35[pie-1::GFP::rab-11.1 + unc-119(+)]*, *pwIs20[pie-1p::GFP::rab-5 + unc-119(+)]*, *weIs15[pie-1p::GFP::eea-1(FYVEx2) + unc-119(+)]*, *ocfIs2[pie-1p:mCherry::sp12::pie-1 3'UTR + unc-119(+)]*, *ojIs23 [pie-1p::GFP::C34B2.10]*, *pwIs28[pie-1p::cav-1::GFP(7) + unc-119(+)]*, *unc-26(s1710)* and *R11G1.6(tm10626)*. Details of *C. elegans* strains used in this study are listed in Supplementary Table 1.

### Genome editing

*Deletion.* Specific deletion alleles of C53B4.4/*pdzd-8* [*pdzd-8(syb664)* and *pdzd-8(syb2977)*], F55C12.5/*tex-2* [*tex-2(syb670)* and *tex-2(syb2190)*], and *esyt-2(syb709)* were generated by SunyBiotech Corporation using CRISPR/Cas9 genome editing method. For null deletion alleles, 1864 bp (315 bp to 2178 bp) was deleted from *pdzd-8* to generate *pdzd-8(syb664)*, 1960 bp (3877 bp to 5836 bp) was deleted from *tex-2* to generate *tex-2(syb670)*, and 1902 bp (627 bp to 2528 bp) was deleted from *esyt-2* to generate *esyt-2(syb709)*. For SMP domain-specific deletion alleles, 1237 bp (317 bp to 1553 bp) was deleted from *pdzd-8* to generate *pdzd-8(syb2977)*, and 1082 bp (4784 bp to 5865 bp) was deleted from *tex-2* to generate *tex-2(syb2190)*. "A" of the initiation codon (ATG) of each gene is defined as 1 bp.

*Knock-in.* A knock-in allele of *tex-2(yas46[F55C12.5::EGFP^3xFLAG])* was generated using the CRISPR/Cas9 genome editing method[121]. A CRISPR/Cas9 target site around 3' end of *tex-2* gene was selected using the Cas9 guide RNA design tool (http://crispr.mit.edu)[122]. The gRNA used was: TATTGTATCCCATGTGGAG-TAGG. The Cas9-sgRNA construct, pDD162_F55C12.5_gRNA (DJ53), together with co-injection markers and the plasmid pDD282_F55C12.5_GFP (DJ44), containing the donor template [GFP and self-excision cassette (SEC) flanked by *tex-2* homology arms], were microinjected into N2 worms. The SEC was removed by heat shock, and SAH235 carrying TEX-2 that is tagged with EGFP at its C-terminus was generated. Worms expressing wrmScarlet tagged to the N-terminus of ANI-1 at endogenous expression levels, *ani-1(syb1710[wrmScarlet::ANI-1])*, were generated by SunyBiotech Corporation using the CRISPR/Cas9 genome editing method. cDNA encoding wrmScarlet was inserted right after the initiation codon (ATG) of *ani-1*. Worms expressing mNeonGreen tagged to the C-terminus of C53B4.4/PDZD-8 at endogenous expression levels, *C53B4.4 (syb4099[C53B4.4::mNeongreen])*, were generated by SunyBiotech Corporation using the CRISPR/Cas9 genome editing method. cDNA encoding mNeonGreen was inserted in-frame in the last common exon of C53B4.4/*pdzd-8*. Worms expressing EGFP tagged to the C-terminus of PPK-1 at endogenous expression levels, *ppk-1(syb4109[PPK-1::EGFP])*, were generated by SunyBiotech Corporation using the CRISPR/Cas9 genome editing method. cDNA encoding EGFP was inserted right after the stop codon (TGA) of *ppk-1*. Worms expressing mCherry tagged to the N-terminus of RAB-7 at endogenous expression levels, *rab-7(syb4141[mCherry::RAB-7])*, were generated by SunyBiotech Corporation using the CRISPR/Cas9 genome editing method. cDNA encoding mCherry was inserted right after the initiation codon (ATG) of *rab-7*. CTT at 493 bp to 495 bp (Leucine) and ATC at 1107 bp to 1109 bp (Isoleucine) of C53B4.4/*pdzd-8* were sequentially replaced with TGG (Tryptophan) using CRISPR/Cas9 genome editing method to generate *C53B4.4(syb3276)* by SunyBiotech Corporation.

*Single copy transgene integration by MosSCI.* Worms carrying the following single copy transgene *rab-7(sybIs2380[pie-1P::GFP::rab-7::pie-1UTR)* was generated via MosSCI method[123]. For the generation of the MosSCI line, pCFJ352_GFP_RAB-7 (YS16) was constructed and microinjected into the EG6701 strain to integrate respective transgenes to chromosome I (position: -5.32) by SunyBiotech Corporation.

**Molecular cloning.** Details of oligos and primers used in this study are listed in Supplementary Table 1.

*For C. elegans experiments*

*pDD162_F55C12.5_gRNA (DJ53).* The Cas9 containing vector, pDD162 (Addgene #47549)[124], was mutated to insert CRISPR gRNA sequence via site-directed mutagenesis using the primer set, F55C12.5_GUIDE-5_FWD and CAS9_MUT_REV, to generate pDD162_F55C12.5_gRNA.

*pDD282_F55C12.5_GFP (DJ44).* Homology arms were amplified by PCR using *C. elegans* N2 genomic DNA as a template and two primer sets, F55C12.5_C_282_5' FW and F55C12.5_C_5'_REV, and F55C12.5_C_3'_FWD and F55C12.5_C_3'_REV. The resulting PCR products were inserted into the donor template plasmid, pDD282 (Addgene #66823)[121], to flank EGFP-SEC-FLAG3X via Gibson assembly strategy to generate pDD282_F55C12.5_GFP.

*pCFJ352_GFP_RAB-7 (YS16).* *pie-1* promoter, GFP::RAB-7 (N-terminal GFP tag) cDNA, and *pie-1* 3' untranslated region (UTR) were obtained and subcloned into pCFJ352, (Addgene #30539), by SunyBiotech Corporation to generate pCFJ352_GFP_RAB-7. MluI, AvrII, and BssHII sites were introduced between the *pie-1* promoter and GFP cDNA, between the GFP cDNA and the *rab-7* cDNA, and between the *rab-7* cDNA and *pie-1* 3' UTR, respectively.

*For expression in mammalian cells*

*pCePDZD8-EGFP (TN216).* cDNAs of *C. elegans* PDZD-8 (C53B4.4b.1) was amplified by PCR using the following primer set, 5'_HINDIII_CEPDZD8_NS and 3'_KPNI_CEPDZD8_CAS. The PCR products were then ligated at HindIII and KpnI sites in pEGFP-N1 to generate pCePDZD8-EGFP.

*pCeTEX-2-EGFP (TN215).* cDNA of *C. elegans* TEX-2 (F55C12.5b.1) was amplified by PCR using the following primer set, 5'_ECORI_CETEX2_NS and 3'_KPNI_CETEX2_CAS. The PCR products were then ligated at EcoRI and KpnI sites in pEGFP-N1 to generate pCeTEX-2-EGFP.

*For recombinant protein purification*

*His-SMP$_{PDZD-8}$.* cDNA corresponding to the SMP domain (residues 60-266) of *C. elegans* PDZD-8 protein was PCR amplified using the C53B4.4/*pdzd-8* as a template and the primer set 5'NCOI_PDZD8_VC02 and 3'BAMHI_PDZD8_VC03. The PCR products were then ligated at NcoI and BamHI sites in pNIC28-Bsa4 to generate pNIC28-Bsa4 His-SMP-PDZD-8 vc03.

*PH$_{FAPP}$ (T13C/C37S/C94S).* gBlocks (IDT) containing the PH domain of FAPP with the respective mutations (T13C/C37S/C94S) (PH-FAPP) was synthesized and amplified by PCR, using the primer sets 5'NCOI_PH-FAPP and 3'BAMHI_PH-FAPP. The PCR products were then ligated at NcoI and BamHI sites in pNIC28-Bsa4 to generate pNIC28-Bsa4 PH-FAPP (T13C/C37S/C94S).

*C2$_{Lact}$ (C270A/C427A/H352C).* gBlocks (IDT) containing the C2 domain of Lactadherin with the respective mutations (C270A/C427A/H352C) (C2$_{Lact}$) was synthesized and amplified by PCR, using the primer sets 5'NcoI_Lact-C2_w_mutations and 3'BamHI_Lact-C2_w_mutations. The PCR products were then ligated at NcoI and BamHI sites in pNIC28-Bsa4 to generate pNIC28-Bsa4 Lact-C2 (C270A/C427A/H352C).

**Cell culture and transfection.** COS-7 cells were cultured in Dulbecco's modified Eagle's medium (DMEM) containing 20% fetal bovine serum (FBS) and 1% penicillin/streptomycin at 37 °C and 5% CO$_2$. Transfection of plasmids was carried out with Lipofectamine 2000 (ThermoFisher Scientific) 1 day before the imaging. Cells were routinely verified as free of mycoplasma contamination at least every 2 months, using MycoGuard Mycoplasma PCR Detection Kit (Genecopoeia). No cell lines used in this study were found in the database of commonly misidentified cell lines that is maintained by ICLAC and NCBI Biosample.

**RNAi treatment.** RNAi experiments were performed by the feeding method as previously described[125]. In brief, *E. coli* strain HT115 transformed with a feeding RNAi vector were grown at 37 °C for 5 h in LB media that contained 100 μg/ml Ampicillin and then seeded onto NGM plates that were supplemented with 1 mM IPTG and 25 μg/ml Carbenicillin. The bacterial lawn was allowed to grow overnight at room temperature. L4 worms were passed onto these plates and further grown, at 25 °C for 24 h, to gravid adult worms, and the resulting progeny of these worms were assessed. For Supplementary Fig. 5a, double RNAi for control; *ocrl-1* and *ocrl-1; dyn-1* were performed by mixing bacteria expressing L4440 vector control and *ocrl-1* RNAi clone as well as *ocrl-1* RNAi clone and *dyn-1* RNAi clone in a 1:1 ratio. RNAi clones used are listed in Supplementary Table 1.

**Staining of embryos.** For staining *C. elegans* embryos with fluorescent dyes (Supplementary Fig. 2a), the germline of live worms were stained as previously described[126]. In brief, either LysoTracker™ Red DND-99 (ThermoFisher Scientific #L7528) or MitoTracker™ Red CMXRos (ThermoFisher Scientific #M7512) was added to both OP50 culture and NGM plate to achieve a final concentration of 2 μM of the fluorescent dye. L4 worms were passed onto these plates and further grown, at 20 °C for 24 h, without exposure to light. After staining, gravid adult worms were dissected for live-cell imaging of the embryos.

**Microscopy**. Gravid adult worms were picked to a coverslip, suspended in standard egg buffer (118 mM NaCl, 40 mM KCl, 3.4 mM CaCl₂, 3.4 mM MgCl₂, 5 mM HEPES, pH 7.4), containing uniformly sized polystyrene beads (15.6 ± 0.03 μm diameter, Bangs labs, #NT29N), and dissected along the mid-region of the gonads. Coverslip was then placed on a 3% agarose pad for live-cell imaging.

SDC microscopy (Figs. 1b, 2a–c, 3a, 4a, e, g, 5a, 6a–c, 7e, f, 8a, c and Supplementary Figs. 2a, c, 3a, 4a, b, 5a, 6a, b, e, 7m, 8a, c–e, 9a) and super-resolution SDC-structured illumination microscopy (SDC-SIM) (Fig. 2c–g, and Supplementary Figs. 2b, 6c, d), were performed on a setup built around a Nikon Ti2 inverted microscope equipped with a Yokogawa CSU-W1 confocal spinning head, a Plan-Apo objective (100 × 1.45 N.A.), a back-illuminated sCMOS camera (Prime 95B; Photometrics), and a super-resolution module (Live-SR; Gataca Systems) that was based on structured illumination with optical reassignment and image processing[127]. The method, known as multifocal structured illumination microscopy[128], makes it possible to double the resolution and the optical sectioning capability of confocal microscopy simultaneously. The maximum resolution is 128 nm with a pixel size in super-resolution mode of 64 nm. Excitation light was provided by 488 nm/150 mW (Coherent) (for GFP/mNeonGreen) and 561 nm/100 mW (Coherent) (for mCherry/RFP/wrmScarlet) (power measured at optical fiber end) DPSS laser combiner (iLAS system; Gataca systems). All image acquisition and processing was controlled by MetaMorph (Molecular Device) software. Images were acquired using exposure times in the 500 msec range. For time-lapse imaging, images were acquired at 0.2 Hz (every 5 s) for up to 301 timepoints.

**Viability assay**. Viability assay (Figs. 5c, d and Supplementary Figs. 3d, e, 5c, d) was performed to determine the number of eggs laid and percentage of eggs that hatched to produce L1 larvae. L4 worms were individually plated and treated with RNAi as described above at 25 °C for 48 h (instead of 24 h). The number of hatched and unhatched eggs from each plate was counted to measure brood size and viability.

**Molecular modeling**. The modeled structure of the SMP domain of PDZD-8 (Supplementary Fig. 7l) was obtained by submitting the primary sequence (residues 38–340) of PDZD-8 (coding cDNA, C53B4.4b.1) to the I-TASSER server[129], using the SMP domain of E-Syt2 structure (PDB: 4P42)[65] as a template. The modeled structure of the SMP domain of PDZD-8 with mutated residues (L98W, I252W) were generated using the "Mutagenesis" function of the PyMOL.

**Purification of SMP_PDZD-8, PH_FAPP (T13C/C37S/C94S), and C2_Lact (C270A/C427A/H352C)**. All proteins were overexpressed in E. coli BL21-DE3 Rosetta cells. A 750 mL culture was grown until OD₆₀₀ ~0.5–0.7 with appropriate antibiotics. About 0.1 mM IPTG (ThermoFisher Scientific) was then added, and the culture was further grown at 18 °C for 18 h to allow protein expression. Cells were harvested by centrifugation at 4700 × g, at 4 °C for 15 min, and resuspended in 30 ml of lysis buffer [100 mM HEPES, 500 mM NaCl, 10 mM imidazole, 10% glycerol, 0.5 mM TCEP (pH 7.5)], supplemented with protease inhibitors (Complete, EDTA-free; Roche) together with the cocktail of 100 μg/ml lysozyme [at least 30 min incubation for C2_Lact (C270A/C427A/H352C)] (Sigma-Aldrich/Merck) and 50 μg/ml DNAse I (Sigma-Aldrich/Merck). Cells were lysed with sonication on ice in a Vibra Cell (Sonics and Materials, Inc) (70% power, 3 s pulse on, 3 s pulse off for 3 min for three to five rounds). The lysate was clarified by centrifugation at 47,000 × g, at 4 °C for 20 min. The supernatants were incubated at 4 °C for 30 min with Ni-NTA resin (ThermoFisher Scientific), which had been equilibrated with 2.5 ml of wash buffer 1 (20 mM HEPES, 500 mM NaCl, 10 mM Imidazole, 10% glycerol, 0.5 mM TCEP, pH 7.5). The protein-resin mixtures were then loaded onto a column to be allowed to drain by gravity. The column was washed with 10 ml of wash buffer 1 once and 10 ml of wash buffer 2 (20 mM HEPES, 500 mM NaCl, 25 mM imidazole, 10% glycerol, 0.5 mM TCEP, pH 7.5) once, and then eluted with 1.25 ml of elution buffer 1 (20 mM HEPES, 500 mM NaCl, 500 mM imidazole, 10% glycerol, 0.5 mM TCEP, pH 7.5). The proteins were then concentrated using Vivaspin 20 MWCO 10 kDa (GE Healthcare) and further purified by gel filtration (Superdex 200 increase 10/300 GL, GE Healthcare) with elution buffer 2 (20 mM HEPES, 300 mM NaCl, 10% glycerol, 0.5 mM TCEP, pH 7.5), using the AKTA Pure system (GE Healthcare). Relevant peaks were pooled, and the protein sample was concentrated.

**Tagging of PH_FAPP (T13C/C37S/C94S) and C2_Lact (C270A/C427A/H352C) with NBD**. Purified PH_FAPP (T13C/C37S/C94S) proteins or purified C2_Lact (C270A/C427A/H352C) proteins were mixed with a tenfold excess of N,N′-dimethyl-N-(iodoacetyl)-N′-(7-nitrobenz-2-oxa-1,3-diazol-4-yl)ethylenediamine (IANBD-amide, Molecular Probes) after removal of TCEP by gel filtration on Superdex 75 increase 10/300 GL (GE Healthcare). After ~14 h incubation by rotating the mixture in a cold room, the reaction was stopped by adding a tenfold excess of L-cysteine in the mixture. The free IANBD-amide and excess L-cysteine were removed from the mixture by gel filtration on Superdex 75 increase 10/300 GL (GE Healthcare). The success of NBD labeling was checked by SDS–PAGE.

**Liposome-based experiments**. Lipids in chloroform were dried under a stream of N₂ gas, followed by further drying in the vacuum for 2 h. Mole% of lipids used for the acceptor and donor liposomes in FRET-based lipid transfer assays are shown in Supplementary Table 1. The dried lipid films were hydrated with HK buffer [50 mM HEPES, 120 mM potassium acetate (pH 7.5)]. Liposomes were then formed by five freeze-thaw cycles (liquid N₂ and 37 °C water bath) followed by extrusion 11 times using Nanosizer with a pore size of 100 nm (T&T Scientific Corporation). All liposome-based lipid transfer assays were performed in a 96-well plate (Corning) using a Synergy H1 microplate reader (Biotek; Gen5 3.09 software). All reactions were performed in 60 μl volumes with a final lipid concentration at 0.6 mM. A buffer of the purified proteins was replaced with HK buffer prior to all the lipid transfer assays. The values of blank solution (buffer only without liposomes or proteins) were subtracted from all the values from each time point.

*FRET-based PIP [PI(3)P, PI(4)P, PI(3,5)P₂, PI(4,5)P₂, and PI(3,4,5)P₃] transfer assays*. The PIP transfer assays were performed as previously described[97] with slight modifications. Donor [4% PIP (PI(3)P, PI(4)P, PI(3,5)P₂, PI(4,5)P₂, or PI(3,4,5)P₃), 2% Rhodamine-PE, 94% DOPC], and acceptor [100% DOPC] liposomes were added at a 1:1 ratio. PIP transfer mediated by SMP_PDZD-8 [0, 2, 4, 6, or 8 μM for PI(4,5)P₂ and 8 μM for all other PIPs] was followed by measuring the NBD fluorescence signals of NBD-PH_FAPP (0.5 μM) at 530 nm (bandwidth 12.5 nm) on excitation at 460 nm (bandwidth 12.5 nm) for 60 min at every 15 s at room temperature. The amount of PIP (in μM) transferred from donor to acceptor liposomes corresponds to $6 \times F_{Norm}$, where $F_{Norm} = 0.5 \times (F - F_0/F_{eq} - F_0)$. $F_0$ corresponds to the NBD signal prior to the addition of SMP_PDZD-8, and $F_{eq}$ corresponds to the equilibrium condition where both donor and acceptor liposomes contain equal percentages of PIP [donor liposome: 2% PIP, 2% Rhodamine-PE, 96% DOPC; acceptor liposome: 2% PIP, 98% DOPC]. Liposome-binding assays indicated that NBD-PH_FAPP (0.5 μM) is completely bound to the membrane for a surface-accessible amount of PIP above 4 μM (Supplementary Fig. 7j). Initial transport rates (Fig. 7c, d) were calculated by determining the slope of the initial linear portion of the transfer reaction (first 10 min after addition of the proteins). To obtain the stable NBD signal at the equilibrium condition, the NBD signal was monitored for 15 min before the actual reading for each condition tested.

*FRET-based PS transfer assays*. The PS transfer assays were performed as previously described[98] with slight modifications. Donor [4% brain PS, 2% Rhodamine-PE, 94% DOPC] and acceptor [100% DOPC] liposomes were added at a 1:1 ratio. PS transfer mediated by SMP_PDZD-8 (0, 2, 4, 6, or 8 μM) was followed by measuring the NBD fluorescence signals of NBD-C2_Lact (0.5 μM) at 530 nm (bandwidth 12.5 nm) on excitation at 460 nm (bandwidth 12.5 nm) for 60 min at every 15 s at room temperature. The amount of PS (in μM) transferred from donor to acceptor liposomes corresponds to $6 \times F_{Norm}$, where $F_{Norm} = 0.5 \times (F - F_0/F_{eq} - F_0)$. $F_0$ corresponds to the NBD signal prior to the addition of SMP_PDZD-8, and $F_{eq}$ corresponds to the equilibrium condition where both donor and acceptor liposomes contain equal percentages of PS [donor liposome: 2% brain PS, 2% Rhodamine-PE, 96% DOPC; acceptor liposome: 2% brain PS, 98% DOPC]. Initial transport rates (Supplementary Fig. 7d) were calculated by determining the slope of the initial linear portion of the transfer reaction (first 10 min after addition of the proteins). To obtain the stable NBD signal at the equilibrium condition, the NBD signal was monitored for 15 min before the actual reading for each condition tested.

*FRET-based NBD-labeled lipid (NBD-PE, NBD-ceramide, and NBD-PA) transfer assays*. Donor [4% NBD-labeled lipid (NBD-PE, NBD-PA, or NBD-ceramide), 2% Rhodamine-PE, 98% DOPC] and acceptor [100% DOPC] liposomes were added at a 1:1 ratio. Transfer of NBD-labeled lipid-mediated by SMP_PDZD-8 (0, 2, 4, 6, or 8 μM) was followed by measuring the NBD fluorescence signals of NBD-labeled lipid at 538 nm on excitation at 460 nm for 60 min at every 15 s at room temperature. Ten microliters of 2.5% DDM solution was added into each well at the end of 60 min recording, and maximum NBD signals of NBD-labeled lipid in solubilized liposomes were measured for another 10 min. The values at the end of 10 min recording were considered as maximum NBD fluorescence and used to calculate the percentage of max NBD fluorescence (Supplementary Fig. 7e–h). All data were corrected by subtracting the $t = 0$ values obtained in the absence of the protein.

*FRET-based DHE transfer assays*. The DHE transfer assays were performed as previously described[99]. Donor [10% DHE, 90% DOPC] and acceptor [2.5% DNS-PE, 97.5% DOPC] liposomes were added at a 1:1 ratio. Reactions were initiated by the addition of SMP_PDZD-8 (0, 2, 4, or 6 μM) to the mixture of donor and acceptor liposomes. The fluorescence intensity of DNS-PE (i.e., FRET signals), resulting from FRET between DHE (excited at 310 nm) and DNS-PE, was monitored at 525 nm every 15 s over 30 min at room temperature. Data were expressed as the number of DHE molecules transferred using a calibration curve, which was obtained by measuring FRET signals of liposomes containing various percentages of DHE, 2.5% DNS-PE, and DOPC[99]. Then, the mole number of the transferred DHE from the donor to acceptor liposomes was obtained using the formula derived from the linear fit of the calibration curve. The DHE transport rates (Supplementary Fig. 7i) were calculated by determining the slope of the transfer reaction (first 10 min after addition of the proteins and buffer).

**Image processing and quantification analysis**. Fiji (ImageJ)[130] was used to perform all image processing and quantitative analysis.

*Quantification of PI(4,5)P₂ puncta (Fig. 1c–e and Supplementary Fig. 1b–d).* The same arbitrary threshold (110-65535) was applied to the time-lapse images of 5 s interval for both control and mutant embryos, expressing PI(4,5)P₂ biosensor, mCherry::PH$^{PLC\delta1}$ to segment PI(4,5)P₂ puncta from the background (pixel size cut-off: 12-infinity). Quantification was performed every 12 frame, or 60 s interval. The area and mean fluorescence intensity of each PI(4,5)P₂ puncta were measured, and total fluorescence intensity of all PI(4,5)P₂ puncta was calculated for each frame. The number of PI(4,5)P₂ puncta was manually determined based on the segmented images for each frame.

*Quantification of NMY-2 puncta (Figs. 3b–d, 7e and Supplementary Figs. 3b, c, 7m).* The same arbitrary threshold (118-65535) was applied to the time-lapse images of 5 s interval for both control and mutant embryos, expressing NMY-2::GFP, to segment NMY-2 puncta from the background (pixel size cut-off: 15-infinity). Quantification was performed every 12 frame, or 60 s interval. The area and mean fluorescence intensity of each NMY-2 puncta were measured, and total fluorescence intensity of all NMY-2 puncta was calculated for each frame. The number of NMY-2 puncta was manually determined based on the segmented images for each frame.

*Quantification of cytoplasmic PI(4,5)P₂ structures (Fig. 4b).* The same arbitrary threshold (110-65535) was applied to the time-lapse images of 5 s interval for both control and mutant embryos, expressing PI(4,5)P₂ biosensor, mCherry::PH$^{PLC\delta1}$, to segment cytoplasmic PI(4,5)P₂ structures from the background (pixel size cut-off: 12-infinity). Process>Noise>Despeckle command was used to further remove the background signals from the segmented images. Quantification was performed every 12 frame, or 60 s interval. ROI was drawn around the entire cytoplasmic region excluding the PM, and the area and mean fluorescence intensity of mCherry::PH$^{PLC\delta1}$ signals were measured. Total fluorescence intensity of mCherry::PH$^{PLC\delta1}$ (product of area and mean fluorescence intensity) was then plotted.

*Quantification of cytoplasmic PI(4,5)P₂ for early embryos in the uterus region of adult worms (Supplementary Fig. 5b).* The same arbitrary threshold (110-65535) was applied to the images from RNAi-treated control and mutant worms, expressing PI(4,5)P₂ biosensor, mCherry::PH$^{PLC\delta1}$, to segment cytoplasmic PI(4,5)P₂ structures from the background (pixel size cut-off: 12-infinity). Process>Noise>Despeckle command was used to further remove the background signals from the segmented images. ROI was drawn around the entire cytoplasmic region excluding the PM of the first embryo in the uterus region of each adult worm, and the area and mean fluorescence intensity of mCherry::PH$^{PLC\delta1}$ signals were measured. Total fluorescence intensity of mCherry::PH$^{PLC\delta1}$ (product of area and mean fluorescence intensity) was then plotted.

*Quantification of cytoplasmic myosin structures (Fig. 4c).* The same arbitrary threshold (188-65535) was applied to the time-lapse images of 5 s interval for both control and mutant embryos, expressing NMY-2::GFP to segment cytoplasmic myosin structures from the background (pixel size cut-off: 15-infinity). Quantification was performed every 12 frame, or 60 s interval. ROI was drawn around the entire cytoplasmic region excluding the PM, and the area and mean fluorescence intensity of NMY-2::GFP signals were measured. Total fluorescence intensity of NMY-2::GFP (product of area and mean fluorescence intensity) was then plotted.

*Quantification of the size distribution of cytoplasmic PI(4,5)P₂ structures (Fig. 4d).* Line was manually drawn across the largest cytoplasmic PI(4,5)P₂ structure in each frame of the time-lapse images of 5 s interval for both control and mutant embryos, expressing PI(4,5)P₂ biosensor, mCherry::PH$^{PLC\delta1}$, and the diameter of the PI(4,5)P₂ structure was determined. Quantification was performed every 12 frame, or 60 s interval. Gaussian curves were superimposed onto the frequency distribution of the diameter of cytoplasmic PI(4,5)P₂ structures.

*Quantification of PI(4,5)P₂ levels in endosomal membranes and the PM (Figs. 4f, 7f).* The same arbitrary threshold (108-65535) was applied to a single frame (beginning of cleavage furrow ingression) of the original time-lapse images of 5 s interval for both control and mutant embryos, expressing PI(4,5)P₂ biosensor, mCherry::PH$^{PLC\delta1}$, to segment PI(4,5)P₂-positive membranes from the background (pixel size cut-off: 5-infinity). Process>Noise>Despeckle command was used to further remove the background signals from the segmented images. The area and mean fluorescence intensity of mCherry::PH$^{PLC\delta1}$ signals at endosomal membranes and total membranes were measured, and total fluorescence intensity of mCherry::PH$^{PLC\delta1}$ (product of area and mean fluorescence intensity) was used as an estimate of PI(4,5)P₂ levels in each compartment. First, ROI was drawn around cytoplasmic mCherry::PH$^{PLC\delta1}$ signals to determine PI(4,5)P₂ levels in endosomal membranes. Second, ROI was drawn around the entire embryo to determine total PI(4,5)P₂ levels. To obtain PI(4,5)P₂ levels in the PM, mCherry::PH$^{PLC\delta1}$ signals in endosomal membranes was subtracted from the total mCherry::PH$^{PLC\delta1}$ signals.

*Line scan analysis (Fig. 2d–g and Supplementary Fig. 8c–e).* Line was drawn across vesicular structures to plot fluorescence profiles of both GFP and mCherry/wrmScarlet signals. The normalized fluorescence profiles were plotted on the same graph to visualize the extent of co-localization.

*Quantification of cortical ruffling (Fig. 5b).* Representative images were selected from the polarity establishment phase and the mitosis phase of the same embryos undergoing early embryogenesis. The outline of each embryo was drawn manually for both the polarity establishment phase and mitosis phase, and the perimeter of the outline was measured. The perimeter of the embryo at the polarity establishment phase was divided by that of the polarity maintenance phase to obtain the cortical ruffling index.

*Quantification of cytoplasmic and PM CAV-1 (Fig. 8b and Supplementary Fig. 9c).* The same arbitrary threshold (160-65535) was applied to the time-lapse images of 5 s interval for both control and mutant embryos expressing CAV-1::GFP to segment cytoplasmic and PM CAV-1 from the background (pixel size cut-off: 12-infinity). Process>Noise>Despeckle command was used to further remove the background signals from the segmented images. Quantification was performed every 12 frame or 60 s interval. For cytoplasmic CAV-1 signals, ROI was drawn around the entire cytoplasmic region excluding the PM, and the area and mean fluorescence intensity of cytoplasmic CAV-1::GFP signals were measured. For PM CAV-1 signals, ROI was drawn around the entire embryo to determine total CAV-1 levels. Cytoplasmic CAV-1::GFP signals were then subtracted from the total CAV-1::GFP signals to obtain PM CAV-1::GFP signals. The total fluorescence intensity of CAV-1::GFP (product of area and mean fluorescence intensity) was then plotted.

*Quantification of Manders' Overlap Coefficient (Fig. 8c and Supplementary Figs. 6c, d, 8b, 9b).* JACoP plugin was used to obtain the Manders' Overlap Coefficient[131]. In brief, the arbitrary threshold was applied to a single frame (beginning of cleavage furrow ingression) of the images of control and mutant embryos (for Fig. 8c and Supplementary Figs. 8b, 9b) or images of COS-7 cells expressing indicated proteins (for Supplementary Fig. 6c, d) to segment fluorescence signals of interest from the background. The overlapping fractions of the fluorescence signals derived from the plugin were then plotted.

**Statistics and reproducibility**. No statistical method was used to predetermine sample size, and the experiments were not randomized for live-cell imaging. Sample size and information about replicates are described in the figure legends. For obtaining representative micrographs, experiments were independently conducted at least eight times to confirm reproducibility. An exception is an experiment in Supplementary Fig. 3, where experiments were conducted at least three times to confirm reproducibility. For biochemical assays, all experiments were independently conducted at least three times to confirm reproducibility. Comparisons of data were carried out by the two-tailed unpaired Student's *t*-test or one-way ANOVA, followed by Dunnett's or Tukey's corrections for multiple comparisons as appropriate with Prism 8.0.1 (GraphPad software). Linear fits for lipid transfer assay and nonlinear fits (Gaussian curves) for the size distribution of cytoplasmic PI(4,5)P₂ structures were performed using Prism 8.0.1. Unless $p < 0.0001$, exact $P$ values are shown within the figure legends for each figure. $p > 0.05$ was considered not significant. All data were presented as mean ± SEM unless otherwise noted.

## Data availability
The authors declare that the data supporting the findings of this study are available within the paper and its supplementary information file. Other data are available from the corresponding author upon reasonable request. Reagents and strains generated for this study are available directly from the authors upon request. The structure for the SMP domain of E-Syt2 can be accessed as PDB 4P42. Source data are provided with this paper.

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

## Acknowledgements

We thank Raihanah Harion and Dylan Hong Zheng Koh for sharing reagents and discussion. This work was supported in part by the Ministry of Education, Singapore, under its Academic Research Fund Tier 2 Award (MOE2017-T2-2-001), Academic Research Fund Tier 1 Award (2018-T1-001-023), a Nanyang Assistant Professorship (NAP), and a Lee Kong Chian School of Medicine startup grant (LKCMedicine-SUG) to Y.S. T.N. was supported by a fellowship from the Japanese Society for Promotion of Science.

## Author contributions

All authors participated in the design of experiments, data analysis, and interpretation. D.J. performed all the *C. elegans* experiments, including all the genetic manipulations and embryo imaging. J.S. participated in the generation of some *C. elegans* transgenic lines with the single copy transgene integration (MosSCI) method. D.J. performed structural modeling of the SMP domain of PDZD-8 and designed mutations with the help from B.E. B.E., D.D., and Y.S. participated in designing the in vitro lipid transport assays that were performed by D.D. and B.E. D.D. performed all the protein purification work with the help of B.E. T.N. performed the mammalian cell-based experiments. Y.S. wrote the manuscript with input from all the authors.

## Competing interests

The authors declare no competing interests.
