## [Peer Review File · Nature Communications]

PDZD-8 and TEX-2 regulate endosomal PI(4,5)P2 homeostasis via lipid transport to promote embryogenesis in *C. elegans*REVIEWER COMMENTS

Reviewer #1 (Remarks to the Author):

The cellular distribution and metabolism of phosphoinositides are major questions in cell biology. PIP2 primarily localizes to the plasma membrane but can presumably enter the cells through endocytosis. How PIP2 is degraded/metabolized in internal organelles e.g. endosomes remains unknown. Here, using genetic and cell biological approaches, the authors demonstrate that the SMP domain containing proteins PDZD-8 and TEX2 can regulate endosomal PIP2 degradation. Specifically, PDZD-8 localizes to contact sites between the ER and late endosomes, and delivers PIP2 to the ER for hydrolysis by ER-localized 5 phosphatases. Overall, this is an exciting study. There is a large amount of data that provide very good support to the central hypothesis. The results are of high general interest. I have only some minor concerns/suggestions.

1. An FP-tagged PIP2 sensor was used for detecting PIP2 in this study. While not absolutely required, it would be nice to also use PIP2 antibodies in one of the experiments.
2. Figure 1B, do we know for sure the source of the PIP2? Is it from the PM? Does it disappear when endocytosis is blocked? ORP2 can also deliver PIP2 from PM to internal compartments. Is ORP2 upregulated in the QKO cells? At least discuss these possibilities.
3. What if some PI4P-5 Kinases mislocalizes to late endosomes in the QKO cells, thereby generating more PIP2? This possibility needs to be discussed if no experiments can be done to prove/disprove it.
4. What happens to endosomal PIP2 if INPP5K, OCRL and UNC-26 are all knocked down? Does PDZD8 colocalize with INPP5K?

Reviewer #2 (Remarks to the Author):

Using *C. elegans* as a model for embryogenesis, the authors have investigated factors that control the levels of PI5P2 in the PM and endosomal pathway and how this effects cortical actinomyosin contraction during cell division. They show that paired KO of two SMP proteins PDZD8 and TEX2 resulted in abnormal accumulation of PI45P2 on large endosomes, and that these endosomes also accumulated proteins involved in actinomyosin contraction. Brood size was only slightly reduced and survival was unaffected. However, ablation of PI45P2 phosphatases OCRL and UNC-26/synaptojanin caused more extensive accumulation of endosomal PI45P2, reduced brood size and a 50% reduction in survival, suggesting two parallel pathways for removal of PI45P2 from endosomes. They further show that the SMP domain of PDZD8 transfers PI45P2 to acceptor liposomes in vitro, and that mutation the SMP domains of PDZD98 and TEX2 in vivo reproduced the PI45P2 accumulation of endosomes observed with the KO animals. Together with evidence that PDZD8 localized to ER-endosomes, they conclude that the SMP proteins transfer PI45P2 from the endosomes thus preventing buildup and proper actinomyosin activity and cytokinesis.

The study is convincing in the sense that it shows that a PDZD8/TEX2-dependent pathway functions in parallel with endosomal phosphatases to maintain PI45P2 levels in the PM/endosomal pathway, which is critical in early stages of embryogenesis. There are several concerns regarding the evidence the authors have presented to support the role of SMP proteins as specific PI45P2 transporters at ER-endosomes contact sites.

1. Both PDZD8 and Tex2 are clearly ER proteins but the evidence that they interact with Rab7 positive endosomes is based on overexpression in Cos cells. Th authors need to show more definitive evidence that endosomes containing endogenous Rab7 interact with PDZD8/Tex2 at the ER.

2. While the authors show that the isolated SMP domain of PDZD8 can transfer PI45P2 in vitro, other studies have suggested that this domain is non-specific and transfers other phospholipids. If the binding of the SMP domain is non-specific, then the authors are measuring a minor activity for PI45P2 that may not be physiologically relevant if there is competition for binding/transfer by other more abundant lipids. Indeed the transfer of PI45P2 seems rather slow and does not go to equilibrium (actually on about 30%) even after 60 min. The authors must show that the transfer of PI45P2 is a preferred ligand relative to other PIPs and phospholipids.

3. In general, the conclusion presented in the discussion (notably at the bottom of page 14 where they claim that PDZD8/Tex2 transfer PI45P2 to the ER etc...) is not convincing. Key evidence of PI45P2 transfer to the ER would be accumulation in that organelle or removal by a ER-associated phosphatase (as they speculate in the Discussion, Fig. 7E). Without evidence of a vectorial transfer to the ER (or a clear preference for PI45P2, see above) it is not possible to conclude that PDZD8/TEX2 are involved in PI45P2 transport. The SMP proteins could be tethering factors that facilitate membrane interactions guided by PI45P2 sensing. In the context of the full-length ER-associated PDZD8, this could be related to a tethering activity at the ER-endosome contact site or to presentation of the lipid to a phosphatase for hydrolysis.

Reviewer #3 (Remarks to the Author):

The manuscript by Jeyasimman et al provides evidence for partly redundant mechanisms for removal of PI(4,5)P2 from endosomal membranes in an intact in vivo model. The authors identify ER associated lipid transfer proteins PDZD-8 and TEX-2 in this process, along with PI5-phosphatases OCRL-1 and UNC-26. Importantly the authors show that without these PI(4,5)P2 regulators actomyosin contractile machinery is inappropriately recruited to endosomes, leading to failure in actomyosin driven cytokinesis during embryogenesis. The authors further provide in vitro evidence that PDZD-8 can transport PI(4,5)P2 between membranes. Overall this work appears well done and is likely to interest a broad group of cell and developmental biologists.

Some weaknesses are detailed below:

(1) The biggest issue that needs to be addressed relates to the claim that PI(4,5)P2 overaccumulation on endosomes affects endosome identity.

The authors write "Indeed, some large PI(4,5)P2-positive vesicles were transiently decorated by GFP::RAB-7. However, a significant fraction of the large PI(4,5)P2-positive vesicles were not decorated by GFP::RAB-7. The aberrant accumulation of PI(4,5)P2 in late endosomes may therefore have caused massive disruption to the identity of endosomes. To this end, we assessed the potential co-localization of large PI(4,5)P2-positive vesicles with two other endosomal markers, namely GFP::RAB-5 for early endosomes and GFP::RAB-11.1 for recycling endosomes (Figure 7A). Strikingly, a significant fraction of these large PI(4,5)P2-positive vesicles were also transiently decorated by GFP::RAB-5 and GFP::RAB-11.1, supporting the notion that massive accumulation of PI(4,5)P2 in late endosomes disrupted their identity."

This statement indicates a misunderstanding of endosome identity. It is entirely normal for endosomes to display multiple sequentially acting Rab proteins on their limiting membranes. This has been known for more than 20 years, as originally shown by Zerial and colleagues (Sönnichsen et al, 2000; PMID 10811830). This would have been clear if control experiments with the same markers was performed on wild-type embryos. The authors would need extensive additional experimentation to support the claim of a change in endosome identity, which should also include tests of endosome function. There are endosomal cargo markers (e.g. GFP::CAV-1) to gain insight into endosomal function, at least the degradative aspect) in the *C. elegans* embryo. I am also concerned that it is only a small fraction of endosomes that appear enlarged and PI(4,5)P2 positive – a feature of the data that was not clearly

acknowledged. What fraction of late endosomes overaccumulate PI(4,5)P2?

(2) The data on lipid transfer in vitro is a nice addition to this paper, but more data is needed to understand the specificity of PDZD-8 SMP-domain-mediated transfer. At the very least other negatively charged phospholipids should be tested, including PA, PS, PI, PI4P, PI3P, and PI(3,5)P2.

(3) Colocalization experiments in general should be quantified for a population (e.g. Manders or Pearson's coefficients), not simply shown as single examples with line-scans. In general quantification was better performed for cytokinesis proteins and less well for endosome proteins (e.g. Fig 6 and 7).

Reviewer #4 (Remarks to the Author):

Jayasimman et al., investigated the functions of SMP proteins in *C.elegans* early embryo. They systematically tested the combinations of SMP protein KO lines and found that PDZD-8 and TEX-2 function redundantly in removing PI(4,5)P2 from the cytoplasmic compartment. The cytoplasmic PI(4,5)P2 in the PDZD-8; Tex-2 double knockouts is overlapped with NMY-2 and mostly with the recycling endosome marker RAB-11. The accumulation of PI(4,5)P2 was enhanced by additional deletion of PI(4,5)P2 phosphatases, UNC-26 and OCRL-1. In the unc-26; DKO embryo treated with OCRL-1 RNAi, the cortical ruffling was impaired and also the percentage of eggs that hatched was reduced. The in vitro analysis showed that the SMP domain of PDZD-8 transfers PI(4,5)P2. Finally, they showed that SMP domains of PDZD-8 and TEX-2 are required for removing PI(4,5)P2 from the cytoplasmic compartment.

Non-vesicular lipid transport at the membrane contact sites is an important topic in the field and the link between the SMP proteins and PI(4,5)P2 accumulation is an interesting finding.

The caveat is that the experiments explaining the roles of SMP domain proteins are mostly done in vitro or poorly done using mammalian cell lines. This hampers the soundness of the hypothesis the paper claims.

Major points.

To understand how cytoplasmic PI(4,5)P2 is generated and removed by SMP proteins, it is necessary to clarify the distribution of TEX-2 and PDZD-8 in the *C.elegans* embryo. The localization of ectopically expressed PDZD-8 together with again ectopically expressed Rab-7 in the mammalian system does not reflect the endogenous localization of PDZD-8 in *C.elegans* embryo. In addition, in the experiment using mammalian cells, TEX-2 and PDZD-8 localize in different compartments. This is not consistent with the model the authors are proposing, namely the redundant function of TEX-2 and PDZD-8 (seen in Figure 7E). These points need to be addressed adequately.

One possible experiment to connect the genetic experiments and the phenotype is overexpressing the phosphatases in the DKO. Experiments using ER-anchored phosphatases might support their hypothesis.

The interpretation of Figure 4A does not seem to be consistent with their hypothesis to this reviewer. According to their model, even in the presence of PDZD-8 and TEX-2, PI(4,5)P2 should be transferred from somewhere to the cytoplasmic compartment, and then PI(4,5)P2 is exposed or further transferred to the ER. Therefore, the accumulation of PI(4,5)P2 should be observed in the unc-26 embryo treated with ocrl-1 RNAi.

The paper would be strengthened if the in vitro lipid transport assay was more thorough. The specificity of SMP's lipid binding has been discussed but is still very controversial. Even though the paper claims PDZD-8 transports PI(4,5)P2 selectively, they examined only PE as a control. At least other lipid species already claimed in other papers (Jeong et al., 2017, Shirane et al. 2020, etc.)

should be tested. If *C.elegans* PDZD-8 and TEX-2 specifically transfer PI(4,5)P₂, the result will reinforce their model.

In line with this, the statements 'the SMP domain of PDZD-8 selectively transports PI(4,5)P₂ between membranes' and 'PDZD-8 transports PI(4,5)P₂ from endosomes to the ER via the SMP domain in vivo to maintain endosomal PI(4,5)P₂ homeostasis' are overstatements.

Minor point.

The data presented show that PDZD-8, TEX-2 DKO is a prerequisite for the phosphatases KO to show the PI(4,5)P₂ accumulation phenotype. In this regard, "PDZD-8 and TEX-2 act in parallel to the PI(4,5)P₂ phosphatases" is misleading.

REBUTTAL TO THE COMMENTS RAISED BY THE REVIEWERS

We thank the editor and the reviewers for the many constructive comments and suggestions. We have carried out extensive new experimentation to address all the concerns and made necessary changes to the texts.

We have now

- 1) determined the subcellular localization of PDZD-8 and RAB-7 both at endogenous expression levels and provided evidence that PDZD-8 localizes to the ER and additionally localizes to the sites of contact between the ER and late endosomes marked by RAB-7 in *C. elegans* early embryos (for Reviewers #2 and #3)
- 2) shown that the PDZD-8 SMP domain is capable of transporting various PIPs, including PI(3)P, PI(4)P, and PI(3,5)P₂, phosphatidylserine (PS), and NBD-labelled phosphatidic acid (PA) in addition to PI(4,5)P₂ *in vitro* [but not cholesterol or other NBD-labelled lipids, including NBD-phosphatidylethanolamine (PE), and NBD-ceramide] (for Reviewers #2, #3, and #4)
- 3) provided further evidence that PI(4,5)P₂ accumulated throughout the endosomal system in mutant embryos (for Reviewers #2 and #3)
- 4) provided evidence that the degradative capacity of the endosomal system is disrupted in the absence of PDZD-8, TEX-2, UNC-26, and OCRL-1 (for Reviewer #3)
- 5) provided evidence that the bulk of PI(4,5)P₂ accumulated in cytoplasm of mutant embryos originated from the plasma membrane (PM) (for Reviewer #1)
- 6) provided more rigorous quantification of images and strengthened the live cell imaging studies with addition of new experiments/images as requested.

A specific list of changes is indicated below, followed by a point-by-point rebuttal from page 4.

Figure 6 is now split into **new Fig. 6** and **new Fig. 7** with additional data.

Supplementary Figure 6 is now split into **new Supplementary Fig. 6** and **new Supplementary Fig. 7** with additional data.

Figure 7 is now in **Supplementary Fig. 8** and replaced by **new Fig. 8**.

New Supplementary Fig. 9 is added with newly generated data.

We now have eight main figures (**Fig. 1-8**) and nine supplementary figures (**Supplementary Fig. 1-9**).

New Supplementary Movie 6 and **New Supplementary Movie 11** are added.

Previous **Video 6** is now **Supplementary Movie 7**.

Previous **Video 7** is now **Supplementary Movie 8**.

Previous **Video 8** is now **Supplementary Movie 9**.

Previous **Video 9** is now **Supplementary Movie 10**.

The following data are new:

Fig. 4a (updated images showing that the presence of some cytoplasmic PI(4,5)P₂-positive structures in *unc-26* mutant embryos treated with OCRL-1 RNAi)

Fig. 6a (updated data showing Manders' coefficient for the colocalization analysis of PDZD-8 with various organelles, including the ER and endosomes)

Fig. 6b (showing the association of late endosomes labelled by endogenously tagged mCherry::RAB-7 with the ER labelled by GFP::C34B2.10)

Fig. 6c, d [showing the presence of distinct PDZD-8::mNeonGreen puncta on a small fraction of mCherry::RAB-7-positive late endosomes (both endogenously tagged)]

Fig. 7d (showing the transport efficiency of the PDZD-8 SMP domain for various PIPs)

Fig. 8a, b [showing the cytoplasmic accumulation of CAV-1::GFP (model cargo for degradation) and PI(4,5)P₂-positive vesicles (marked by mCherry::PH^{PLCδ1}) in mutant embryos compared to wild-type embryos]

Fig. 8c [showing the extensive overlap of undegraded CAV-1::GFP signals with PI(4,5)P₂-positive vesicles (marked by mCherry::PH^{PLCδ1}) and late endosomes (marked by endogenously tagged mCherry::RAB-7) in mutant embryos]

Fig. 8d (updated cartoon showing the suggested functions of PDZD-8 and TEX-2)

Supplementary Fig. 2c [showing the absence of endogenously tagged PPK-1::EGFP (type I PIP kinase or PIP5K) on PI(4,5)P₂-positive vesicles in mutant embryos]

Supplementary Fig. 5a, b [showing the suppression of the cytoplasmic accumulation of PI(4,5)P₂-positive vesicles in mutant embryos upon RNAi against DYN-1 (dynamain)]

Supplementary Fig. 6a, b [updated data additionally showing the localization of endogenously tagged PDZD-8 (PDZD-8::mNeonGreen) in the head region and the early embryo of *C. elegans*]

Supplementary Fig. 6c (updated data showing Manders' coefficient for the colocalization analysis of TEX-2 with various organelles, including the ER and endosomes)

Supplementary Fig. 6d (showing the absence of major association of the endogenously tagged TEX-2::EGFP and late endosomes labelled by endogenously tagged mCherry::RAB-7)

Supplementary Fig. 7c (showing the representative traces of PIP transport mediated by the PDZD-8 SMP domain)

Supplementary Fig. 7d (showing the representative traces of PS transport mediated by the PDZD-8 SMP domain and its transport efficiency for PS)

Supplementary Fig. 7e-h (showing characterization of the transport efficiency of the PDZD-8 SMP domain for various NBD-labelled lipids)

Supplementary Fig. 7i (showing the absence of DHE transport by the PDZD-8 SMP domain)

Supplementary Fig. 7j (showing the binding curve of the NBD-labelled PH domain of FAPP protein to various PIPs and other phospholipids)

Supplementary Fig. 7k (showing the efficiency of PI(4,5)P₂ transport by the PDZD-8 SMP domain in the presence of various anionic lipids in donor liposomes)

Supplementary Fig. 8b (showing Manders' coefficient for the presence of PI(4,5)P₂ on various endosomes)

Supplementary Fig. 9a [showing the accumulation of CAV-1::GFP in late endosomes (marked by endogenously tagged mCherry::RAB-7) in mutant embryos compared to wild-type embryos]

Supplementary Fig. 9b [showing Manders' coefficient for the association of CAV-1::GFP, late endosomes (marked by endogenously tagged mCherry::RAB-7) and PI(4,5)P₂-positive vesicles (marked by mCherry::PH^{PLCδ1}) in mutant embryos]

Supplementary Fig. 9c [showing the quantification of PM CAV-1::GFP signals in mutant embryos compared to wild-type embryos]

One additional experiment is shown for the reviewers only (response to Reviewer #4).

The following *C. elegans* strains were newly generated for the revision:

- SAH634: *rab-7(syb4141[mCherry::RAB-7] II; unc-119(ed3) III; ojl523 [pie-1p::GFP::C34B2.10]*
- SAH607: *C53B4.4(syb4099[C53B4.4::mNeonGreen]) IV; rab-7(syb4141[mCherry::RAB-7]) II*
- SAH631: *pwIs28[pie-1p::cav-1::GFP(7) + unc-119(+)]*; *ItIs44[pie-1p::mCherry::PH(PLC1delta1) + unc-119(+)]*

- SAH587: *F55C12.5(syb670) II; C53B4.4(syb664) unc-26(s1710) IV; pwls28[pie-1p::cav-1::GFP(7) + unc-119(+)]*; *Itls44[pie-1p::mCherry::PH(PLC1delta1) + unc-119(+)]*
- SAH655: *F55C12.5(syb670) rab-7(syb4141[mCherry::RAB-7]) II ; C53B4.4(syb664) unc-26(s1710) IV; pwls28[pie-1p::cav-1::GFP(7) + unc-119(+)]*
- SAH630: *ppk-1(syb4109[PPK-1::EGFP] I; Itls44[pie-1p::mCherry::PH(PLC1delta1) + unc-119(+)]*
- SAH614: *ppk-1(syb4109[PPK-1::EGFP] I; F55C12.5(syb670) II ; esyt-2(syb709) III; C53B4.4(syb664) IV; R11G1.6(tm10626) X; Itls44[pie-1p::mCherry::PH(PLC1delta1) + unc-119(+)]*
- SAH638: *C53B4.4(syb4099[C53B4.4::mNeongreen]) IV; ocfls2[pie-1p::mCherry::sp12::pie-1 3'UTR + unc-119(+)]*
- SAH633: *F55C12.5(yas46[F55C12.5::EGFP^{3xFLAG}]) rab-7(syb4141[mCherry::RAB-7]) II*
- SAH617: *rab-7(syb4141[mCherry::RAB-7]) II; pwls28[pie-1p::cav-1::GFP(7) + unc-119(+)]*

-Sentences from the reviewers' comments are *in italics*
-Our responses are in blue

Reviewer #1 (Remarks to the Author):

The cellular distribution and metabolism of phosphoinositides are major questions in cell biology. PIP2 primarily localizes to the plasma membrane but can presumably enter the cells through endocytosis. How PIP2 is degraded/metabolized in internal organelles e.g. endosomes remains unknown. Here, using genetic and cell biological approaches, the authors demonstrate that the SMP domain containing proteins PDZD-8 and TEX2 can regulate endosomal PIP2 degradation. Specifically, PDZD-8 localizes to contact sites between the ER and late endosomes, and delivers PIP2 to the ER for hydrolysis by ER-localized 5 phosphatases. Overall, this is an exciting study. There is a large amount of data that provide very good support to the central hypothesis. The results are of high general interest. I have only some minor concerns/suggestions.

We thank the reviewer for these very positive comments.

1. An FP-tagged PIP2 sensor was used for detecting PIP2 in this study. While not absolutely required, it would be nice to also use PIP2 antibodies in one of the experiments.

Reply: We thank the reviewer for this suggestion. We attempted to detect the aberrant cytoplasmic PI(4,5)P₂ accumulation in *unc-26*; DKO early embryos treated with OCRL-1 RNAi using immunostaining with anti-PIP2 antibodies [2C11] (ab11039) (e.g., PMID: 30201859). However, the mutant embryos were extremely fragile, making it difficult to maintain the integrity of cellular membranes during the relatively harsh immunostaining protocol often used for *C. elegans* embryos. Thus, we were unable to perform experiments successfully using anti-PIP2 antibodies.

2. Figure 1B, do we know for sure the source of the PIP2? Is it from the PM? Does it disappear when endocytosis is blocked? ORP2 can also deliver PIP2 from PM to internal compartments. Is ORP2 upregulated in the QKO cells? At least discuss these possibilities.

Reply: We thank the reviewer for these comments. To determine whether PI(4,5)P₂ accumulated in cytoplasm of the mutant embryos originated from the plasma membrane (PM), endocytosis was blocked by treating the mutant embryos with RNAi against DYN-1, a sole *C. elegans* homolog of dynamin, an essential protein that catalyzes membrane fission during clathrin-mediated endocytosis (PMID: 22233676). DYN-1 RNAi resulted in severe embryonic lethality after fertilization, making it difficult to isolate embryos. Thus, one cell embryos were directly imaged within uterus of adult worms without dissection. However, we could not detect cytoplasmic PI(4,5)P₂ puncta reliably in QKO mutants because of attenuated fluorescence signals with this method. This made it difficult to evaluate the effect of DYN-1 RNAi in this particular condition.

Therefore, we performed similar experiments in *unc-26*; DKO early embryos treated with OCRL-1 RNAi instead (simultaneous depletion of UNC-26, OCRL-1, PDZD-8, and TEX-2 to enhance cytoplasmic PI(4,5)P₂ accumulation for better fluorescence signals). Remarkably, cytoplasmic accumulation of PI(4,5)P₂ was significantly suppressed by DYN-1 RNAi, suggesting that the bulk of PI(4,5)P₂ accumulated in cytoplasm of *unc-26*; DKO early embryos treated with OCRL-1 RNAi originated from the PM. These new results are now included in **new Supplementary Fig. 5a, b**.

We discussed the new data and also discussed the potential dysregulation of other lipid transfer proteins, including ORP2, in the section of discussion as shown below (please also see our comments on #3 regarding PIP5K).

“Our results suggested that PI(4,5)P₂ on these endosomes originated from the PM based on 1) the absence of ectopic accumulation of PPK-1, a sole homolog of PIP5K in *C. elegans*, on PI(4,5)P₂-positive endosomes in the absence of SMP proteins; 2) the suppression of cytoplasmic PI(4,5)P₂ accumulation in early embryos from *unc-26*; DKO mutants treated with OCRL-1 RNAi by blocking endocytosis with RNAi of DYN-1, a sole homolog of dynamin in *C. elegans*. In mammals, ORP2, a LTP that belongs to OSBP family, has been implicated in regulating the levels of PI(4,5)P₂ in various cellular compartments, including the PM⁵⁰ and recycling endosomes¹¹⁶. Thus, potential dysregulation of other LTPs involved in PI(4,5)P₂ regulation may have also contributed to aberrant PI(4,5)P₂ accumulation in the absence of SMP proteins.” (Page 16-17 of the revised manuscript)

3. What if some PI4P-5 Kinases mislocalizes to late endosomes in the QKO cells, thereby generating more PIP2? This possibility needs to be discussed if no experiments can be done to prove/disprove it.

Reply: We thank the reviewer for this comment. We used the CRISPR/Cas9-method to introduce a sequence encoding GFP-tagged PPK-1 (PPK-1::EGFP), a sole *C. elegans* homolog of phosphatidylinositol-4-phosphate 5' kinase (PIP5K) (PMID: 18037397), into its endogenous locus. We found that PPK-1::EGFP localized exclusively to the PM, where PI(4,5)P₂ is most enriched, in wild-type embryos, consistent with the major role of PPK-1 in generating the vast majority of PI(4,5)P₂ in the cell. We then examined the localization of PPK-1::EGFP in QKO embryos and found that PPK-1::EGFP remained tightly associated with the PM and that cytoplasmic PI(4,5)P₂ puncta in QKO embryos were not associated with PPK-1::EGFP. These results indicate that PPK-1 is not aberrantly recruited to late endosomes in QKO embryos to ectopically generate PI(4,5)P₂ in these organelles. These new data are now included in **new Supplementary Fig. 2c**.

4. What happens to endosomal PIP2 if INPP5K, OCRL and UNC-26 are all knocked down? Does PDZD8 colocalize with INPP5K?

Reply: We first attempted to deplete INPP5K, OCRL, and UNC-26/synaptojanin simultaneously by performing double RNAi against *cil-1* (INPP5K homolog in *C. elegans*) and *ocrl-1* (OCRL homolog in *C. elegans*) in the background of *unc-26* null mutant. However, *cil-1* RNAi was not effective; it failed to induce infertility [as observed in *cil-1(my15)* hypomorph mutant (PMID: 19781942)]. As an alternative approach, we attempted to generate a *unc-26; cil-1* double mutant by crossing *unc-26* null mutant and *cil-1(my15)* hypomorph mutant, but we could not obtain a viable progeny. This may suggest that UNC-26 becomes essential when the activity of CIL-1 is greatly attenuated.

To examine the endogenous localization of PDZD-8 in *C. elegans* early embryos, we generated a new knock-in strain that expresses mNeonGreen-tagged PDZD-8 from its endogenous locus (PDZD-8::mNeonGreen) by the CRISPR/Cas9-method (in this new strain, we inserted a sequence encoding mNeonGreen in-frame in the last common exon of the *pdzd-8* gene). Using this new strain, we found that PDZD-8 localizes to the ER in early embryos [based on extensive overlap of PDZD-8::mNeonGreen signal with the ER marker, mCherry::SP-12 (**new Supplementary Fig. 6b**)]. We also generated a new knock-in strain that expresses mNeonGreen-tagged CIL-1 from its endogenous locus (mNeonGreen::CIL-1). However, we could not detect the expression of mNeonGreen::CIL-1 in early embryos, most likely due to very low expression of CIL-1 during embryonic development of *C. elegans*. Overexpressed CIL-1 (in *C. elegans* intestine cells) and INPP5K (in mammalian cells) were both shown to localize to the ER (PMID: 30087126). Thus, although we do not have direct evidence, it is most likely that CIL-1 also localizes to the ER (where PDZD-8 localizes) in early embryos.

Reviewer #2 (Remarks to the Author):

Using *C. elegans* as a model for embryogenesis, the authors have investigated factors that control the levels of PI45P2 in the PM and endosomal pathway and how this affects cortical actinomyosin contraction during cell division. They show that paired KO of two SMP proteins PDZD8 and TEX2 resulted in abnormal accumulation of PI45P2 on large endosomes, and that these endosomes also accumulated proteins involved in actinomyosin contraction. Brood size was only slightly reduced and survival was unaffected. However, ablation of PI45P2 phosphatases OCRL and UNC-26/synaptojanin caused more extensive accumulation of endosomal PI45P2, reduced brood size and a 50% reduction in survival, suggesting two parallel pathways for removal of PI45P2 from endosomes. They further show that the SMP domain of PDZD8 transfers PI45P2 to acceptor liposomes in vitro, and that mutation the SMP domains of PDZD98 and TEX2 in vivo reproduced the PI45P2 accumulation of endosomes observed with the KO animals.

Together with evidence that PDZD8 localized to ER-endosomes, they conclude that the SMP proteins transfer PI45P2 from the endosomes thus preventing buildup and proper actinomyosin activity and cytokinesis. The study is convincing in the sense that it shows that a PDZD8/TEX2-dependent pathway functions in parallel with endosomal phosphatases to maintain PI45P2 levels in the PM/endosomal pathway, which is critical in early stages of embryogenesis.

We thank the reviewer for all the constructive comments to improve our manuscript.

There are several concerns regarding the evidence the authors have presented to support the role of SMP proteins as specific PI45P2 transporters at ER-endosomes contact sites.

1. Both PDZD8 and Tex2 are clearly ER proteins but the evidence that they interact with Rab7 positive endosomes is based on overexpression in Cos cells. The authors need to show more definitive evidence that endosomes containing endogenous Rab7 interact with PDZD8/Tex2 at the ER.

Reply: We thank the reviewer for these comments. To determine the subcellular localization of RAB-7 at endogenous expression levels, we generated a new knock-in strain that expresses mCherry-tagged RAB-7 from its endogenous locus (mCherry::RAB-7). We found that the ER, visualized by the ER marker, GFP::C34B2.10, frequently wraps around late endosomes that are marked by endogenously tagged mCherry::RAB-7, supporting the occurrence of ER-late endosome contacts in early embryos (**new Fig. 6b**). We also generated a new knock-in strain that expresses mNeonGreen-tagged PDZD-8 from its endogenous locus (PDZD-8::mNeonGreen) (in this new strain, we inserted a sequence encoding mNeonGreen in-frame in the last common exon of the *pdzd-8* gene). Using this new strain, we found that PDZD-8 localizes to the ER in early embryos [based on extensive overlap of PDZD-8::mNeonGreen signal with the ER marker, mCherry::SP-12 (**new Supplementary Fig. 6b**)]. These newly generated strains allowed us to examine the association of PDZD-8 with the late endosome marker, RAB-7, both at endogenous expression levels in *C. elegans* early embryos. We found that distinct PDZD-8::mNeonGreen puncta were present on a small fraction of mCherry::RAB-7-positive late endosomes. The association of PDZD-8::mNeonGreen puncta with mCherry::RAB-7-positive late endosomes was often transient, with the longest association lasting up to 45 seconds during imaging period. Because PDZD-8::mNeonGreen localizes to the ER, these results support the presence of endogenous PDZD-8 at ER-late endosome contacts in early embryos. We now included these new data in **new Fig. 6c, d**.

In contrast, TEX-2, visualized by endogenously tagged TEX-2::EGFP, showed no particular association with late endosomes marked by endogenously tagged mCherry::RAB-7 in early

embryos (**new Supplementary Fig. 6d**). This is consistent with our localization analysis of exogenously expressed TEX-2 in COS-7 cells, which did not show association of TEX-2 with particular endosome markers (**old Figure S6B, now in Supplementary Fig. 6c**). Although we could not detect particular association of TEX-2 with endosomes, TEX-2 may transiently populate ER-endosome contacts to help maintain endosomal PI(4,5)P₂ homeostasis [our genetic study supports the redundant role of PDZD-8 and TEX-2 in the regulation of endosomal PI(4,5)P₂ homeostasis]. We updated our model figure (**new Fig. 8d**) and discussed this point further in the revised manuscript (**Page 16** of the revised manuscript).

2. While the authors show that the isolated SMP domain of PDZD8 can transfer PI45P₂ *in vitro*, other studies have suggested that this domain is non-specific and transfers other phospholipids. If the binding of the SMP domain is non-specific, then the authors are measuring a minor activity for PI45P₂ that may not be physiologically relevant if there is competition for binding/transfer by other more abundant lipids. Indeed the transfer of PI45P₂ seems rather slow and does not go to equilibrium (actually on about 30%) even after 60 min. The authors must show that the transfer of PI45P₂ is a preferred ligand relative to other PIPs and phospholipids.

Reply: We thank the reviewer for these thoughtful comments. We performed additional *in vitro* lipid transport assays to further examine the preference/selectivity of the purified SMP domain of PDZD-8 (SMP_{PDZD-8}). First, we measured transport efficiency of other PIPs, namely PI3P, PI4P, PI(3,5)P₂, and PI(3,4,5)P₃, using the same FRET-based *in vitro* lipid transport assay that we used for measuring PI(4,5)P₂ transport [NBD-labelled PH domain of FAPP protein can recognize any one of the PIPs on liposomal membranes (e.g., PMID: 30783101)]. Second, we used a similar FRET-based approach and examined whether SMP_{PDZD-8} could transport phosphatidylserine (PS) between membranes [for this assay, we newly generated NBD-labelled C2 domain of Lactadherin, which can recognize PS on liposomal membranes (e.g., PMID: 26206936)]. Third, we measured transport efficiency of other NBD-labelled lipids, including NBD-PA and NBD-ceramide (in addition to NBD-PE that we showed in the original manuscript). Finally, we measured transport efficiency of cholesterol, using FRET-based dehydroergosterol (DHE) transport assay that we had previously used for measuring GRAMD1-dependent sterol transport (PMID: 31724953; PMID: 33604931) (DHE is a naturally occurring fluorescence sterol often used in *in vitro* FRET-based sterol transport assays).

We found that SMP_{PDZD-8} transported various PIPs, including PI(3)P, PI(4)P, and PI(3,5)P₂, and PS with similar transport efficiency to PI(4,5)P₂ while it transported PI(3,4,5)P₃ less efficiently (**new Fig. 7d and Supplementary Fig. 7c, d**). SMP_{PDZD-8} was not capable of transporting NBD-PE and NBD-ceramide between liposomes but was capable of moderately facilitating NBD-PA transport (**new Supplementary Fig. 7e-h**). No DHE transport was detected by SMP_{PDZD-8} (**new Supplementary Fig. 7i**). Based on these results, SMP_{PDZD-8} transports various PIPs, including PI(4,5)P₂, and other phospholipids, such as PS and possibly PA, but not cholesterol and some phospholipids. We now toned down our statements regarding the preference of SMP_{PDZD-8} toward PI(4,5)P₂ and stated its preference/selectivity more clearly throughout the revised manuscript. We also discussed a possibility that PDZD-8 may also play a role in regulating the distribution of other lipids in addition to PI(4,5)P₂ in the revised manuscript (**Page 16** of the revised manuscript: the following new sentence was inserted "As the SMP domain of PDZD-8 is able to transport various lipids between membranes *in vitro*, PDZD-8 may also regulate the distribution of other lipids in addition to PI(4,5)P₂.").

We acknowledge the slow PI(4,5)P₂ transport mediated by SMP_{PDZD-8}. A more physiological membrane lipid composition or a native environment (e.g., membrane contact sites) may allow the SMP domain of PDZD-8 to transport PI(4,5)P₂ and possibly other PIPs more efficiently inside the cell (e.g., PMID: 33605998). To examine if SMP domain-dependent

PI(4,5)P₂ transport could be enhanced in the context of full-length protein, we attempted to purify near full-length PDZD-8, but purification was not successful.

Finally, we tested whether the presence of other anionic phospholipids in donor membranes would affect the efficiency of PI(4,5)P₂ transport mediated by SMP_{PDZD-8}. Addition of either one of PS, PI, or PA, in donor liposomes did not reduce the efficiency of PI(4,5)P₂ transport between liposomes, suggesting that the presence of other anionic lipids in donor membranes does not interfere or compete with the property of SMP_{PDZD-8} to transport PI(4,5)P₂. These new data are now included in **new Supplementary Fig. 7j, k**.

3. In general, the conclusion presented in the discussion (notably at the bottom of page 14 where they claim that PDZD8/TEX2 transfer PI45P2 to the ER etc...) is not convincing. Key evidence of PI45P2 transfer to the ER would be accumulation in that organelle or removal by a ER-associated phosphatase (as they speculate in the Discussion, Fig. 7E). Without evidence of a vectoral transfer to the ER (or a clear preference for PI45P2, see above) it is not possible to conclude that PDZD8/TEX2 are involved in PI45P2 transport. The SMP proteins could be tethering factors that facilitate membrane interactions guided by PI45P2 sensing. In the context of the full-length ER-associated PDZD8, this could be related to a tethering activity at the ER-endosome contact site or to presentation of the lipid to a phosphatase for hydrolysis.

Reply: We agree with the reviewer that PDZD-8 and TEX-2 could potentially act as a tethering factor at ER-endosome contact sites and indirectly promote lipid exchange (or lipid presentation) that are mediated by other proteins. The possibility that PDZD-8 and TEX-2 are involved in presentation of PI(4,5)P₂ to a phosphatase for hydrolysis can also not be excluded (we had discussed this possibility in our original manuscript). Currently, we do not have direct evidence that supports a vectoral transfer of PI(4,5)P₂ from endosomes to the ER by PDZD-8/TEX-2. Thus, we toned down our statements throughout the revised manuscript and suggested such function (i.e., the vectoral transport mediated by PDZD-8) as one of the potential functions of PDZD-8 in the discussion. We also stated the limitation of our current studies more clearly. Major changes/points are shown and highlighted below.

Original sentences: "Further, we found that the SMP domain of PDZD-8 was able to extract and transport PI(4,5)P₂ between membranes. Based on these results, we propose that PDZD-8 acts at ER-late endosome contacts to extract PI(4,5)P₂ from late endosomes via its SMP domain, which can capture this lipid." (Page 14 of the original manuscript)

New sentences: "Further, we found that the SMP domain of PDZD-8 was able to extract and transport various PIPs, including PI(4,5)P₂, between membranes *in vitro*. Based on these results, we suggest that PDZD-8 may act at ER-late endosome contacts to extract PI(4,5)P₂ from late endosomes via its SMP domain, while a possibility that PDZD-8 may primarily act as a tethering protein for ER-late endosome contacts cannot be excluded (**Fig. 8d**)." (Page 16 of the revised manuscript)

Original sentences: "There are two possible fates for PI(4,5)P₂ that is extracted from endosomes. First, PI(4,5)P₂ could be transported to the ER via the SMP domains of PDZD-8 (and/or TEX-2) for its dephosphorylation at the ER (**Figure 7E**). Second, PI(4,5)P₂ could be presented to OCRL-1 and UNC-26 via the SMP domains for its rapid dephosphorylation on endosomes. As the phenotype resulting from the combined depletion of PDZD-8, TEX-2, OCRL-1, and UNC-26 was much stronger than the phenotype resulting from the depletion of OCRL-1 and UNC-26 alone, we favor the first scenario in which other PI(4,5)P₂ phosphatases act on the ER "*in cis*" to remove PI(4,5)P₂ that is transported from endosomes." (Page 15 of the original manuscript)

New sentences: "There are two possible fates for PI(4,5)P₂ that is extracted from endosomes. First, PI(4,5)P₂ could be transported to the ER for its dephosphorylation at the ER. Second, PI(4,5)P₂ could be presented to OCRL-1 and UNC-26 for its rapid dephosphorylation on endosomes. As the phenotype resulting from the combined depletion of PDZD-8, TEX-2, OCRL-1, and UNC-26 was much stronger than the phenotype resulting from the depletion of OCRL-1 and UNC-26 alone, we favor the first scenario in which other PI(4,5)P₂ phosphatases act on the ER "*in cis*" to remove PI(4,5)P₂ that is transported from endosomes, although we do not have direct evidence supporting a vectorial transfer of PI(4,5)P₂ from endosomes to the ER by PDZD-8/TEX-2 in the current study." (Page 16 of the revised manuscript)

Reviewer #3 (Remarks to the Author):

The manuscript by Jeyasimman et al provides evidence for partly redundant mechanisms for removal of PI(4,5)P₂ from endosomal membranes in an intact in vivo model. The authors identify ER associated lipid transfer proteins PDZD-8 and TEX-2 in this process, along with PI5-phosphatases OCRL-1 and UNC-26. Importantly the authors show that without these PI(4,5)P₂ regulators actomyosin contractile machinery is inappropriately recruited to endosomes, leading to failure in actomyosin driven cytokinesis during embryogenesis. The authors further provide in vitro evidence that PDZD-8 can transport PI(4,5)P₂ between membranes. Overall this work appears well done and is likely to interest a broad group of cell and developmental biologists.

We thank the reviewer for these very positive comments.

Some weaknesses are detailed below:

(1) *The biggest issue that needs to be addressed relates to the claim that PI(4,5)P₂ overaccumulation on endosomes affects endosome identity.*

The authors write “Indeed, some large PI(4,5)P₂-positive vesicles were transiently decorated by GFP::RAB-7. However, a significant fraction of the large PI(4,5)P₂-positive vesicles were not decorated by GFP::RAB-7. The aberrant accumulation of PI(4,5)P₂ in late endosomes may therefore have caused massive disruption to the identity of endosomes. To this end, we assessed the potential co-localization of large PI(4,5)P₂-positive vesicles with two other endosomal markers, namely GFP::RAB-5 for early endosomes and GFP::RAB-11.1 for recycling endosomes (Figure 7A). Strikingly, a significant fraction of these large PI(4,5)P₂-positive vesicles were also transiently decorated by GFP::RAB-5 and GFP::RAB-11.1, supporting the notion that massive accumulation of PI(4,5)P₂ in late endosomes disrupted their identity.”

This statement indicates a misunderstanding of endosome identity. It is entirely normal for endosomes to display multiple sequentially acting Rab proteins on their limiting membranes. This has been known for more than 20 years, as originally shown by Zerial and colleagues (Sönnichsen et al, 2000; PMID 10811830). This would have been clear if control experiments with the same markers was performed on wild-type embryos. The authors would need extensive additional experimentation to support the claim of a change in endosome identity, which should also include tests of endosome function.

Reply: We thank the reviewer for these comments. We acknowledge that our statement regarding the endosome identity was not appropriate. Careful analysis of wild-type early embryos revealed that each endosome was indeed often positive for multiple RABs as the reviewer pointed out. Thus, we fixed the above-mentioned statement as shown below and updated other statements, where we mentioned endosome identity, throughout the revised manuscript.

Old sentences: “Indeed, some large PI(4,5)P₂-positive vesicles were transiently decorated by GFP::RAB-7. However, a significant fraction of the large PI(4,5)P₂-positive vesicles were not decorated by GFP::RAB-7. The aberrant accumulation of PI(4,5)P₂ in late endosomes may therefore have caused massive disruption to the identity of endosomes. To this end, we assessed the potential co-localization of large PI(4,5)P₂-positive vesicles with two other endosomal markers, namely GFP::RAB-5 for early endosomes and GFP::RAB-11.1 for recycling endosomes (**Figure 7A**). Strikingly, a significant fraction of these large PI(4,5)P₂-positive vesicles were also transiently decorated by GFP::RAB-5 and GFP::RAB-11.1, supporting the notion that massive accumulation of PI(4,5)P₂ in late endosomes disrupted their identity (**Figure 7A**).” (Page 11-12 of the original manuscript)

New sentences: “Indeed, 19.4 ± 1.2 % of large PI(4,5)P₂-positive vesicles were transiently decorated by GFP::RAB-7 in mutant embryos (n=14 embryos) (**Supplementary Fig. 8a, b**). However, a significant fraction of the large PI(4,5)P₂-positive vesicles was not decorated by GFP::RAB-7. Thus, we assessed the potential co-localization of large PI(4,5)P₂-positive vesicles with two other endosomal markers, namely GFP::RAB-5 for early endosomes and GFP::RAB-11.1 for recycling endosomes (**Supplementary Fig. 8a**). A significant fraction of the large PI(4,5)P₂-positive vesicles was transiently decorated by GFP::RAB-5 [11.1 ± 1.0 % (n=12 embryos)] and GFP::RAB-11.1 [39.2 ± 3.4 % (n=14 embryos)], suggesting that PI(4,5)P₂ accumulates throughout the endosomal system in the absence of PDZD-8, TEX-2, and the PI(4,5)P₂ phosphatases (**Supplementary Fig. 8a, b**).” (**Page 13** of the revised manuscript)

There are endosomal cargo markers (e.g. GFP::CAV-1) to gain insight into endosomal function, at least the degradative aspect) in the C. elegans embryo.

Reply: We thank the reviewer for this excellent suggestion. Using GFP-tagged CAV-1 (CAV-1::GFP) as model cargo for degradation, we assessed potential changes in the degradative capacity of endosomes in early embryos from OCRL-1 RNAi-treated *unc-26* mutants lacking PDZD-8 and TEX-2 [*unc-26; DKO; ocr1-1 (RNAi)*]. In wild-type (WT) embryos, CAV-1::GFP was rapidly endocytosed from the plasma membrane (PM) and degraded before completion of cytokinesis as previously reported (PMID: 16672374). In embryos from *unc-26; DKO; ocr1-1 (RNAi)*, however, we found that degradation of CAV-1::GFP was markedly delayed and that undegraded CAV-1::GFP signals were clearly present in cytoplasm even after completion of cytokinesis (**new Fig. 8a, b**). These results support major disruption of the degradative capacity of the endosomal system in the absence of PDZD-8, TEX-2, and the PI(4,5)P₂ phosphatases. We also noted that PM CAV-1::GFP signals were more sustained in early embryos from *unc-26; DKO; ocr1-1 (RNAi)* compared to WT, indicating that endocytosis of CAV-1::GFP was also delayed in these mutant embryos (**new Supplementary Fig. 9c**). PM PI(4,5)P₂ plays a critical role in clathrin-mediated endocytosis via its property to recruit various endocytic factors to the PM (PMID: 29661000; PMID: 29661000; PMID: 21779028). Thus, aberrant cytoplasmic accumulation of PI(4,5)P₂ in these mutant embryos (e.g., original **Fig. 4f**) may have also resulted in defects in clathrin-mediated endocytosis, which is required for CAV-1::GFP internalization. We included these new results in **new Fig. 8a-8c and Supplementary Fig. 9a-c**.

I am also concerned that it is only a small fraction of endosomes that appear enlarged and PI(4,5)P₂ positive – a feature of the data that was not clearly acknowledged. What fraction of late endosomes overaccumulate PI(4,5)P₂?

Reply: Thank you for this suggestion. We quantified our original data (**previously in Figure 7, now in Supplementary Fig. 8**), using the Manders' Coefficients. Indeed, only a small fraction (3.5 ± 0.3 %) of late endosomes marked by GFP-tagged RAB-7 (GFP::RAB-7) showed accumulation of PI(4,5)P₂ [monitored by mCherry-tagged PH domain of PLC δ 1 (mCherry::PH^{PLC δ 1})], while 19.4 ± 1.2 % of PI(4,5)P₂-positive vesicles were positive for GFP::RAB-7 (n=14 embryos). In contrast, approximately 8.6 ± 1.4 % of recycling endosomes marked by GFP-tagged RAB-11 (GFP::RAB-11) showed accumulation of PI(4,5)P₂, while 39.2 ± 3.4 % of PI(4,5)P₂-positive vesicles were positive for GFP::RAB-11 (n=14 embryos).

Among the total pool of late endosomes marked by “endogenously tagged mCherry::RAB-7” (using a new knock-in strain that we generated for the revision), 30.8 ± 3.1 % of them showed accumulation of CAV-1::GFP (n=13 embryos). $33\% \pm 3.6$ % of the total pool of CAV-1::GFP-positive vesicles were positive for PI(4,5)P₂ (n=15 embryos) (**new Fig. 8c and Supplementary Fig. 9b**). With simple calculation, this can be converted to approximately 10% ($30.8\% \times 33\%$) of the total pool of mCherry::RAB-7-positive late endosomes being

positive for PI(4,5)P₂. Thus, there is a possibility that above-mentioned experiments (**previously in Figure 7, now in Supplementary Fig. 8**) may have underestimated the accumulation of PI(4,5)P₂ in late endosomes. This may be partly due to the potential difference between single copy insertion of a transgene for GFP::RAB-7 (typically resulting in overexpression) in **Supplementary Fig. 8** and endogenous tagging for mCherry::RAB-7 (endogenous expression levels) in **new Fig. 8 and Supplementary Fig. 9** used for these experiments.

(2) *The data on lipid transfer in vitro is a nice addition to this paper, but more data is needed to understand the specificity of PDZD-8 SMP-domain-mediated transfer. At the very least other negatively charged phospholipids should be tested, including PA, PS, PI, PI4P, PI3P, and PI(3,5)P₂.*

Reply: We thank the reviewer for these thoughtful comments. We performed additional *in vitro* lipid transport assays to further examine the preference/selectivity of the purified SMP domain of PDZD-8 (SMP_{PDZD-8}). First, we measured transport efficiency of other PIPs, namely PI3P, PI4P, PI(3,5)P₂, and PI(3,4,5)P₃, using the same FRET-based *in vitro* lipid transport assay that we used for measuring PI(4,5)P₂ transport [NBD-labelled PH domain of FAPP protein can recognize any one of the PIPs on liposomal membranes (e.g., PMID: 30783101)]. Second, we used a similar FRET-based approach and examined whether SMP_{PDZD-8} could transport phosphatidylserine (PS) between membranes [for this assay, we newly generated NBD-labelled C2 domain of Lactadherin, which can recognize PS on liposomal membranes (e.g., PMID: 26206936)]. Third, we measured transport efficiency of other NBD-labelled lipids, including NBD-PA and NBD-ceramide (in addition to NBD-PE that we showed in the original manuscript). Finally, we measured transport efficiency of cholesterol, using FRET-based dehydroergosterol (DHE) transport assay that we had previously used for measuring GRAMD1-dependent sterol transport (PMID: 31724953; PMID: 33604931) (DHE is a naturally occurring fluorescence sterol often used in *in vitro* FRET-based sterol transport assays). Because NBD-labelled PI (nor a probe that can selectively recognize PI) was not available, we attempted to measure PI transport using an end-point assay with thin-layer chromatography (TLC), but the experiment did not work due to relatively low sensitivity of this method.

We found that SMP_{PDZD-8} transported various PIPs, including PI(3)P, PI(4)P, and PI(3,5)P₂, and PS with similar transport efficiency to PI(4,5)P₂ while it transported PI(3,4,5)P₃ less efficiently (**new Fig. 7d and Supplementary Fig. 7c, d**). SMP_{PDZD-8} was not capable of transporting NBD-PE and NBD-ceramide between liposomes but was capable of moderately facilitating NBD-PA transport (**new Supplementary Fig. 7e-h**). No DHE transport was detected by SMP_{PDZD-8} (**new Supplementary Fig. 7i**). Based on these results, SMP_{PDZD-8} transports various PIPs, including PI(4,5)P₂, and other phospholipids, such as PS and possibly PA, but not cholesterol and some phospholipids. We now toned down our statements regarding the preference of SMP_{PDZD-8} toward PI(4,5)P₂ and stated its preference/selectivity more clearly throughout the revised manuscript. We also discussed a possibility that PDZD-8 may also play a role in regulating the distribution of other lipids in addition to PI(4,5)P₂ in the revised manuscript (**Page 16** of the revised manuscript: the following new sentence was inserted "As the SMP domain of PDZD-8 is able to transport various lipids between membranes *in vitro*, PDZD-8 may also regulate the distribution of other lipids in addition to PI(4,5)P₂.").

Finally, we tested whether the presence of other anionic phospholipids in donor membranes would affect the efficiency of PI(4,5)P₂ transport mediated by SMP_{PDZD-8}. Addition of either one of PS, PI, or PA, in donor liposomes did not reduce the efficiency of PI(4,5)P₂ transport between liposomes, suggesting that the presence of other anionic lipids in donor membranes does not interfere or compete with the property of SMP_{PDZD-8} to transport PI(4,5)P₂ (**new Supplementary Fig. j, k**).

(3) Colocalization experiments in general should be quantified for a population (e.g. Manders or Pierson's coefficients), not simply shown as single examples with line-scans. In general quantification was better performed for cytokinesis proteins and less well for endosome proteins (e.g. Fig 6 and 7).

Reply: Thank you for this suggestion. We quantified our original data (previously in Figure 6 and Figure 7), using the Manders' Coefficients, and add the quantified results in the new Figures (**now in new Fig. 6 and Supplementary Fig. 8**).

Reviewer #4 (Remarks to the Author):

Jayasimman et al., investigated the functions of SMP proteins in *C.elegans* early embryo. They systematically tested the combinations of SMP protein KO lines and found that PDZD-8 and TEX-2 function redundantly in removing PI(4,5)P2 from the cytoplasmic compartment. The cytoplasmic PI(4,5)P2 in the PDZD-8; Tex-2 double knockouts is overlapped with NMY-2 and mostly with the recycling endosome marker RAB-11. The accumulation of PI(4,5)P2 was enhanced by additional deletion of PI(4,5)P2 phosphatases, UNC-26 and OCRL-1. In the *unc-26*; DKO embryo treated with OCRL-1 RNAi, the cortical ruffling was impaired and also the percentage of eggs that hatched was reduced. The *in vitro* analysis showed that the SMP domain of PDZD-8 transfers PI(4,5)P2. Finally, they showed that SMP domains of PDZD-8 and TEX-2 are required for removing PI(4,5)P2 from the cytoplasmic compartment. Non-vesicular lipid transport at the membrane contact sites is an important topic in the field and the link between the SMP proteins and PI(4,5)P2 accumulation is an interesting finding. The caveat is that the experiments explaining the roles of SMP domain proteins are mostly done *in vitro* or poorly done using mammalian cell lines. This hampers the soundness of the hypothesis the paper claims.

We thank the reviewer for all the constructive comments to improve our manuscript.

Major points.

To understand how cytoplasmic PI(4,5)P2 is generated and removed by SMP proteins, it is necessary to clarify the distribution of TEX-2 and PDZD-8 in the *C.elegans* embryo. The localization of ectopically expressed PDZD-8 together with again ectopically expressed Rab-7 in the mammalian system does not reflect the endogenous localization of PDZD-8 in *C.elegans* embryo. In addition, in the experiment using mammalian cells, TEX-2 and PDZD-8 localize in different compartments. This is not consistent with the model the authors are proposing, namely the redundant function of TEX-2 and PDZD-8 (seen in Figure 7E). These points need to be addressed adequately.

Reply: We thank the reviewer for these comments. To determine the subcellular localization of RAB-7 at endogenous expression levels, we generated a new knock-in strain that expresses mCherry-tagged RAB-7 from its endogenous locus (mCherry::RAB-7). We found that the ER, visualized by the ER marker, GFP::C34B2.10, frequently wraps around late endosomes that are marked by endogenously tagged mCherry::RAB-7, supporting the occurrence of ER-late endosome contacts in early embryos (**new Fig. 6b**). We also generated a new knock-in strain that expresses mNeonGreen-tagged PDZD-8 from its endogenous locus (PDZD-8::mNeonGreen) (in this new strain, we inserted a sequence encoding mNeonGreen in-frame in the last common exon of the *pdzd-8* gene). Using this new strain, we found that PDZD-8 localizes to the ER in early embryos [based on extensive overlap of PDZD-8::mNeonGreen signal with the ER marker, mCherry::SP-12 (**new Supplementary Fig. 6b**)]. These newly generated strains allowed us to examine the association of PDZD-8 with the late endosome marker, RAB-7, both at endogenous expression levels in *C. elegans* early embryos. We found that distinct PDZD-8::mNeonGreen puncta were present on a small fraction of mCherry::RAB-7-positive late endosomes. The association of PDZD-8::mNeonGreen puncta with mCherry::RAB-7-positive late endosomes was often transient, with the longest association lasting up to 45 seconds during imaging period. Because PDZD-8::mNeonGreen localizes to the ER, these results support the presence of endogenous PDZD-8 at ER-late endosome contacts in early embryos. We now included these new data in **new Fig. 6c, d**.

In contrast, TEX-2, visualized by endogenously tagged TEX-2::EGFP, showed no particular association with late endosomes marked by endogenously tagged mCherry::RAB-7 in early embryos (**new Supplementary Fig. 6d**). This is consistent with our localization analysis of exogenously expressed TEX-2 in COS-7 cells, which did not show association of TEX-2 with

particular endosome markers (**old Figure S6B, now in Supplementary Fig. 6c**). Although we could not detect particular association of TEX-2 with endosomes, TEX-2 may transiently populate ER-endosome contacts to help maintain endosomal PI(4,5)P₂ homeostasis [our genetic study supports the redundant role of PDZD-8 and TEX-2 in the regulation of endosomal PI(4,5)P₂ homeostasis]. We updated our model figure (**new Fig. 8d**) and discussed this point further in the revised manuscript (**Page 16** of the revised manuscript).

One possible experiment to connect the genetic experiments and the phenotype is overexpressing the phosphatases in the DKO. Experiments using ER-anchored phosphatases might support their hypothesis.

Reply: We thank the reviewer for this suggestion. We generated a new strain that expresses a 5-phosphatase domain of UNC-26/Synaptojanin that is targeted to the ER (by fusing the 5-phosphatase domain to the cytosolic region of a ER protein, CP450) using a single copy insertion of a transgene (MosSCI method) (PMID: 18953339). We then crossed this strain to DKO strain lacking both PDZD-8 and TEX-2 and monitored cytoplasmic PI(4,5)P₂ accumulation in early embryos with a PI(4,5)P₂ biosensor [mCherry-tagged PH domain of PLCδ1 (mCherry::PH^{PLCδ1})]. Compared to control DKO embryos (which shows cytoplasmic PI(4,5)P₂ accumulation), no significant decrease of cytoplasmic PI(4,5)P₂ accumulation was observed in DKO embryos expressing the ER-anchored 5 phosphatase (please see below the results of these experiments). These results suggest that PDZD-8 and TEX-2 are needed to facilitate PI(4,5)P₂ removal from endosomes and that ER-anchored phosphatases (e.g., INPP5K etc.) would most likely act “*in cis*” to dephosphorylate PI(4,5)P₂ on ER membranes (although the identification of such 5-phosphatase requires future investigation).

Expression of ER-anchored 5-phosphatase domain of UNC-26/synaptojanin does not reduce cytoplasmic accumulation of PI(4,5)P₂ in *C. elegans* early embryos lacking PDZD-8 and TEX-2. [mean ± SEM, n=14 (WT), n=16 (DKO), n=15 (DKO; CP450::UNC-26 5'ptase); Tukey's multiple comparisons test, *p=0.0159 (WT vs. DKO), ns denotes not significant]

The interpretation of Figure 4A does not seem to be consistent with their hypothesis to this reviewer. According to their model, even in the presence of PDZD-8 and TEX-2, PI(4,5)P₂ should be transferred from somewhere to the cytoplasmic compartment, and then PI(4,5)P₂ is exposed or further transferred to the ER. Therefore, the accumulation of PI(4,5)P₂ should

be observed in the *unc-26* embryo treated with *ocrl-1* RNAi.

Reply: Thank you for these comments. We actually observed some cytoplasmic accumulation of PI(4,5)P₂ in *unc-26* embryos treated with *ocrl-1* RNAi compared to control embryos (please see the original Video 4). To be clearer, we chose different frames of the same embryo for the relevant figures (**Fig. 4a, 7f**) and updated the relevant sentences as shown below. Even when the two PI(4,5)P₂ phosphatases are depleted, however, PDZD-8 and TEX-2 are present to prevent the build-up of PI(4,5)P₂ in endosomes by their activities to regulate endosomal PI(4,5)P₂ levels (based on our model, which is now more clearly discussed along with other possibilities in the **page 16** of the revised manuscript), and thus, we do not expect to see much accumulation of PI(4,5)P₂ in endosomes in *unc-26* embryos treated with *ocrl-1* RNAi compared to mutant embryos simultaneously lacking PDZD-8, TEX-2, UNC-26 and OCRL-1.

Original sentences: "In WT or *unc-26* mutant early embryos treated with OCRL-1 RNAi, PI(4,5)P₂ and NMY-2 remained tightly associated with the PM; PI(4,5)P₂/NMY-2 did not ectopically accumulate in the cytoplasm (except some occasional cytoplasmic PI(4,5)P₂-positive structures in *unc-26* mutant embryos treated with OCRL-1 RNAi) (**Figures 4A-C, S4A; Video 4**)." (Page 8 of the original manuscript)

New sentences: "In WT or *unc-26* mutant early embryos treated with OCRL-1 RNAi, PI(4,5)P₂ and NMY-2 remained tightly associated with the PM, although some cytoplasmic PI(4,5)P₂-positive structures were observed in *unc-26* mutant embryos treated with OCRL-1 RNAi compared to WT embryos treated with OCRL-1 RNAi (**Fig. 4a-c and Supplementary Fig. 4a; Supplementary Movie 4; compare with Fig. 1b and Supplementary Movie 1**)" (Page 8 of the revised manuscript)

The paper would be strengthened if the in vitro lipid transport assay was more thorough. The specificity of SMP's lipid binding has been discussed but is still very controversial. Even though the paper claims PDZD-8 transports PI(4,5)P₂ selectively, they examined only PE as a control. At least other lipid species already claimed in other papers (Jeong et al., 2017, Shirane et al. 2020, etc.) should be tested. If C.elegans PDZD-8 and TEX-2 specifically transfer PI(4,5)P₂, the result will reinforce their model.

Reply: We thank the reviewer for these thoughtful comments. We performed additional *in vitro* lipid transport assays to further examine the preference/selectivity of the purified SMP domain of PDZD-8 (SMP_{PDZD-8}). First, we measured transport efficiency of other PIPs, namely PI3P, PI4P, PI(3,5)P₂, and PI(3,4,5)P₃, using the same FRET-based *in vitro* lipid transport assay that we used for measuring PI(4,5)P₂ transport [NBD-labelled PH domain of FAPP protein can recognize any one of the PIPs on liposomal membranes (e.g., PMID: 30783101)]. Second, we used a similar FRET-based approach and examined whether SMP_{PDZD-8} could transport phosphatidylserine (PS) between membranes [for this assay, we newly generated NBD-labelled C2 domain of Lactadherin, which can recognize PS on liposomal membranes (e.g., PMID: 26206936)]. Third, we measured transport efficiency of other NBD-labelled lipids, including NBD-PA and NBD-ceramide (in addition to NBD-PE that we showed in the original manuscript). Finally, we measured transport efficiency of cholesterol, using FRET-based dehydroergosterol (DHE) transport assay that we had previously used for measuring GRAMD1-dependent sterol transport (PMID: 31724953; PMID: 33604931) (DHE is a naturally occurring fluorescence sterol often used in *in vitro* FRET-based sterol transport assays).

We found that SMP_{PDZD-8} transported various PIPs, including PI(3)P, PI(4)P, and PI(3,5)P₂, and PS with similar transport efficiency to PI(4,5)P₂ while it transported PI(3,4,5)P₃ less efficiently (**new Fig. 7d and Supplementary Fig. 7c, d**). SMP_{PDZD-8} was not capable of transporting NBD-PE and NBD-ceramide between liposomes but was capable of moderately

facilitating NBD-PA transport (**new Supplementary Fig. 7e-h**). No DHE transport was detected by SMP_{PDZD-8} (**new Supplementary Fig. 7i**). Based on these results, SMP_{PDZD-8} transports various PIPs, including PI(4,5)P₂, and other phospholipids, such as PS and possibly PA, but not cholesterol and some phospholipids. We now toned down our statements regarding the preference of SMP_{PDZD-8} toward PI(4,5)P₂ and stated its preference/selectivity more clearly throughout the revised manuscript. We also discussed a possibility that PDZD-8 may also play a role in regulating the distribution of other lipids in addition to PI(4,5)P₂ in the revised manuscript (**Page 16** of the revised manuscript: the following new sentence was inserted "As the SMP domain of PDZD-8 is able to transport various lipids between membranes *in vitro*, PDZD-8 may also regulate the distribution of other lipids in addition to PI(4,5)P₂.").

Finally, we tested whether the presence of other anionic phospholipids in donor membranes would affect the efficiency of PI(4,5)P₂ transport mediated by SMP_{PDZD-8}. Addition of either one of PS, PI, or PA, in donor liposomes did not reduce the efficiency of PI(4,5)P₂ transport between liposomes, suggesting that the presence of other anionic lipids in donor membranes does not interfere or compete with the property of SMP_{PDZD-8} to transport PI(4,5)P₂ (**new Supplementary Fig. 7j, k**). We could not purify the SMP domain of TEX-2 despite multiple attempts.

In line with this, the statements 'the SMP domain of PDZD-8 selectively transports PI(4,5)P₂ between membranes' and 'PDZD-8 transports PI(4,5)P₂ from endosomes to the ER via the SMP domain in vivo to maintain endosomal PI(4,5)P₂ homeostasis' are overstatements.

Reply: We agree with this reviewer. Based on our new results, we revised these statements as shown below.

Original sentences: "... demonstrate that the SMP domain of PDZD-8 selectively transports PI(4,5)P₂ between membranes. We further show that PDZD-8 transports PI(4,5)P₂ from endosomes to the ER via the SMP domain *in vivo* to maintain endosomal PI(4,5)P₂ homeostasis, providing novel insights into the function of PDZD-8 at ER-late endosome contacts." (Page 14 of the original manuscript)

New sentences: "Our current study shows that the SMP domain of PDZD-8 transports PIPs, including PI(4,5)P₂, *in vitro*. Furthermore, we show that PDZD-8 function together with TEX-2 and PI(4,5)P₂ phosphatases to play a critical role in maintaining endosomal PI(4,5)P₂ homeostasis via its SMP domain in *C. elegans* early embryos, providing novel insights into the physiological function of PDZD-8 at ER-late endosome contacts." (**Page 17** of the revised manuscript)

Minor point.

The data presented show that PDZD-8, TEX-2 DKO is a prerequisite for the phosphatases KO to show the PI(4,5)P₂ accumulation phenotype. In this regard, "PDZD-8 and TEX-2 act in parallel to the PI(4,5)P₂ phosphatases" is misleading.

Reply: Thank you for pointing it out. To avoid confusion, we changed "act in parallel to" to "act together" throughout the revised manuscript.

REVIEWERS' COMMENTS

Reviewer #1 (Remarks to the Author):

The authors have addressed all of my concerns very well. The amount of work completed under the current covid situation is remarkable. The paper should now be published without further delay.

Reviewer #2 (Remarks to the Author):

The authors have gone to exceptional effort to address all my concerns with additional convincing experiments showing the localization and transfer activities of Tex-2 and PDZ-8, and accompanying modifications to the text.

Reviewer #3 (Remarks to the Author):

The authors have addressed all of my concerns. The manuscript is significantly improved.

Reviewer #4 (Remarks to the Author):

The fact that three out of four reviewers asked to examine the localization of PDZD8 and Tex2 in *C. elegans*, the localization of these proteins is one of the most important aspects to describing the mechanism of PI(4,5)P2 removal from endosomes. It is indeed an appropriate strategy to investigate the localization of PDZD8 and Tex2 in the newly generated knock-in strains. However, contrary to the authors' claim, the localization of PDZD8 on the endosome in Fig. 6c and d appears rather as a random encounter of these two proteins than a specific recruitment of PDZD8 on (to the surface of?) the endosome. Thus, the Summary sentence "PDZD-8 localizes to sites of contact between the ER and late endosomes and regulates endosomal PI(4,5)P2 levels via its SMP domain" and the corresponding sentence in the Abstract should be removed. Accordingly, the figure showing the localization of overexpressed PDZD8 in Cos7 cells (Fig. 6a) should be moved to a supplementary Figure. If the authors insist on their interpretation that this is a specific recruitment, then, as the reviewer 3 pointed out, convincing statistical data ought to be provided. Otherwise, the mechanism of endosomal PI(4,5)P2 regulation should be left open to further discussion and interpretation.

REBUTTAL TO THE COMMENTS RAISED BY THE REVIEWERS

We thank the editor and the reviewers for the comments and suggestions. We have addressed the remaining concerns with changes to the texts.

-Sentences from the reviewers' comments are *in italics*
-Our responses are in blue

Reviewer #1 (Remarks to the Author):

The authors have addressed all of my concerns very well. The amount of work completed under the current covid situation is remarkable. The paper should now be published without further delay.

Reply: Thank you for the support. We thank the reviewer for constructive suggestions throughout the revision process.

Reviewer #2 (Remarks to the Author):

The authors have gone to exceptional effort to address all my concerns with additional convincing experiments showing the localization and transfer activities of Tex-2 and PDZ-8, and accompanying modifications to the text.

Reply: We thank the reviewer for all the constructive comments and suggestions to improve our manuscript.

Reviewer #3 (Remarks to the Author):

The authors have addressed all of my concerns. The manuscript is significantly improved.

Reply: We thank the reviewer for all the suggestions and critical comments on the statistical analysis to improve our manuscript.

Reviewer #4 (Remarks to the Author):

The fact that three out of four reviewers asked to examine the localization of PDZD8 and Tex2 in C. elegans, the localization of these proteins is one of the most important aspects to describing the mechanism of PI(4,5)P2 removal from endosomes. It is indeed an appropriate strategy to investigate the localization of PDZD8 and Tex2 in the newly generated knock-in strains.

Reply: We thank the reviewer for these comments.

However, contrary to the authors' claim, the localization of PDZD8 on the endosome in Fig. 6c and d appears rather as a random encounter of these two proteins than a specific recruitment of PDZD8 on (to the surface of?) the endosome. Thus, the Summary sentence "PDZD-8 localizes to sites of contact between the ER and late endosomes and regulates endosomal PI(4,5)P2 levels via its SMP domain" and the corresponding sentence in the Abstract should be removed.

Reply: We removed the section of "Summary" entirely. We toned down our statement in the abstract by removing the yellow highlighted phrase as shown below.

Old sentence: “PDZD-8 localizes to the endoplasmic reticulum (ER) and regulates endosomal PI(4,5)P₂ levels via its lipid harboring SMP domain at ER-late endosome contacts.”

New sentence: “PDZD-8 localizes to the endoplasmic reticulum (ER) and regulates endosomal PI(4,5)P₂ levels via its lipid harboring SMP domain.”

Accordingly, the figure showing the localization of overexpressed PDZD8 in Cos7 cells (Fig. 6a) should be moved to a supplementary Figure.

Reply: We moved the relevant figure (previous Fig 6a) to supplementary figure (**new Supplementary Fig. 6c**) and updated relevant parts of the main text and figure legends.

If the authors insist on their interpretation that this is a specific recruitment, then, as the reviewer 3 pointed out, convincing statistical data ought to be provided. Otherwise, the mechanism of endosomal PI(4,5)P₂ regulation should be left open to further discussion and interpretation.

Reply: Thank you for the suggestion. We toned down our statement as shown above to let the mechanism of endosomal PI(4,5)P₂ regulation be open for further discussion and interpretation.